# Microbiome-pathogen interactions drive epidemiological dynamics of antibiotic resistance: A modeling study applied to nosocomial pathogen control

David RM Smith[1,2,3]*, Laura Temime[3,4†], Lulla Opatowski[1,2†]

[1]Institut Pasteur, Epidemiology and Modelling of Antibiotic Evasion (EMAE), Paris, France; [2]Université Paris-Saclay, UVSQ, Inserm, CESP, Anti-infective evasion and pharmacoepidemiology team, Montigny-Le-Bretonneux, France; [3]Modélisation, épidémiologie et surveillance des risques sanitaires (MESuRS), Conservatoire national des arts et métiers, Paris, France; [4]PACRI unit, Institut Pasteur, Conservatoire national des arts et métiers, Paris, France

**Abstract** The human microbiome can protect against colonization with pathogenic antibiotic-resistant bacteria (ARB), but its impacts on the spread of antibiotic resistance are poorly understood. We propose a mathematical modeling framework for ARB epidemiology formalizing within-host ARB-microbiome competition, and impacts of antibiotic consumption on microbiome function. Applied to the healthcare setting, we demonstrate a trade-off whereby antibiotics simultaneously clear bacterial pathogens and increase host susceptibility to their colonization, and compare this framework with a traditional strain-based approach. At the population level, microbiome interactions drive ARB incidence, but not resistance rates, reflecting distinct epidemiological relevance of different forces of competition. Simulating a range of public health interventions (contact precautions, antibiotic stewardship, microbiome recovery therapy) and pathogens (*Clostridioides difficile*, methicillin-resistant *Staphylococcus aureus*, multidrug-resistant Enterobacteriaceae) highlights how species-specific within-host ecological interactions drive intervention efficacy. We find limited impact of contact precautions for Enterobacteriaceae prevention, and a promising role for microbiome-targeted interventions to limit ARB spread.

*For correspondence:
david.smith@pasteur.fr

†These authors contributed equally to this work

Competing interests: The authors declare that no competing interests exist.

## Introduction

Bacteria are fundamental drivers of human health and disease. On one hand, bacterial pathogens are leading causes of global infectious disease burden, with antibiotic-resistant and healthcare-associated infections posing significant risks to patient safety and public health (*Cassini et al., 2019*; *Cassini et al., 2016*; *Friedman et al., 2016*; *Naylor et al., 2018*). On the other, the bacterial microbiome – the trillions of individual bacteria that collectively inhabit the human body – provides support to development and homeostasis, facilitates core physiological processes like digestion, and protects against diseases ranging from colitis to cancer (*Bäckhed et al., 2005*; *Kamada et al., 2013b*; *Lynch and Pedersen, 2016*; *Round and Mazmanian, 2009*; *Roy and Trinchieri, 2017*). The microbiome can also protect against colonization with infectious bacterial pathogens, a phenomenon known as colonization resistance, limiting their capacities to establish colonies, grow, persist, and transmit (*Bäumler and Sperandio, 2016*; *Buffie and Pamer, 2013*). This is perhaps best exemplified by the canonical *Clostridioides difficile*: growth within the intestine is inhibited by secondary metabolites of commensal gut bacteria, protecting colonized hosts from disease and limiting propagation of the infectious spores that drive between-host transmission (*Kamada et al., 2013a*;

*Pamer, 2016*). The human microbiome is purported to play a defensive role against a range of bacterial pathogens, but the mechanistic nature and epidemiological consequences of within-host microbiome-pathogen interactions are poorly understood.

This relationship between microbiome ecology and bacterial epidemiology is important in the context of widespread antibiotic use and global dissemination of high-risk pathogenic antibiotic-resistant bacteria (ARB). When prescribed appropriately, antibiotics target particular bacterial pathogens, but co-colonizing microbiota are also exposed (*Tedijanto et al., 2018*). This can unintentionally destabilize healthy microbial communities, resulting in dysbiosis, a state of population dynamic disequilibrium (*Bhalodi et al., 2019*; *Coyte et al., 2015*; *Dethlefsen and Relman, 2011*). Microbiome dysbiosis is associated with reduced abundance and diversity of commensal bacteria, impaired host immune responses, and loss of colonization resistance, altogether increasing host susceptibility to ARB colonization (*Kim et al., 2017*; *Sorbara and Pamer, 2019*; *Zhang et al., 2015*). Antibiotic-induced dysbiosis may further result in elevated expression of antibiotic resistance genes, increased rates of horizontal transfer of such genes, and ecological release, whereby subdominant ARB are released from competition via clearance of drug-sensitive bacteria, growing out into dominant colonies (*Doan et al., 2019*; *Letten et al., 2021*; *Ruppé et al., 2019*; *Stecher et al., 2013*). These associations may be particularly relevant for healthcare settings, where antibiotic use is widespread, and where microbiome dysbiosis is increasingly recognized as a key driver of ARB colonization and infection (*Baggs et al., 2018*; *Prescott et al., 2015*; *Ravi et al., 2019*). From a clinical perspective, this motivates a need for public health interventions that minimize or reverse harm to patient microbiota, from antibiotic stewardship, to fecal microbiota transplantation, to microbiome protective therapies. (*de Gunzburg et al., 2018*; *Relman and Lipsitch, 2018*).

In the face of uncertainty, mathematical modeling is a useful tool to analyze epidemiological dynamics of antibiotic resistance and evaluate control measures (*Heesterbeek et al., 2015*; *Opatowski et al., 2011*). However, an incomplete understanding of key eco-evolutionary principles is highlighted as a limitation to using modeling to predict future trends and inform decision-making (*Birkegård et al., 2018*; *Knight et al., 2019*). Disruption of the host microbiome is a long-standing theory explaining how antibiotics select for the spread of resistance at both the individual and population levels (*Lipsitch and Samore, 2002*), but most epidemiological models consider just one species of bacteria at a time, under the traditional assumption that antibiotic selection for resistance results from intraspecific competition between co-circulating strains (*Blanquart, 2019*; *Ramsay et al., 2018*; *Spicknall et al., 2013*). This simple framework has been particularly useful for bacteria like *Streptococcus pneumoniae* and *Staphylococcus aureus*, in which different strains – sometimes conceptualized as drug-sensitive vs. drug-resistant, or community-associated vs. healthcare-associated – are believed to be in close ecological competition (*Blanquart, 2019*; *Domenech de Cellès et al., 2011*; *Kardaś-Słoma et al., 2011*; *Pressley et al., 2010*; *van Kleef et al., 2013*).

Accounting for other forms of complexity in epidemiological models – from treatment intensity, to age-assortative contact behavior, to hospital referral networks, to animal-human interactions, to genetic linkage between resistance and non-resistance genes – has helped to unravel the many, disparate forces that contribute to drive the spread of resistance (*Blanquart et al., 2018*; *Cobey et al., 2017*; *Colijn and Cohen, 2015*; *Donker et al., 2017*; *Lehtinen et al., 2017*; *van Bunnik and Woolhouse, 2017*). Nevertheless, within-host bacterial competition remains a key mechanism of selection for antibiotic resistance dissemination, and an active area of research at the forefront of resistance modeling (*Lipsitch and Samore, 2002*; *Mulberry et al., 2020*; *Spicknall et al., 2013*). For instance, the 'mixed-carriage' model by Davies et al. demonstrates how intraspecific competition results in negative frequency-dependent selection for either of two competing strains, and provides a satisfying mechanistic explanation for widespread strain coexistence at the population level (*Davies et al., 2019*). However, contemporary work has stopped short of evaluating consequences of between-species competition on resistance epidemiology. Yet for many ARB, including emerging high-priority multidrug-resistant bacteria like extended-spectrum beta-lactamase (ESBL) producing Enterobacteriaceae, interactions with the host microbiome may be important mediators of nosocomial colonization dynamics (*Kim et al., 2017*; *Lerminiaux and Cameron, 2019*; *Pilmis et al., 2020*).

Here, we use mathematical modeling to evaluate how microbiome ecology and antibiotic consumption combine to drive the spread and control of antibiotic-resistant bacteria in the healthcare setting. This is presented in two parts. First, we propose a modeling framework for ARB colonization

dynamics, accounting for different within-host ecological interactions – including intraspecific pathogen strain competition, interspecific microbiome-pathogen competition, and horizontal gene transfer (HGT) – in the context of antibiotic treatment. Synthesizing these into a final model, we show how different combinations of ecological interactions drive antibiotic selection for the spread of resistance, with heterogeneous impacts on classic epidemiological indicators. Second, using parameter estimates from the literature, we apply this framework to simulate the nosocomial epidemiology of four high-risk ARB: *C. difficile*, methicillin-resistant *S. aureus* (MRSA), ESBL-producing *Escherichia coli* (ESBL-EC) and carbapenemase-producing *Klebsiella pneumoniae* (CP-KP). Expert elicitation interviews were conducted to characterize the clinical relevance of microbiome dysbiosis for each species, and to qualify and quantify interaction coefficients with uncertainty. By simulating a range of different public health interventions, we demonstrate the theoretical importance of microbiome-pathogen interactions as mediators of ARB epidemiology, and determining factors in the control of resistance dissemination.

## Model and results

### Part 1: A modeling framework for antibiotic resistance epidemiology in healthcare settings

We propose a series of five models describing colonization dynamics of an antibiotic-resistant bacterial *pathogen*, denoted $P^R$, among hospital inpatients in an acute care setting. Each model accounts for different within-host ecological interactions between $P^R$ and other bacteria. Models are described using systems of ordinary differential equations (ODEs), and are evaluated deterministically using numerical integration. Across models, three primary epidemiological outcomes are calculated at steady-state equilibrium: $P^R$ prevalence (the proportion of patients colonized), $P^R$ incidence (the daily rate of colonization acquisition within the hospital), and the pathogen resistance rate (the proportion of patients colonized with the focal antibiotic-resistant strain $P^R$ relative to a competing drug-sensitive strain $P^S$). We also derive and evaluate the basic reproduction number $R_0$ for $P^R$, an indicator of pathogen epidemicity representing the average number of patients expected to acquire a novel pathogen from an initial index patient. See Materials and methods for technical details, and the supplementary appendix for the complete modeling framework and assumptions (Appendix *Equation A1*).

### A simple transmission model for bacterial colonization

We start with a Susceptible-Colonized transmission model (*Figure 1A*) representing a population of N hospital patients as either susceptible to colonization (S) or colonized ($C^R$) by $P^R$, the focal strain or species:

$$\begin{aligned}\frac{dS}{dt} &= N \times (1-f) \times \mu - S \times (\lambda_R + \alpha_R + \mu) + C^R \times (\gamma_R + \sigma_R) \\ \frac{dC^R}{dt} &= N \times f \times \mu + S \times (\lambda_R + \alpha_R) - C^R \times (\gamma_R + \sigma_R + \mu)\end{aligned} \quad (1)$$

This model is adapted from classic colonization models of antibiotic-resistant bacteria (*Austin et al., 1997*), includes no ecological interactions with non-focal bacteria, and reflects a suite of common assumptions relevant to the healthcare setting, including: (i) a symmetric rate of patient admission and discharge μ, holding N constant; (ii) a proportion of patients colonized upon admission *f*, reflecting pathogen prevalence in the community; (iii) a dynamic rate of colonization acquisition $\lambda_R = \beta \times C^R / N$, for host-to-host transmission; (iv) a static rate of acquisition $\alpha_R$, for endogenous routes of acquisition; (v) a rate of natural clearance $\gamma_R$; and (vi) a rate of effective antibiotic treatment $\sigma_R$. Note that all state variables are functions of time *t*, though this is omitted from ODEs for brevity.

Efficacy of antibiotic treatment is assumed to depend both on the distribution of antibiotics consumed in the hospital and on the intrinsic antibiotic resistance profile of $P^R$. We express this as

$$\sigma_R = \mathrm{a} \times (1 - r_R) \times \theta_C \quad (2)$$

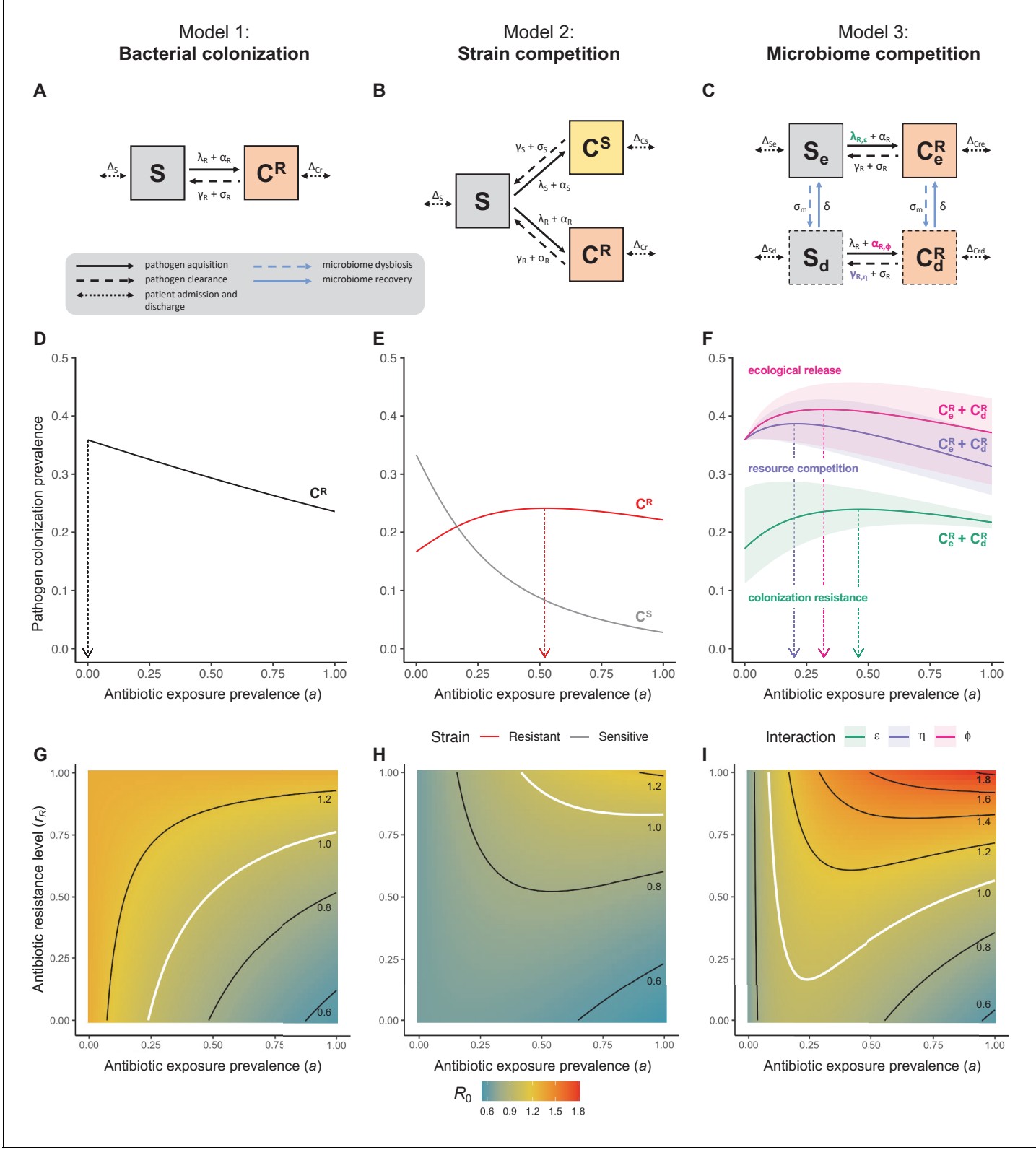

**Figure 1.** Comparison of models describing bacterial colonization dynamics in healthcare settings: in contrast to predictions from a model with no ecological competition (A, D, G), models including strain competition (B, E, H) or microbiome competition (C, F, I) can explain how antibiotics select for the epidemiological spread of antibiotic-resistant bacteria. For all models, ODEs are integrated numerically using the same parameter values representing a generic nosocomial pathogen $P^R$ (see **Appendix 1—table 1**). (A,B,C) Compartmental model diagrams representing corresponding ODE

*Figure 1 continued on next page*

*Figure 1 continued*

systems from the main text (A: *Equation 1*; B: *Equation 3*; C: *Equation 4*). (D,E,F) Pathogen colonization prevalence as a function of antibiotic exposure prevalence ($a$), assuming partial antibiotic resistance ($r_R = 0.8$). For (E) $P^S$ and $P^R$ circulate simultaneously, assuming strain-specific differences in antibiotic resistance ($r_S = 0$, $r_R = 0.8$), natural clearance ($\gamma_S = 0.03$ day$^{-1}$, $\gamma_R = 0.06$ day$^{-1}$) and transmission ($\lambda_S = \beta \times C^S/N$, $\lambda_R = \beta \times C^R/N$). For (F), epidemiological dynamics are evaluated independently for each interaction and superimposed ($\varepsilon$ = colonization resistance; $\eta$ = resource competition; $\phi$ = ecological release); shaded intervals represent outcomes across the range of values considered for each interaction (see *Figure 2*); and antibiotics are assumed to induce dysbiosis after 1 day ($\theta_m = 1$ day$^{-1}$), from which microbiome stability recovers after 7 days ($\delta = 1/7$ day$^{-1}$). Dashed vertical arrows denote the levels of antibiotic use that maximize $P^R$ prevalence (the sum of colonized compartments $C^R$). (G,H,I) Numerical evaluation of the basic reproduction number ($R_0$) of $P^R$ as a function of $a$ and $r_R$. White contour lines indicate $R_0 = 1$, above which a single colonized patient admitted to a naïve hospital population is expected to trigger an outbreak. For I, all three microbiome-pathogen interactions are applied simultaneously using baseline values ($\varepsilon = 0.5$; $\eta = 0.5$; $\phi = 5$).

where $a$ is the hospital population's antibiotic exposure prevalence (the proportion of patients exposed to antibiotics at any $t$), $\theta_C$ is the antibiotic-induced clearance rate (the rate at which effective antibiotics clear pathogen colonization), and $r_R$ is the antibiotic resistance level (the proportion of antibiotics that are ineffective against $P^R$). Modeling the latter as a continuous proportion reflects that bacteria are not necessarily fully drug-sensitive ($r_R = 0$) nor -resistant ($r_R = 1$), but can range in their sensitivity to different antibiotics ($0 \leq r_R \leq 1$). The resistance level $r_R$ is thus a model input interpreted as an overall measure of the pathogen's innate degree of resistance to the particular antibiotics to which it is exposed.

Following models build upon these assumptions, representing the same pathogen $P^R$, but altering its ecological interactions with other bacteria from one model to the next. Models were evaluated over the same generic parameter space, to isolate impacts of model structure on epidemiological outcomes in the context of antibiotic use (see parameters in *Appendix 1—table 1*).

## Antibiotic selection for resistance: the role of strain competition

Antibiotic consumption selects for the epidemiological spread of antibiotic-resistant bacteria (*Chatterjee et al., 2018*). To explain this mechanistically, a classic modeling assumption is that selection results from intraspecific competition between two or more drug-sensitive strains $P^S$ and drug-resistant strains $P^R$ (*Spicknall et al., 2013*). The reasoning goes: strains of the same species occupy the same ecological niche, so colonization with one strain inhibits colonization with another. In turn, antibiotics that preferentially clear $P^S$ render the within-host niche available to potential colonization with co-circulating $P^R$, indirectly favouring $P^R$ spread through the host population. A simple two-strain 'exclusive colonization' model (*Figure 1B*) can be written as:

$$\frac{dS}{dt} = -S \times (\lambda_S + \lambda_R + \alpha_S + \alpha_R) + C^S \times (\gamma_S + \sigma_S) + C^R \times (\gamma_R + \sigma_R) + \Delta_S$$

$$\frac{dC^S}{dt} = S \times (\lambda_S + \alpha_S) - C^S \times (\gamma_S + \sigma_S) + \Delta_{C^S} \tag{3}$$

$$\frac{dC^R}{dt} = S \times (\lambda_R + \alpha_R) - C^R \times (\gamma_R + \sigma_R) + \Delta_{C^R}$$

where patients can be colonized ($C^S$, $C^R$) by either strain ($P^S$, $P^R$). Subscripts $S$ and $R$ denote strain-specific rates, accounting for ecological differences between strains (e.g. antibiotic resistance levels, fitness costs of resistance; see Appendix *Equations 3–7*). Strains are labeled as sensitive or resistant, but here this is interpreted as relative ($r_S < r_R$). For simplicity, patient demography (admission and discharge) is given as $\Delta_j$ for each compartment $j$.

Unlike the single-strain Susceptible-Colonized model (*Figure 1D and G*), the strain competition model can explain how antibiotic consumption selects for the spread of resistance (*Figures 1E,H*). When $P^R$ is resistant to all antibiotics ($r_R = 1$), its prevalence increases monotonically with increasing antibiotic exposure (*Appendix 1—figure 1*). When resistance is partial – when $P^R$ is still cleared by antibiotics, but at a lower rate than $P^S$ ($r_S < r_R < 1$) – selection for resistance can peak at intermediate antibiotic exposure, owing to a trade-off in how antibiotics both clear and facilitate $P^R$. As such, $R_0$ for $P^R$ tends to increase with antibiotic use when $P^R$ is highly resistant to antibiotics (high $r_R$), but decrease when still largely sensitive (low $r_R$).

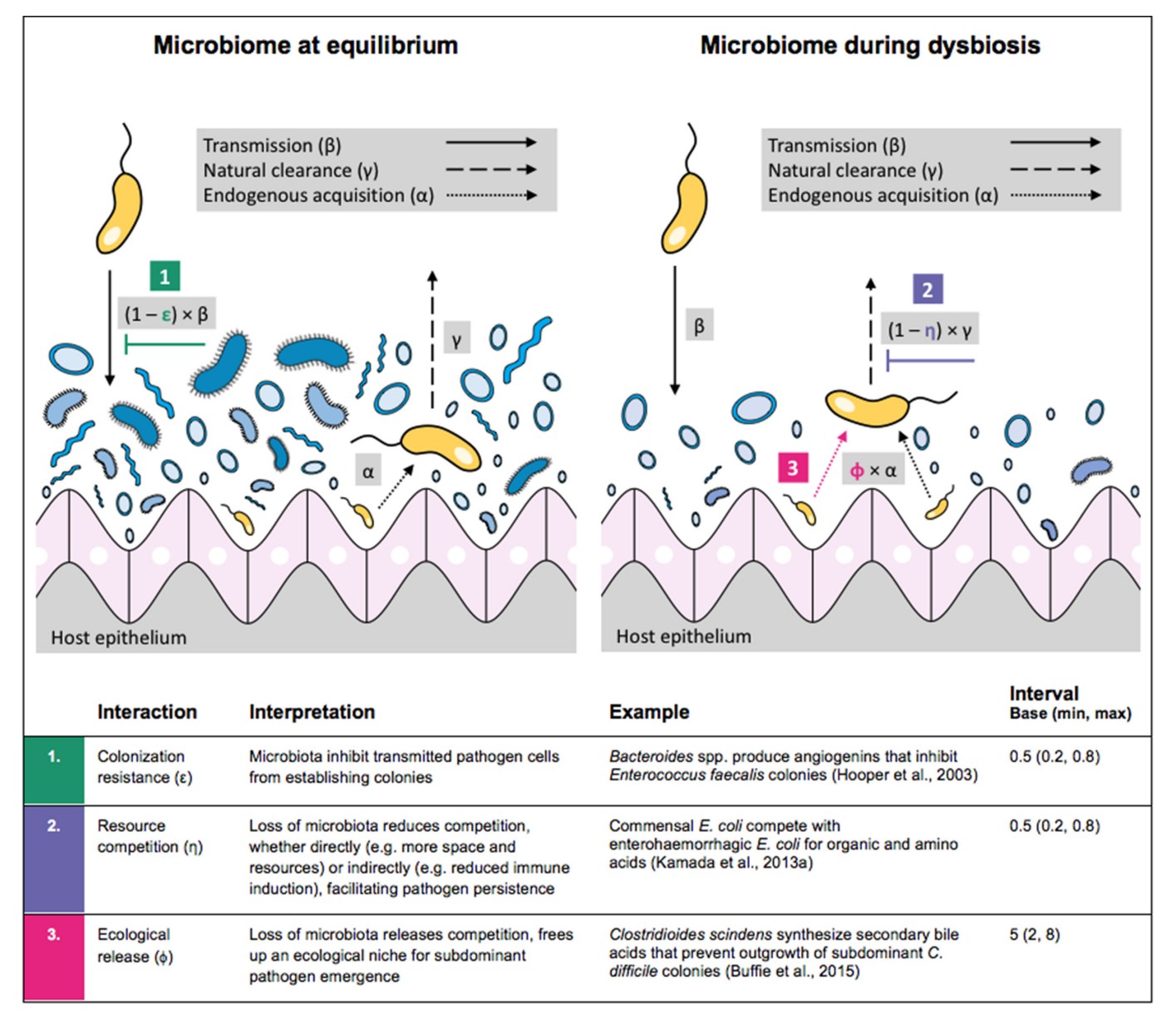

**Figure 2.** Illustration of within-host ecological interactions between the host microbiome (blue) and a transmissible bacterial pathogen P[R] (yellow), and their impact on P[R]'s vital epidemiological parameters β (transmission rate), γ (clearance rate) and α (endogenous acquisition rate). To illustrate the latter: sub-dominant, non-transmissible P[R] colonies inhibited by microbiota are represented by small cartoon pathogens, which can grow into dominant, transmissible colonies (large cartoon pathogens) via endogenous acquisition. Microbiome-pathogen interactions are assumed to differ between hosts with a stable microbiome at population dynamic equilibrium (left) and hosts experiencing antibiotic-induced microbiome dysbiosis (right). Interaction coefficients can be interpreted as terms explaining variation in host susceptibility to pathogen colonization, as depending on their recent history of antibiotic exposure. For interaction coefficient parameter values, broad intervals are assumed for the baseline analysis.

Most contemporary antibiotic resistance models are variants of this typology (*Blanquart, 2019*; *Spicknall et al., 2013*), but intraspecific strain competition is not the only mechanism by which antibiotic consumption can drive ARB spread (*Lipsitch and Samore, 2002*), and may have limited relevance for certain species, settings, timescales and epidemiological indicators.

## Antibiotic selection for resistance: the role of microbiome competition

Antibiotic-induced microbiome dysbiosis is an alternative mechanism by which antibiotics may select for the spread of resistance. We propose a model in which (i) bacterial pathogens and commensal microbiota compete ecologically within the host, and (ii) antibiotic-induced dysbiosis disrupts these interactions, predisposing hosts to pathogen colonization. We consider three competitive within-host microbiome-pathogen interactions and conceptualize how they affect pathogen epidemiology (*Figure 2*; *Buffie et al., 2015*; *Hooper et al., 2003*; *Kamada et al., 2013a*). First, a stable microbiome can act as a barrier preventing introduced pathogens from establishing colonies. We modeled this *colonization resistance* ($\varepsilon$) as a reduced rate of pathogen transmission ($\beta$) to hosts with stable microbiota, $\beta_\varepsilon = (1 - \varepsilon) \times \beta$. Second, co-colonizing bacteria can compete for space, nutrients and other limited resources within the host. In this case, antibiotics that reduce the density of potential competitors may favour pathogen persistence. We modeled this *resource competition* ($\eta$) as a reduced rate of pathogen clearance ($\gamma$) among hosts undergoing dysbiosis, $\gamma_\eta = (1 - \eta) \times \gamma$. Lastly, microbiome dysbiosis can favor the emergence or outgrowth of subdominant pathogen colonies, and this *ecological release* ($\phi$) was modeled as an increased rate of endogenous pathogen acquisition ($\alpha$) upon dysbiosis, $\alpha_\phi = \phi \times \alpha$.

We integrate the microbiome, these three interactions, and antibiotic-induced microbiome dysbiosis into *Equation 1*. The resulting 'microbiome competition model' (*Figure 1C*) is given by

$$
\begin{aligned}
\frac{dS_e}{dt} &= -S_e \times \left(\lambda_{R,\varepsilon} + \alpha_R + \sigma_m\right) + S_d \times \delta + C_e^R \times \left(\gamma_R + \sigma_R\right) + \Delta_{S_e} \\
\frac{dS_d}{dt} &= S_e \times \sigma_m - S_d \times \left(\lambda_R + \alpha_{R,\phi} + \delta\right) + C_d^R \times \left(\gamma_{R,\eta} + \sigma_R\right) + \Delta_{S_d} \\
\frac{dC_e^R}{dt} &= S_e \times \left(\lambda_{R,\varepsilon} + \alpha_R\right) - C_e^R \times \left(\gamma_R + \sigma_m + \sigma_R\right) + C_d^R \times \delta + \Delta_{C_e^R} \\
\frac{dC_d^R}{dt} &= S_d \times \left(\lambda_R + \alpha_{R,\phi}\right) + C_e^R \times \sigma_m - C_d^R \times \left(\gamma_{R,\eta} + \delta + \sigma_R\right) + \Delta_{C_d^R}
\end{aligned}
\tag{4}
$$

describing colonization ($C^R$) with a single pathogen strain ($P^R$) across two host types: patients with a microbiome at dynamic equilibrium (subscript $e$) and those undergoing dysbiosis (subscript $d$). Antibiotics induce dysbiosis at a rate $\sigma_m$ given by

$$
\sigma_m = \mathrm{a} \times \theta_m
\tag{5}
$$

such that microbiome dysbiosis depends on both antibiotic exposure prevalence ($\mathrm{a}$) and the rate at which antibiotic exposure causes dysbiosis ($\theta_m$). Accordingly, the same level of antibiotic exposure ($\mathrm{a}$) can have asymmetric effects on microbiome stability (via $\theta_m$) and $P^R$ colonization (via $(1 - r_R) \times \theta_c$, as in *Equation 2*). Dysbiosis can result in long-term changes to microbiome composition, but ecological function and population dynamic stability tend to recover in the days or weeks following antibiotic therapy (*Lozupone et al., 2012*), represented here by microbiome recovery rate $\delta$.

Included microbiome-pathogen interactions predict in different ways how antibiotic consumption selects for the epidemiological spread of antibiotic-resistant bacterial pathogens (*Figure 1F*). Like strain competition, microbiome competition underlies a trade-off in antibiotic selection: $P^R$ prevalence and $R_0$ increase monotonically with antibiotic exposure when $P^R$ is completely antibiotic-resistant ($r_R = 1$) (*Appendix 1—figure 1*), but can peak at intermediate antibiotic exposure when resistance is partial ($0 < r_R < 1$) because antibiotics simultaneously clear pathogen colonization and induce greater host susceptibility through dysbiosis (*Figure 1I*). This trade-off is more pronounced when multiple interactions are combined: when microbiota simultaneously limit multiple colonization processes (transmission, emergence, persistence), patients with stable microbiota are more protected from $P^R$ colonization, but antibiotic use is predicted to have greater epidemiological costs, selecting more strongly for $P^R$ spread (*Appendix 1—figure 2*).

## Antibiotic selection for resistance: combining strain and microbiome competition

Strain competition and microbiome competition are not mutually exclusive: when combined in a two-strain 'microbiome-strain competition' model, different strains of the same pathogen species compete for hosts whose microbiota and history of antibiotic consumption influence susceptibility to

colonization. We assume that microbiome-pathogen interactions are species- and not strain-specific, that is that ε, η and ϕ apply equally to $P^S$ and $P^R$. This is given by:

$$\frac{dS_e}{dt} = -S_e \times (\lambda_{S,\varepsilon} + \lambda_{R,\varepsilon} + \alpha_S + \alpha_R + \sigma_m) + S_d \times \delta + C_e^S \times (\gamma_S + \sigma_S) + C_e^R \times (\gamma_R + \sigma_R) + \Delta_{S_e}$$

$$\frac{dS_d}{dt} = -S_d \times (\lambda_S + \lambda_R + \alpha_{S,\phi} + \alpha_{R,\phi} + \delta) + S_e \times \sigma_m + C_d^S \times (\gamma_{S,\eta} + \sigma_S) + C_d^R \times (\gamma_{R,\eta} + \sigma_R) + \Delta_{S_d}$$

$$\frac{dC_e^S}{dt} = S_e \times (\lambda_{S,\varepsilon} + \alpha_S) + C_d^S \times \delta - C_e^S \times (\gamma_S + \sigma_S + \sigma_m) + \Delta_{C_e^S}$$

$$\frac{dC_d^S}{dt} = S_d \times (\lambda_S + \alpha_{S,\phi}) + C_e^S \times \sigma_m - C_d^S \times (\delta + \gamma_{S,\eta} + \sigma_S) + \Delta_{C_d^S} \qquad (6)$$

$$\frac{dC_e^R}{dt} = S_e \times (\lambda_{R,\varepsilon} + \alpha_R) + C_d^R \times \delta - C_e^R \times (\gamma_R + \sigma_R + \sigma_m) + \Delta_{C_e^R}$$

$$\frac{dC_d^R}{dt} = S_d \times (\lambda_R + \alpha_{R,\phi}) + C_e^R \times \sigma_m - C_d^R \times (\delta + \gamma_{R,\eta} + \sigma_R) + \Delta_{C_d^R}$$

Introducing a drug-sensitive strain $P^S$ to the microbiome model dampens $R_0$ of the focal strain $P^R$, because fewer patients are susceptible to colonization when the competing strain co-circulates in the population (*Appendix 1—figure 3*). However, antibiotic use makes way for $P^R$ not only through preferential clearance of $P^S$, but remaining $P^R$ also benefit from increased host susceptibility to colonization when antibiotics cause dysbiosis. Accordingly, antibiotics that both disrupt host microbiota and clear drug-sensitive pathogen strains tend to select more strongly for the spread of resistant strains than antibiotics that only target one or the other (*Figure 3*). Overall, antibiotics with stronger impacts on pathogen clearance (higher $\theta_C$) lead to increased resistance rates, while consequences for $P^R$ colonization prevalence are modest and depend on potential interactions with the microbiome. Conversely, antibiotics with stronger impacts on microbiome stability (higher $\theta_m$) lead to higher prevalence, with only modest effects on resistance rates. Further, impacts of antibiotic treatment on resistance rates are greater when $P^R$ resists a greater share of antibiotics (higher $r_R$), while impacts on prevalence are greater when microbiome-pathogen interactions are stronger (higher ε, η, ϕ) (*Appendix 1—figure 4*). Different microbiome-pathogen interactions also underlie distinct dynamic responses to theoretical public health interventions. For the same generic pathogen $P^R$, antibiotic stewardship interventions generally, but do not always prevent colonization, with predictions depending on the ecological interactions in effect (e.g. strain competition, microbiome competition), the impact of the intervention (e.g. reduced microbiome disruption, reduced overall prescribing), and the epidemiological outcome considered (e.g. colonization prevalence, resistance rate) (*Appendix 1—figure 5*).

## Antibiotic selection for resistance: introducing interspecific horizontal gene transfer

Horizontal transfer of resistance-encoding genes may also contribute to hospital resistance dynamics (*Evans et al., 2020*; *Lerminiaux and Cameron, 2019*). We extend our model to include interspecific HGT, conceptualized as two-way within-host transfer of a focal resistance gene, either from a resistant pathogen strain $P^R$ to co-colonizing microbiota, or from resistance-bearing microbiota to a co-colonizing drug-sensitive pathogen strain $P^S$. (See final model and assumptions, Appendix *Equations A1–A11*). Overall, an increasing rate of within-host HGT (χ) is found to drive increasing $P^R$ prevalence ($C^R$), regardless of other microbiome-pathogen interactions (*Figure 4*). Under our modeling assumptions, HGT-driven gains in $C^R$ result from symmetric declines in $C^S$, such that HGT's potential impact depends on the presence of both sufficient resistance donors and sufficient recipients. This results in non-linear feedbacks between HGT and other processes that drive strain prevalence (*Appendix 1—figure 7*). For instance, impacts of HGT on $C^R$ are greatest at intermediate antibiotic exposure (a). This is linked to another antibiotic selection trade-off: higher a affords a selective advantage to $P^R$ relative to $P^S$, increasing the potential impact of HGT on $C^R$; but higher a also reduces the pool of recipient $C^S$, ultimately limiting HGT's ability to drive $C^R$.

## Part 2: Model application to high-risk nosocomial pathogens

We applied this modeling framework to simulate colonization dynamics of four nosocomial pathogens in the hospital setting: *C. difficile*, methicillin-resistant *S. aureus* (MRSA), extended-spectrum

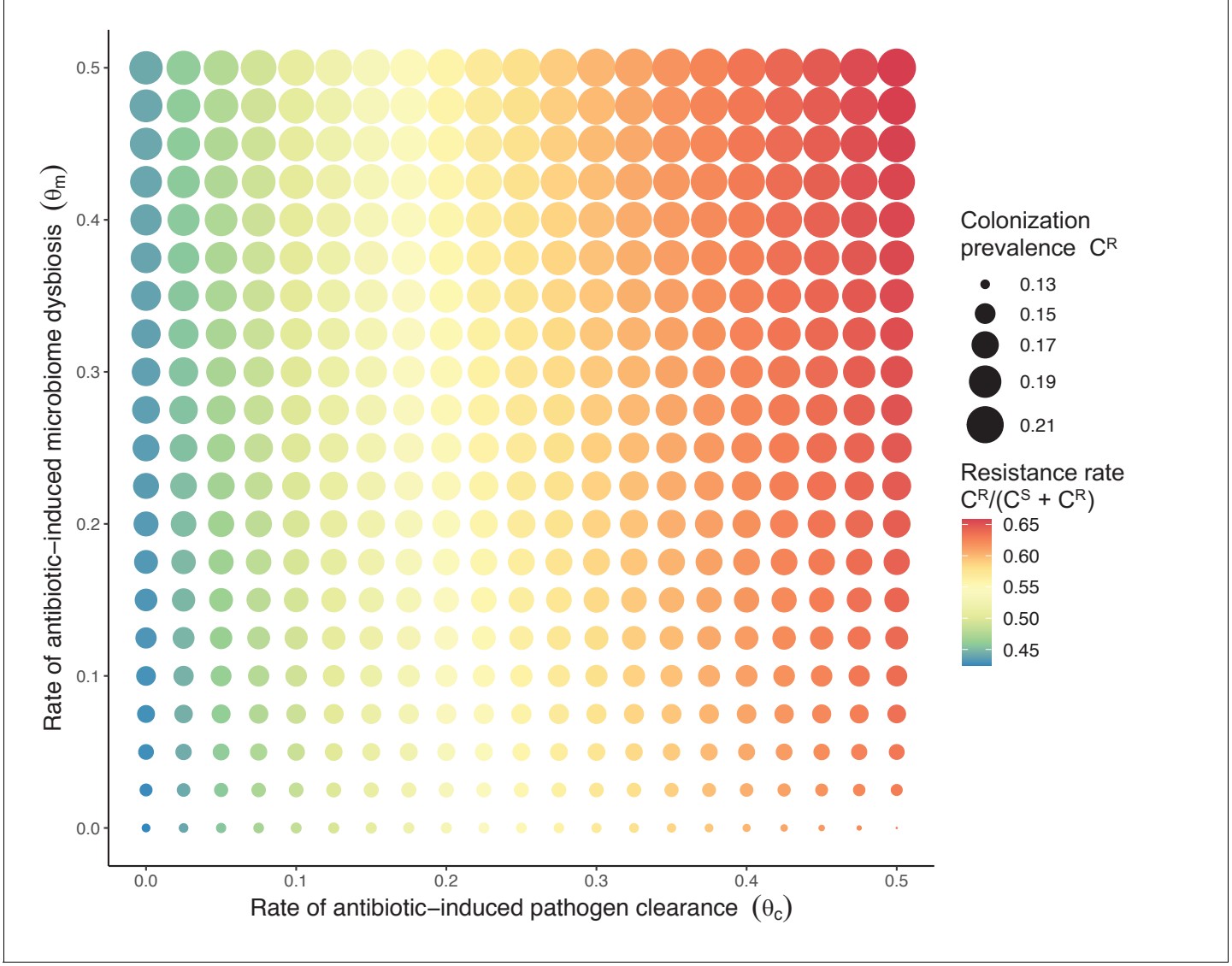

**Figure 3.** Strain competition and microbiome competition as simultaneous forces of antibiotic selection, with asymmetric impacts on epidemiological indicators. In a mixed microbiome-strain competition model (*Equation 6*), colonization prevalence of $P^R$ ($C^R$; circle size) and the pathogen's resistance rate ($C^R/(C^S + C^R)$; color) depend on the relative rates at which antibiotics disrupt microbiota ($\theta_m$) and clear pathogen colonization ($\theta_c$). Antibiotics with a stronger effect on pathogen clearance (higher $\theta_c$) increase the resistance rate, while antibiotics with a stronger effect on microbiota (higher $\theta_m$) increase prevalence.

beta-lactamase-producing *E. coli* (ESBL-EC), and carbapenemase-producing *K. pneumoniae* (CP-KP). Data from the literature were used to parameterize the model to each pathogen (*Appendix 1—figure 5*, *Appendix 1—table 2–6*). Literature estimates for microbiome-pathogen interaction coefficients are scarce, and the species-specific relevance of different within-host interactions (intraspecific strain competition, microbiome competition, HGT) in the hospital environment are not well-defined. To characterize ecological interactions for each species and inform model structure, we conducted interviews with a panel of subject-matter experts in medical microbiology and antibiotic resistance epidemiology (details in Materials and methods). Based on their beliefs, all pathogens were assumed to compete with microbiota; MRSA, ESBL-EC and CP-KP were further assumed to compete intra-specifically with non-focal strains, for simplicity characterized as methicillin-sensitive *S. aureus* (MSSA), *E. coli* (EC), and *K. pneumoniae* (KP); and both ESBL resistance and carbapenem resistance were assumed to be borne by plasmids capable of horizontal transfer between patient microbiota and, respectively, EC and KP (*Figure 5A*). To quantify species-specific strengths of microbiome-

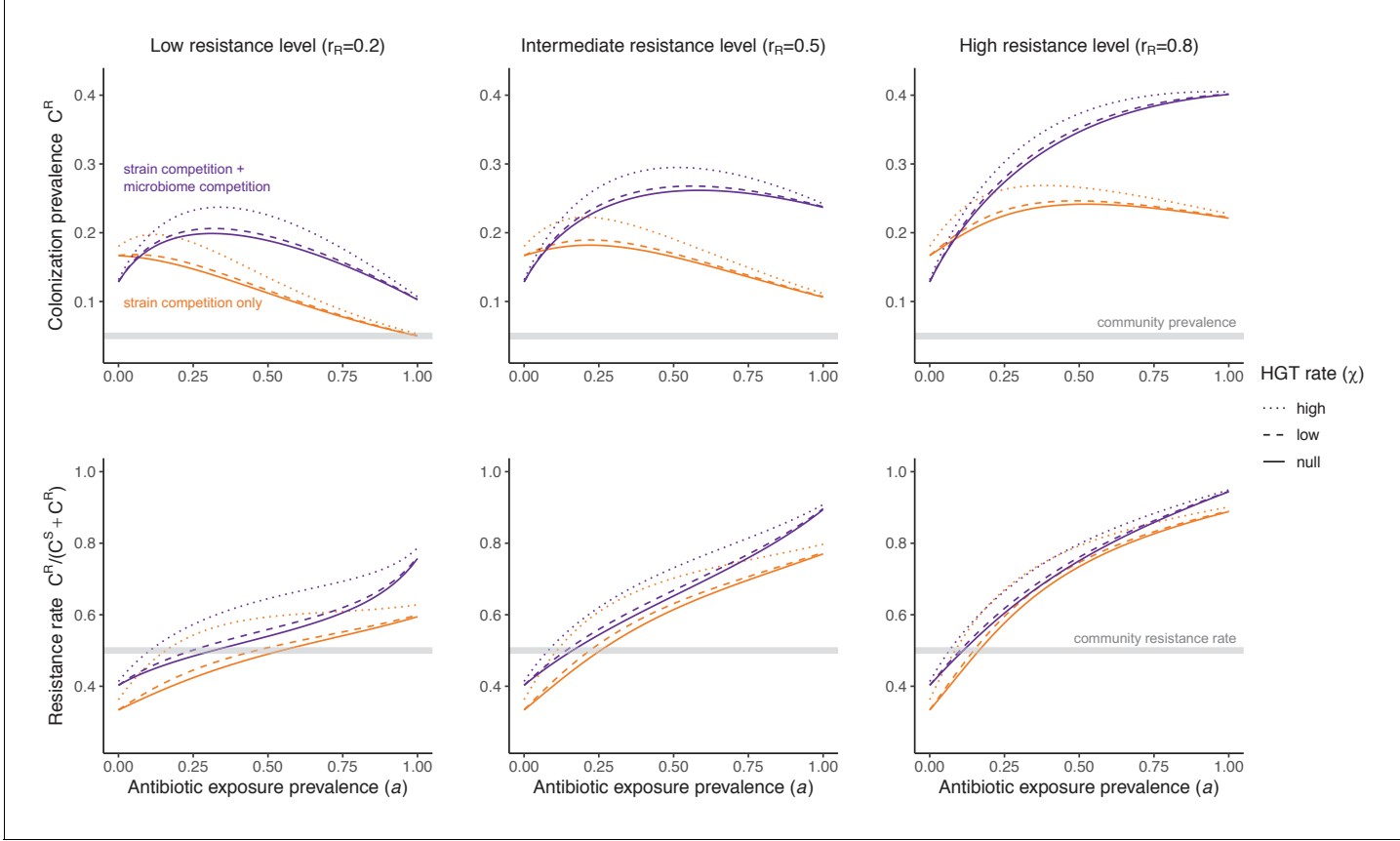

**Figure 4.** Impacts of horizontal gene transfer (HGT) on antibiotic selection for resistance. Allowing a resistance gene to transfer horizontally increases prevalence (top row) of the strain $P^R$ that bears the gene, as well as its resistance rate (bottom row). The relative impact of HGT depends on the gene's rate of transfer ($\chi$, line type), antibiotic exposure prevalence (a, x-axis), competitive interactions between pathogen strains and host microbiota (colors), the level of resistance conferred by the gene ($r_R$, columns), and any other parameters that drive the dynamics of donor and recipient strains. We assume that $\chi_d/\chi_e = 10$, such that the low HGT rate corresponds to {$\chi_e$=0.01 day$^{-1}$, $\chi_d$=0.1 day$^{-1}$} and the high rate to {$\chi_e$=0.1 day$^{-1}$, $\chi_d$=1 day$^{-1}$}. Impacts of HGT on colonization incidence are shown in *Appendix 1—figure 6*. Alternative HGT assumptions are explored in *Appendix 1—figure 7A*.

pathogen interactions, within-host ecological parameters were translated into clinical parameters (*Appendix 1—table 7*), and experts were asked to quantify these using standardized expert elicitation methodology (*Figure 5B*, *Appendix 1—figures 8–10*).

Colonization dynamics for each ARB were modeled against a common backdrop of antibiotic consumption. This was parameterized at the level of antibiotic class, using national data from French hospitals in 2016 (*Appendix 1—table 8*; *Agence nationale de sécurité du médicament et des produits de santé, 2017*). Using data from the literature, antibiotic classes were assumed to vary in their impact on microbiome dysbiosis (very low, low, medium, or high rate of inducing dysbiosis) and on each pathogen strain (classified as sensitive, intermediate, or resistant) (*Figure 5C and D*; *Baggs et al., 2018*; *Brown et al., 2013*; *McCormack and Lalji, 2015*). For each of 10,000 Monte Carlo simulations, in which parameter values were sampled randomly from their respective probability distributions, epidemiological outcomes were evaluated at population dynamic equilibrium. For all outcomes, we compare results from simulations that include microbiome-pathogen interactions and dysbiosis ('microbiome simulations') and those that exclude them ('single-species simulations'). Multivariate sensitivity analyses were also conducted, using partial rank correlation coefficients (PRCCs) to evaluate impacts of parameter uncertainty on model outcomes (details in Methods).

## Species-specific hospital colonization dynamics

Across simulations, *C. difficile* was the most prevalent pathogen (*Figure 6A*), MRSA had the highest resistance rate (*Figure 6B*), and ESBL-EC had the highest rate of incidence within the hospital

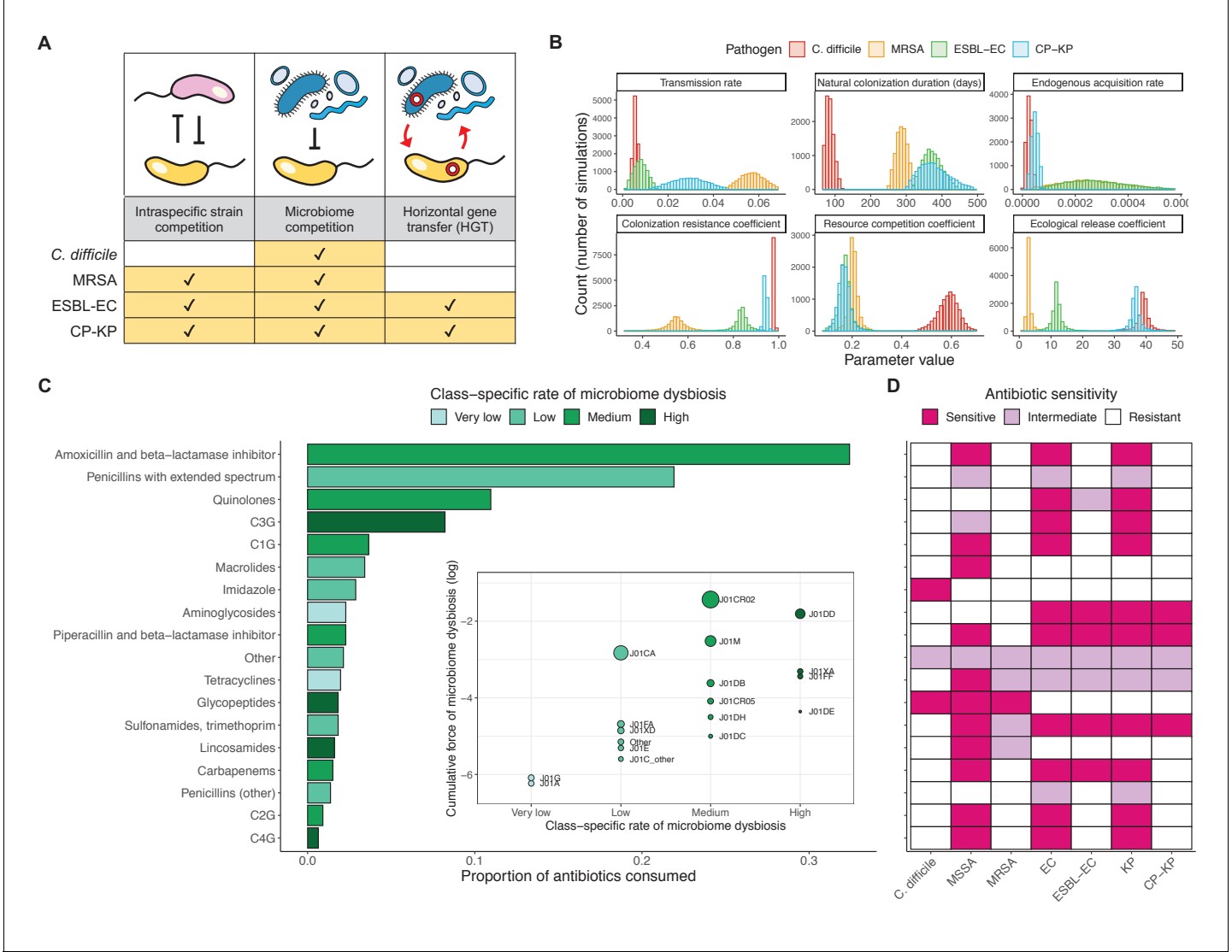

**Figure 5.** Characterizing the species-specific ecology of four selected antibiotic-resistant bacterial pathogens in the hospital setting. (**A**) Model structure: the within-host ecological interactions assumed for each pathogen, based on expert elicitation. (**B**) Simulation inputs: 95% distributions for selected model parameters drawn stochastically over 10,000 runs (all parameter distributions in *Appendix 1—table 2–6*). (**C**) The distribution of antibiotic classes consumed in French hospitals in 2016 (*Agence nationale de sécurité du médicament et des produits de santé, 2017*), shaded by their assumed impact on intestinal microbiome dysbiosis. Inset: the cumulative impact of each antibiotic class (given as ATC codes, see *Appendix 1— table 8* for corresponding names) on dysbiosis ($a_j \times e^{-k}$, see *Equation 7*); circle size represents each class's contribution to exposure prevalence ($a_j$). (**D**) Antibiograms for each pathogen strain and antibiotic class, adapted from the Therapeutics Education Collaboration (*McCormack and Lalji, 2015*).

(*Figure 6C*). CP-KP had the lowest prevalence, resistance rate and incidence, but was the pathogen most favoured by the hospital environment, its prevalence increasing by approximately 5.4-fold (95% uncertainty interval: 2.1–10.9) among hospital patients relative to baseline prevalence in the community (*Appendix 1—figure 12A*). For pathogens subject to intraspecific strain competition, resistance rates in the hospital also tended to exceed rates in the community (*Appendix 1—figure 12B*). Patient-to-patient transmission was the primary route of MRSA acquisition, while endogenous acquisition was the primary route for the enteric pathogens *C. difficile*, ESBL-EC and CP-KP (*Appendix 1—figure 13*). HGT played a potentially important but highly uncertain role for ESBL-EC and CP-KP, accounting for 8.7% (<0.01–49.7%) and 2.1% (<0.01–22.8%) of acquisition events, respectively. In multivariate sensitivity analysis, community prevalence ($f_C$ for *C. difficile*, $f_R$ for others) and rates of endogenous acquisition ($\alpha_R$) had overall the strongest positive impacts (highest PRCCs) on hospital colonization prevalence across ARB, while rates of hospital admission/discharge (μ) and microbiome

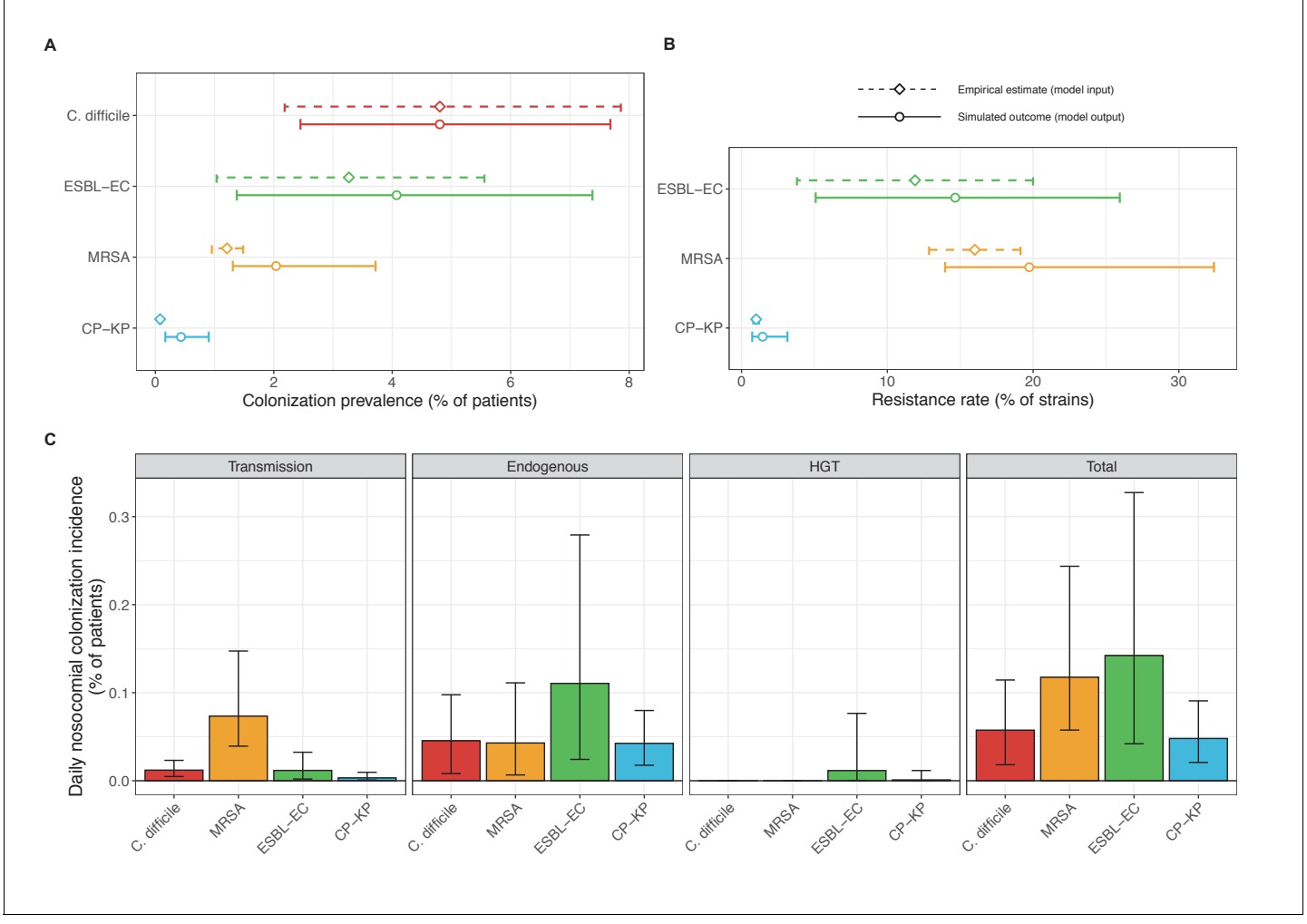

**Figure 6.** Baseline steady-state pathogen colonization outcomes. (**A**) Colonization prevalence, the percentage of patients colonized with the focal strain. Dashed lines (model inputs) represent assumed community prevalence, that is the proportion of patients already colonized upon hospital admission (see *Appendix 1—table 3–6*). Solid lines represent simulated prevalence within the hospital, as resulting from both importation from the community and within-hospital epidemiology. (**B**) Resistance rates, the proportion of *S. aureus* carriers bearing methicillin-resistant strains, *E. coli* carriers bearing ESBL-producing strains, and *K. pneumoniae* carriers bearing carbapenemase-producing strains. As in panel A, dashed lines (model inputs) represent assumed community resistance rates, and solid lines represent simulated resistance rates within the hospital. (**C**) Pathogen incidence (daily rate of within-hospital colonization acquisition), stratified by route of acquisition. Points (in panels A and B) and bar height (panel C) represent medians, and error bars represent 95% uncertainty intervals across 10,000 Monte Carlo simulations. For comparison, the same information for single-species simulations excluding the microbiome is presented in *Appendix 1—figure 11*.

recovery (δ) had the strongest negative impacts (lowest PRCCs; *Appendix 1—figure 14A*). For resistance rates, parameters with the strongest positive impacts (highest PRCCs) were community prevalence ($f_R$), the rate of endogenous acquisition ($\alpha_R$) and the rate of antibiotic-induced pathogen clearance ($\theta_C$). Parameters with the strongest negative impacts (lowest PRCCs) were rates of hospital admission/discharge (μ) and endogenous acquisition of competing drug-sensitive strains ($\alpha_S$) (*Appendix 1—figure 14B*). Across ARB, prevalence estimates, but not resistance rates, were generally sensitive to microbiome parameters.

## Microbiome ecology underlies epidemiological responses to public health interventions

We simulated three types of public health intervention, each across three levels of intervention compliance: (i) contact precautions, which reduced transmission rates of all strains by 20%, 35%, or 50%; (ii) antibiotic stewardship interventions, which reduced or modified hospital antibiotic consumption

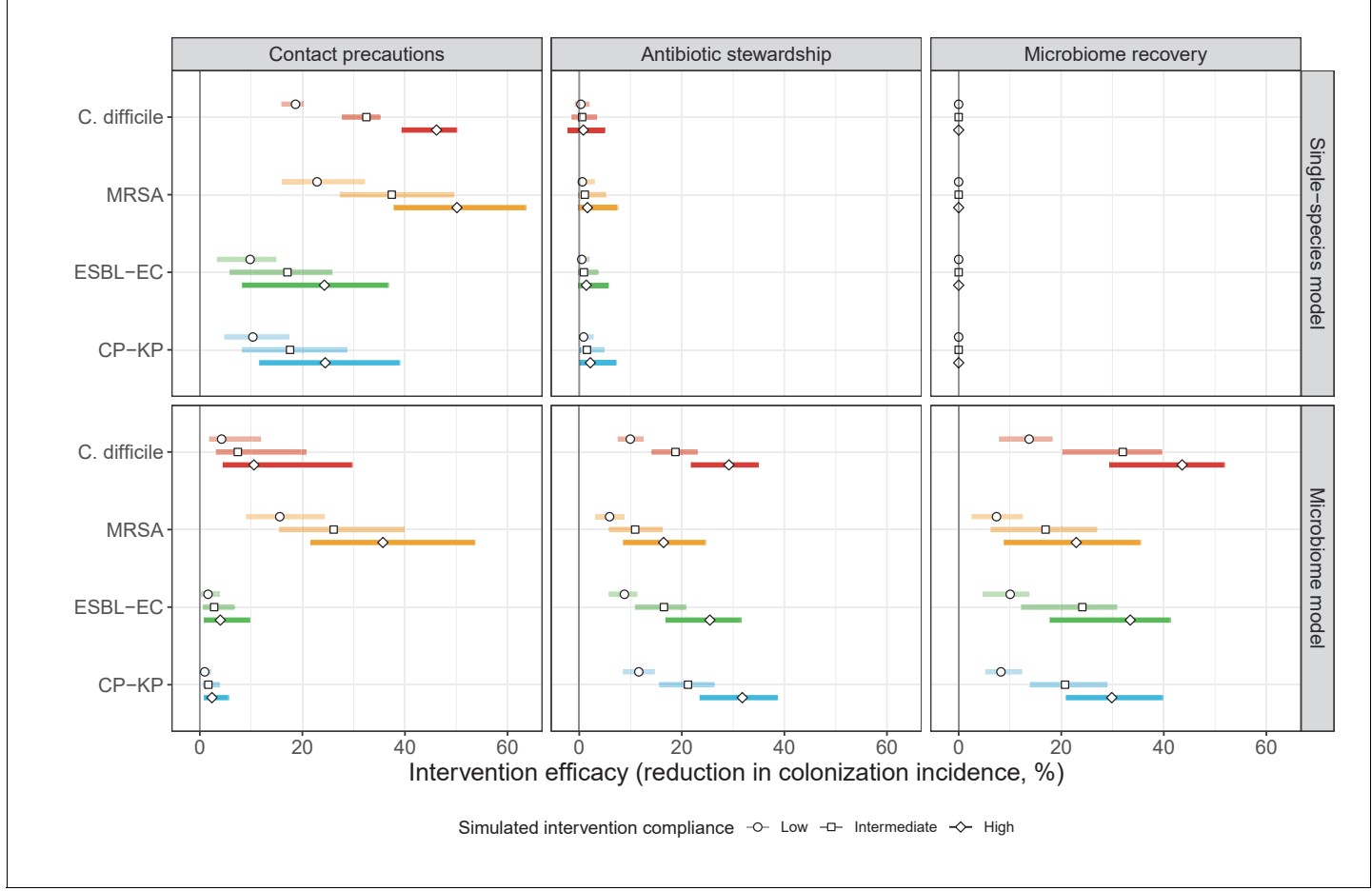

**Figure 7.** The microbiome drives pathogen-specific responses to simulated public health interventions (left panels, contact precautions; middle, antibiotic stewardship; right, microbiome recovery therapy). Top panels show results from simulations using classical 'single-species models' that only account for the focal pathogen species (including intraspecific strain competition for MRSA, ESBL-EC and CP-KP); bottom panels show simulation results when models also include microbiome-pathogen interactions and antibiotic-induced microbiome dysbiosis. For each intervention, three levels of intervention compliance (shading) are simulated. For antibiotic stewardship, simulation results are pooled across three different types of stewardship (see *Appendix 1—figure 17*). Points correspond to medians, and bars to 95% uncertainty intervals across 10,000 Monte Carlo simulations.

patterns by 20%, 35%, or 50%; and (iii) a theoretical 'microbiome recovery therapy' intervention that facilitates recovery from dysbiosis. We assumed a mean 2 day delay to microbiome recovery, and compliance levels corresponding to use among 10%, 30%, or 50% of antibiotic-exposed patients (see Materials and methods). Intervention efficacy was evaluated using: reduction in colonization incidence, 1 – IRR (where IRR is the incidence rate ratio of post-intervention to pre-intervention incidence); and reduction in the resistance rate, 1 – RRR (where RRR is the resistance rate ratio, the ratio of the post-intervention to pre-intervention resistance rate).

Efficacies of simulated interventions varied considerably by pathogen and type of intervention (*Figure 7*). Contact precautions were highly effective for reducing MRSA incidence, of intermediate efficacy for *C. difficile*, and minimally effective for ESBL-EC and CP-KP. Contact precautions had comparatively little impact on resistance rates, with a median 2–4% reduction across simulations and compliance levels for MRSA, 0–2% for CP-KP and negligible impact for ESBL-EC (*Appendix 1—figure 15*). These interventions were overall less effective in microbiome simulations, which tended to limit the role of between-host transmission (via colonization resistance) and favor the role of endogenous acquisition (via ecological release) in the hospital environment when compared to single-species simulations (*Appendix 1—figure 13*).

Antibiotic stewardship interventions led to substantial reductions in nosocomial incidence for all pathogens, but only when the microbiome was taken into account (*Figure 7*). Unlike contact

precautions, which only reduced incidence via transmission, stewardship reduced incidence through all acquisition routes, including HGT (*Appendix 1—figure 16*). Overall efficacy estimates and species-specific responses were similar across three types of stewardship considered (*Appendix 1—figure 17*). Pooling these together under intermediate compliance, colonization incidence was reduced by a median 20% for CP-KP, 18% for C. difficile, 15% for ESBL-EC, and 10% for MRSA. Single-species simulations excluding microbiome competition predicted negligible efficacy of all stewardship interventions for reducing incidence, and non-efficacy for *C. difficile*. Stewardship interventions also had a substantial impact on resistance rates, with overall greater reductions for CP-KP than MRSA and ESBL-EC, and similar outcomes across microbiome and single-species simulations (*Appendix 1—figure 15*).

Lastly, microbiome recovery therapy was potentially highly effective for limiting pathogen incidence, but efficacy varied greatly across different levels of intervention compliance (*Figure 7*). This intervention was most effective against *C. difficile*, of similar efficacy against ESBL-EC and CP-KP, and comparatively least effective against MRSA. Across pathogens, microbiome recovery therapy reduced incidence through all acquisition routes,(*Appendix 1—figure 16*) but had no clear impact on resistance rates (*Appendix 1—figure 15*).

## Discussion

Antibiotics are essential medicines for the treatment and prevention of bacterial infections, but their use selects for the spread of pathogenic antibiotic-resistant bacteria (ARB), and can inadvertently disrupt the host microbiome and its associated immune function (*Buffie and Pamer, 2013*; *Chatterjee et al., 2018*; *Kim et al., 2017*). Within-host ecological interactions between co-colonizing bacteria can have important consequences for their colonization dynamics, which likely extend to influence the epidemiology of human pathogens in clinical settings. Yet microbiome ecology remains largely absent from the epidemiological modeling of antibiotic resistance (*Assab et al., 2017*; *Birkegård et al., 2018*; *Blanquart, 2019*; *Niewiadomska et al., 2019*), suggesting a need to better understand within-host competition between ARB and the host microbiome, its potential epidemiological consequences, and more broadly how antibiotics exert selection pressure on resistant bacteria (*Knight et al., 2019*). We present a modeling framework that includes within-host ecological costs of antibiotic use in the form of microbiome dysbiosis, incorporating a leading hypothesis for antibiotic selection into classical models of resistance epidemiology (*Austin et al., 1997*; *Lipsitch and Samore, 2002*). We formalize three examples of microbiome-pathogen competition potentially affected by dysbiosis, and show how they, either separately or in combination with other forces of selection, help explain how antibiotic use drives the spread of ARB in healthcare settings. We use probabilistic simulations to apply this framework across a panel of characteristic nosocomial pathogens and compare results to a traditional strain-based framework, demonstrating utility of a microbiome-oriented approach for modeling antibiotic resistance epidemiology and associated interventions.

MRSA, *C. difficile* and ESBL-producing Enterobacteriaceae are leading causes of antibiotic-resistant and healthcare-associated infection, while carbapenemase-producing Enterobacteriaceae represent emerging threats of particular concern due to limited therapeutic options for effective treatment of invasive infection (*Cassini et al., 2019*; *Jernigan et al., 2020*; *Miller et al., 2011*; *Rodríguez-Baño et al., 2018*). Antibiotic stewardship is a core component of public health efforts to limit the emergence and spread of these ARB in clinical settings (*Baur et al., 2017*), and an important focus of antibiotic resistance modeling (*Niewiadomska et al., 2019*). Different antibiotics vary in which organisms they intrinsically target, as well as pharmacodynamic factors like route of administration (e.g. oral vs. systemic), clearance mechanism (e.g. biliary vs. renal excretion) and site of absorption (e.g. small vs. large intestine). These differences affect the degree to which bacteria in different host niches are exposed to and cleared by different antibiotics, and are increasingly well described (*Bhalodi et al., 2019*; *Kim et al., 2017*; *Zhang et al., 2015*). Our findings suggest that asymmetric impacts of different antibiotics on competing commensal and pathogenic bacteria can drive antibiotic-driven selection for these high-risk ARB, with important consequences for resistance dynamics and stewardship efficacy.

Findings also suggest promise for interventions that effectively restore microbiome stability and associated colonization resistance as a means to control ARB spread. Fecal microbiota

transplantation is already used to treat recurrent *C. difficile* infection, and is under investigation for multidrug-resistant Enterobacteriaceae decolonization (*Davido et al., 2019a*; *Kassam et al., 2013*; *Saha et al., 2019*). However, its appropriateness for dysbiosis recovery in the absence of other clinical indications is unclear. Transplantation requires rigorous donor screening and close longitudinal follow-up, and cases of donor stool contaminated with toxicogenic and multidrug-resistant bacteria highlight non-negligible risks (*Gupta et al., 2021*; *Zellmer et al., 2021*). Alternative microbiome protective therapies now exist, like DAV132, a novel activated-charcoal product currently undergoing clinical trials. When co-administered with antibiotics by the oral route, DAV132 has been shown to absorb antibiotic residues in the colon and preserve the richness and composition of intestinal microbiota, while maintaining systemic antibiotic exposure (*de Gunzburg et al., 2018*; *de Gunzburg et al., 2015*; *Pinquier et al., 2021*). Modeling has been used previously to evaluate impacts of such microbiome-oriented interventions at the within-host level (*Guittar et al., 2021*; *Guk et al., 2021*), but knock-on impacts on ARB transmission dynamics and epidemiological burden are unknown. Our simulations were limited to a select few ARB, but our framework and findings likely have relevance for other bacteria known to interact with the microbiome, including vancomycin-resistant Enterococci and other multidrug-resistant Enterobacteriaceae (*Davido et al., 2019b*; *Stecher et al., 2013*). and could be further extended to explore impacts of the microbiome on resistance dynamics and intervention efficacy beyond healthcare settings.

Our simulations predicted colonization dynamics broadly consistent with previous findings from the literature. Input from the community was the main driver of hospital prevalence, and both prevalence and resistance rates tended to increase in the hospital relative to the community, as estimated in previous modeling studies (*Knight et al., 2018*; *MacFadden et al., 2019*). Findings reflected an important role for between-host nosocomial transmission for MRSA, as observed clinically (*Khader et al., 2019*; *Nadimpalli et al., 2020*), and a comparatively important role for endogenous acquisition for enteric ARB, as estimated elsewhere (*Bootsma et al., 2007*; *Gurieva et al., 2018*). Our estimates of intervention efficacy were more consistent with previous findings from the literature when microbiome interactions were taken into account. Contact precautions were effective for reducing incidence of MRSA and to a lesser extent *C. difficile*, but had limited impact against Enterobacteriaceae, consistent with clinical trials and modeling estimates. (*Khader et al., 2021*; *Luangasanatip et al., 2015*; *Maechler et al., 2020*). By contrast, simulated antibiotic stewardship interventions were broadly effective for reducing incidence across all included ARB. This is consistent with findings from a meta-analysis of clinical trials evaluating the efficacy of hospital antibiotic stewardship interventions for reducing incidence of ARB colonization and infection (*Baur et al., 2017*), which we updated to exclude studies co-implementing stewardship with alternative interventions (*Appendix 1—figure 18*). In comparison to our findings, estimates from the meta-analysis were associated with greater uncertainty across more heterogeneous interventions, but predicted the same rank order of efficacy across included ARB, and similar mean efficacy for *C. difficile* (19% from n=seven studies, vs. 18% under intermediate compliance in our simulations), ESBL-EC (18% from n=four studies, vs. 15%) and MRSA (12% from n=10 studies, vs. 10%), but higher efficacy for CP-KP (54% from n=one study, vs. 20%).

When excluding microbiome interactions, simulations predicted negligible efficacy of antibiotic stewardship interventions for controlling ARB incidence. Previous models have predicted efficacy using predominantly strain-based approaches, but often focus on resistance rates as the primary outcome, and in many cases assume that patient-to-patient transmission is the only route of colonization acquisition (*Niewiadomska et al., 2019*). Here, microbiome competition was found to have a large impact on incidence but comparatively little impact on resistance rates – both for the theoretical pathogen evaluated in Part 1 (*Figures 3* and *4*) and for the four ARB simulated in Part 2 (*Appendix 1—figures 12* and *14*) – underlying why stewardship interventions were of similar efficacy for reducing resistance rates across single-species and microbiome simulations (*Appendix 1—figure 15*). This reflects the importance of different forces of antibiotic selection for different epidemiological outcomes: while strain-based competition explains relative ecological dynamics of co-circulating strains, microbiome interactions may be better suited to explain how increasing antibiotic use favours ARB incidence, and how antibiotic stewardship and microbiome recovery interventions can help prevent colonization acquisition.

Under strain competition, pathogen colonization is limited by closely related strains sharing an ecological niche, with the same approximate epidemiological profile and transmission

characteristics. Competition against ARB is assumed to depend on the epidemic spread of competing strains, and removal of drug-sensitive strains releases antibiotic selection for resistance. By contrast, microbiome population structure is inherently stable, depending less on the epidemiological transmission of particular taxa, and more on host factors like diet, maternal inheritance, genetics, and antibiotic exposure (*Brito and Alm, 2016*). Despite great inter-individual diversity in microbiome composition, there is functional redundancy from one host to the next, such that colonization resistance and other forms of microbiome competition are shared across healthy hosts colonized with different taxa (*Bashan et al., 2016*; *Kim et al., 2017*). For these reasons, microbiome stability is modeled as a host trait reflecting the functional ecology of the microbiome in different population dynamic states, as opposed to a more traditional bacterial colonization process governed by rates of acquisition and clearance. This is clearly an oversimplification of real microbiome dynamics and complexity (*Hooks and O'Malley, 2017*), but is a useful approximation for the needs of epidemiological modeling, particularly in the absence of data, and reflects the universality of both human microbiome function and of the ecological impacts that antibiotics have on microbiome stability (*Bashan et al., 2016*).

Previous studies have modeled antibiotic exposure as a risk factor for ARB colonization, which can be interpreted as indirectly accounting for microbiome dysbiosis. In transmission models of *C. difficile*, ESBL-EC and *Pseudomonas aeruginosa*, among others, patients undergoing antibiotic therapy are assumed to be at greater risk of colonization and/or infection (*Gingras et al., 2016*; *Hurford et al., 2012*; *MacFadden et al., 2019*). An alternative approach has been to use antibiotic exposure as a coefficient on epidemiological parameters (e.g. transmission, endogenous acquisition), allowing ARB colonization rates to scale with antibiotic use (*Knight et al., 2018*). These strategies reflect widespread recognition that antibiotic use favours ARB acquisition, through erosion of colonization resistance or other supposed mechanisms, and independent of potential competition with other strains. The present work formalizes examples of the microbiome-pathogen interactions that underlie these assumptions, demonstrating their relevance to various epidemiological outcomes, distinguishing them from strain-based selection, and providing a framework for their application.

This study has a number of limitations. First, hospitals and healthcare settings are heterogeneous environments with non-random contact patterns and relatively small population sizes. Stochastic, individual-based models accounting for these factors reproduce more realistic nosocomial transmission dynamics than deterministic ODE simulations, allowing for local extinction events, super-spreaders, and other inherently random epidemiological phenomena. Nonetheless, our goal was to study how ecological mechanisms impact average epidemiological outcomes in the context of different model assumptions and parameter uncertainty, and in this context, ODE modeling was the more appropriate tool, particularly for widely endemic ARB like *C. difficile*, MRSA and ESBL-EC. Still, further insights could certainly be gained by accounting for additional complexity and stochastic heterogeneity in future work, from within-host spatial organization (*Estrela et al., 2015*), to patient and staff contact behavior (*Duval et al., 2019*), to inter-institutional or inter-ward meta-population dynamics (*Shapiro et al., 2020*). These distinctions may be particularly important for rare or non-endemic ARB (e.g. CP-KP in some regions). Second, our evaluation of strain competition was limited to exclusive colonization, a widely used approach (an estimated 12% of published strain competition models allow co-colonization or -infection) (*Niewiadomska et al., 2019*). Yet alternative models predict unique impacts on resistance dynamics (*Spicknall et al., 2013*), and explicit consideration of higher resolution within-host population dynamics has been shown to better reproduce empirical findings in previous work (*Davies et al., 2019*). Further, our exclusive colonization approach precluded assessment of intraspecific HGT, which may have different impacts on resistance dynamics than interspecific HGT.

Third, Monte Carlo simulations were limited by the availability of species-specific model parameters from the literature, in some instances necessitating use of previous modeling results, approximations, or estimates from small studies in specific locations, making the generalizability of results unclear. For instance, Khader et al. estimated a four-fold difference in MRSA transmission rates between hospitals and nursing homes (*Khader et al., 2019*). Such differences could have a substantial impact on dynamics and estimated intervention efficacy, with higher transmission rates favouring use of contact precautions, and higher rates of endogenous acquisition favouring antibiotic stewardship (in the context of a high ecological release coefficient). Uncertainty in endogenous acquisition rates may be particularly important: in multivariate sensitivity analyses, this parameter emerged as a

key driver of both colonization prevalence and resistance rates across ARB (*Appendix 1—figure 14*). Further, data used to estimate class-specific rates of microbiome dysbiosis are specific to the gut; class-specific data for dysbiosis of the skin, the preferred niche of *S. aureus*, were not available. This may over-estimate impacts of dysbiosis on MRSA colonization dynamics.

Finally, the nature of microbiome-pathogen interactions and their epidemiological consequences remain poorly understood and largely unquantified. We show in theory why these interactions matter at the population level, but empirical data were unavailable to inform their parameterization. Instead, we translated microbiome-pathogen competition coefficients into clinical parameters, designed a structured expert elicitation protocol, and conducted interviews allowing subject-matter experts to quantify their beliefs. Although these estimates are subject to substantial bias and uncertainty, they facilitated species-specific characterization of the epidemiological impact of microbiome dysbiosis, and represent useful proxy measures in the absence of clinical data. More broadly, our characterizations of microbiome-pathogen interactions are conceptual, and were mapped mechanistically to particular colonization processes (transmission, clearance, endogenous acquisition), but we note that in other contexts terms like *colonization resistance, resource competition* and *ecological release* may map to specific biochemical processes that could affect epidemiological parameters in different ways.

Despite data limitations, epidemiological conclusions from Monte Carlo simulations were largely consistent with empirical findings (discussed above), suggesting that final parameter distributions were reasonable approximations. Uncertainty in parameter inputs translated to uncertainty in model outputs, reflecting the knowledge gaps underlying our simulations (*Appendix 1—figure 14*). This is exemplified by HGT and its highly uncertain role in driving colonization incidence; to date, HGT modeling has largely been limited to within-host dynamics, and impacts on epidemiological dynamics are only just beginning to come to light (*Evans et al., 2020*; *Leclerc et al., 2019*; *Lerminiaux and Cameron, 2019*). Increasing availability and synthesis of high-quality within-host microbiological data will help to further characterize epidemiological impacts of microbiome-pathogen interactions. Studies are needed that describe ecological impacts of antibiotic exposure on microbiome population structure across control and treatment groups, with longitudinal follow-up evaluating subsequent nosocomial ARB colonization risk. In the absence of clinical data, insights from experiments and within-host models nonetheless suggest that antibiotic disruption of microbiome-pathogen competition is a key driver of selection for resistance (*Baumgartner et al., 2020*; *Estrela and Brown, 2018*; *O'Brien et al., 2021*; *Shaw et al., 2019*; *Stein et al., 2013*; *Tepekule et al., 2019*). The present work highlights the importance of extending these within-host concepts to the population level. Links between within- and between-host dynamics have been widely studied in various contexts, including the theory underlying the evolution of parasite life history and antimicrobial resistance (*Day et al., 2011*; *Greischar et al., 2019*; *zur Wiesch et al., 2011*), but remain largely absent from clinical models of antibiotic resistance (*Blanquart, 2019*). A clear extension of the present work is to explicitly account for simultaneous within- and between-host processes using nested models (*Birkegård et al., 2018*; *Niewiadomska et al., 2019*). However, such models do not necessarily provide more epidemiological clarity, especially when data are lacking and simple heuristic parameters can capture epidemiological consequences of within-host processes (*Gog et al., 2015*; *Mideo et al., 2008*).

In conclusion, we have proposed a mathematical modeling framework for the epidemiology of antibiotic-resistant bacteria that accounts for their potential interactions with the host microbiome. This model simplifies into accessible epidemiological parameters what are in reality highly complex ecological systems, comprising a staggering diversity of microbes and interactions among them. We demonstrate that accounting for at least some of this ecological complexity may help to explain how antibiotics select for the epidemiological spread of resistance, how antibiotic stewardship works to reduce pathogen colonization incidence, and how interventions favouring healthy microbiome function may help to mitigate the epidemiological burden of antibiotic resistance.

## Materials and methods

### Mathematical models of bacterial colonization

We evaluated ODE systems describing colonization dynamics of bacterial pathogens in the health-care setting. The final model, comprising pathogen colonization (*Equation 1*), intraspecific strain competition (*Equation 3*), microbiome-pathogen competition (*Equation 4*), and horizontal gene transfer (HGT), is given alongside all assumptions in the supplementary appendix (Appendix *Equation A1*). ODEs were integrated numerically to calculate steady-state epidemiological outcomes for nosocomial $P^R$ colonization: colonization prevalence (the sum of all compartments $C^R$), colonization incidence (the daily rate of $C^R$ acquisition), and the resistance rate ($C^R/(C^S + C^R)$). (See Appendix 1 for technical details, and R and Mathematica files available online at https://github.com/drmsmith/microbiomeR [*Smith, 2021*; copy archived at swh:1:rev:a3682a24970d79e4-f748952ecc49fcdb16adf48f].) For each model, outcomes were evaluated over the same parameter space representing a generic pathogen $P^R$ (parameters in *Appendix 1—table 1*), while varying specific parameters through univariate and bivariate analysis to assess their impacts on dynamic equilibria in the context of different modeling assumptions. We focused on impacts of the patient population's antibiotic exposure prevalence ($a$), rates of antibiotic-induced pathogen clearance ($\theta_c$) and microbiome dysbiosis ($\theta_m$), the focal pathogen's intrinsic antibiotic resistance level ($r_R$), and mediating impacts of microbiome-pathogen competition ($\varepsilon$, $\eta$, $\phi$) and horizontal gene transfer ($\chi_e$, $\chi_d$).

### Monte Carlo simulations over parameter distributions

We applied the final model to simulate epidemiological dynamics of four high-risk nosocomial pathogens, varying the within-host ecological interactions in effect for each (*Figure 5A*). For MRSA, ESBL-EC and CP-KP, the focal pathogen was taken as the 'drug-resistant' strain $P^R$, while the 'drug-sensitive' strain $P^S$ was taken to represent all other co-circulating strains of the same species. We simulated colonization dynamics using these model characterizations and accounted for parameter uncertainty using Monte Carlo methods. Specifically, 10,000 unique parameter vectors $\Omega$ were created by drawing random values for each parameter from its respective probability distribution (parameters in *Appendix 1—table 2–6*). For each $\Omega$, epidemiological outcomes (prevalence, incidence, resistance rate) were calculated as above. Final outcome distributions were reported as the median and 95% uncertainty interval, that is the 50th (2.5th–97.5th) percentiles across all simulations.

### Parameterizing models for application to specific nosocomial pathogens

Models were parameterized using estimates from the literature, prioritizing clinical studies from the French hospital setting where available. We used expert elicitation to inform model structure and quantify parameter values for microbiome-pathogen interactions. This involved development of a protocol and questionnaire (provided separately), designed using established expert elicitation methodologies for quantitative estimation of unknown parameter values (*Johnson et al., 2010*). To facilitate more reliable parameter interpretation, microbiome-pathogen coefficients were translated into clinical parameters, for example relative risks of pathogen colonization among hospital patients undergoing dysbiosis relative to those with stable microbiota (*Appendix 1—table 7*). Expert estimates were quantified with uncertainty using the MATCH Uncertainty Elicitation Tool (*Morris et al., 2014*), and final parameter distributions were generated by pooling their individual distributions while maintaining estimated species rank order (*Appendix 1—figure 8–10*).

### Characterizing ecological impacts of antibiotic consumption

Antibiotic exposure prevalence was quantified at the level of antibiotic class using nationally representative antibiotic consumption data from French hospitals in 2016 (*Agence nationale de sécurité du médicament et des produits de santé, 2017*). A study from American hospitals in 2006–2012 was used to further stratify antibiotics with ATC code J01F into J01FA and J01FF, J01X into J01XA and J01XD, and J01DD+DE into J01DD and J01DE (*Baggs et al., 2016*). Final antibiotic consumption data are presented in *Appendix 1—table 8*. To simulate class-specific rates of microbiome

dysbiosis, we used a four-point log-linear scale of intestinal microbiome disruption from *Brown et al., 2013*. For interpretation, we present classes as inducing dysbiosis at a high, medium, low, or very low rate. This scale was supplemented with data for additional antibiotic classes from *Baggs et al., 2018*. Applied to our model, the rate that antibiotic treatment induces dysbiosis ($\sigma_m$) is given by

$$\sigma_m = a \times \theta_m \times \sum_{k=0}^{3} \left( a_k \times e^{-k} \right) \tag{7}$$

where $a_k$ is the proportion of antibiotics consumed of each group $k$, and where the most ecologically disruptive group (k=0) causes dysbiosis a mean 12 hr after antibiotic exposure ($\theta_m$ = 2 day$^{-1}$), with classes in less disruptive groups (k=1,2,3) causing dysbiosis at successively slower rates (*Figure 5C*).

To characterize class-specific effects on pathogen clearance, characteristic antibiograms for all strains were adapted from an online compendium from the Therapeutics Education Collaboration (*Figure 5D*; *McCormack and Lalji, 2015*). Each strain $i$ was classified as sensitive ($r_{i,j}$ = 0), of intermediate sensitivity ($0 < r_{i,j} < 1$), or resistant ($r_{i,j}$ = 1) to each antibiotic class $j$. Overall, the rate that antibiotic treatment clears each strain is given by:

$$\sigma_i = a \times \theta_C \times \sum_{j=1}^{18} \left( a_j \times r_{i,j} \right) \tag{8}$$

across the included classes. Under these assumptions, MRSA was resistant to the greatest proportion of antibiotics consumed in hospital (median $r_R$ = 94.5%), followed by *C. difficile* (94.3%), CP-KP (91.7%) and ESBL-EC (84.8%); competing 'drug-sensitive' strains bore considerably less resistance median $r_S$ = 33.0% for MSSA and 23.0% for both *E. coli* and *K. pneumoniae*.

## Sensitivity analyses

Two distinct sensitivity analyses were conducted. First, to evaluate the impact of microbiome competition on model outcomes, a second lot of 'single-species simulations' was run after removing microbiome-pathogen interactions from all $\Omega$ ($\varepsilon$ = 0, $\eta$ = 0, $\phi$ = 1, $\chi$ = 0). Second, the impact of parameter uncertainty on model outcomes was evaluated. For each pathogen, model parameter values were re-sampled from their distributions (*Appendix 1—table 2–6*) using Latin Hypercube Sampling over 10,000 iterations, epidemiological outcomes were re-calculated at population dynamic equilibrium, and partial rank correlation coefficients were calculated between each parameter and the pathogen's (i) colonization prevalence and (ii) resistance rate (using the R package *pse*) (*Chalom and Prado, 2013*; *Marino et al., 2008*).

## Public health interventions

Three public health interventions were incorporated into the final model (see Appendix *Equations A12–A15*). First, contact precautions were assumed to represent physical or behavioral barriers that block opportunities for transmission, reducing transmission rates by the same fraction $\tau_{ipc}$ across all pathogens relative to baseline. Second, antibiotic stewardship programmes were assumed to alter antibiotic consumption patterns in the hospital. Two main types were considered: antibiotic reduction, which limits overall antibiotic prescribing by a fraction $\tau_{asp}$, and antibiotic restriction, which adjusts the distribution of antibiotic classes consumed in the hospital by the same fraction. We considered two types of restriction, the first favouring classically narrow-spectrum antibiotics (e.g. macrolides) over broad-spectrum (e.g. quinolones), and the second favouring antibiotics that cause dysbiosis at very low or low rates (k={3,2}) over those causing dysbiosis at medium or high rates (k={1,0}) (see *Appendix 1—table 8* for antibiotic classification). Third, microbiome recovery therapy was assumed to trigger recovery at rate 0.5 day$^{-1}$ and was apportioned to the fraction $\tau_{pbt}$ of patients. Overall, different values assumed for intervention parameters ($\tau_{ipc}$, $\tau_{asp}$, $\tau_{pbt}$) are interpreted as different levels of compliance to the respective interventions. Intervention efficacy was evaluated using the IRR and RRR (defined in Model and Results). Outcomes were matched across Monte Carlo simulations, such that IRRs and RRRs were calculated for each intervention and compliance level for each $\Omega$. The distribution of outcomes is expressed as the median and 95% uncertainty interval.

## Acknowledgements

We express our gratitude to the subject-matter experts who participated in our expert elicitation exercise: Antoine Andremont, Christian Brun-Buisson, Aurélien Dinh, Stephan Harbarth, Jean-Louis Herrmann, Solen Kernéis, Alban Le Monnier, Benoît Pilmis, Etienne Ruppé and Paul-Louis Woerther. The work was supported directly by internal resources from the French National Institute for Health and Medical Research, the Institut Pasteur, the Conservatoire National des Arts et Métiers, and the University of Versailles–Saint-Quentin-en-Yvelines / University of Paris-Saclay. This study received funding from the French Government's 'Investissement d'Avenir' program, Laboratoire d'Excellence 'Integrative Biology of Emerging Infectious Diseases' (Grant ANR-10-LABX-62-IBEID). DS is supported by a Canadian Institutes of Health Research Doctoral Foreign Study Award (Funding Reference Number 164263) and all authors are supported by the French government through its National Research Agency project SPHINX-17-CE36-0008-01.

## Additional information

### Funding

| Funder | Grant reference number | Author |
| --- | --- | --- |
| Agence Nationale de la Recherche | SPHINX-17-CE36-0008-01 | David RM Smith<br>Laura Temime<br>Lulla Opatowski |
| Canadian Institutes of Health Research | 164263 | David RM Smith |
| Agence Nationale de la Recherche | ANR-10-LABX62-IBEID | Lulla Opatowski |

The funders had no role in study design, data collection and interpretation, or the decision to submit the work for publication.

### Author contributions

David RM Smith, Conceptualization, Data curation, Formal analysis, Investigation, Visualization, Methodology, Writing - original draft, Writing - review and editing; Laura Temime, Lulla Opatowski, Conceptualization, Supervision, Funding acquisition, Writing - review and editing

### Author ORCIDs

David RM Smith https://orcid.org/0000-0002-7330-4262
Laura Temime https://orcid.org/0000-0002-8850-5403

### Decision letter and Author response

Decision letter https://doi.org/10.7554/eLife.68764.sa1
Author response https://doi.org/10.7554/eLife.68764.sa2

## Additional files

### Supplementary files

- Supplementary file 1. Expert elicitation protocol.
- Transparent reporting form

### Data availability

Model equations and parameter values are provided in the manuscript, as well as in supporting R files and a Mathematica notebook available online at https://github.com/drmsmith/microbiomeR (copy archived at https://archive.softwareheritage.org/swh:1:rev:a3682a24970d79e4f748952ecc49fcdb16adf48f).

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

## Appendix 1

## Model and assumptions

### ODEs

The final ODE system, which incorporates horizontal gene transfer (HGT) of a transmissible resistance gene R into the mixed microbiome-strain competition model (*Equation 6* in the main text), is given by:

$$
\begin{aligned}
\frac{dS_{e,s}}{dt} &= -S_{e,s} \times \left( \lambda_{S,\varepsilon} + \lambda_{R,\varepsilon} + \alpha_S + \alpha_R + \sigma_m \right) + S_{d,s} \times \delta + C_{e,s}^S \times \left( \gamma_S + \sigma_S \right) \\
&\quad + C_{e,s}^R \times \left( \gamma_R + \sigma_R \right) + \Delta_{S_{e,s}} \\
\frac{dS_{e,r}}{dt} &= -S_{e,r} \times \left( \lambda_{S,\varepsilon} + \lambda_{R,\varepsilon} + \alpha_S + \alpha_R + \sigma_m \right) + S_{d,r} \times \delta + C_{e,r}^S \times \left( \gamma_S + \sigma_S \right) \\
&\quad + C_{e,r}^R \times \left( \gamma_R + \sigma_R \right) + \Delta_{S_{e,r}} \\
\frac{dS_{d,s}}{dt} &= S_{e,s} \times \left( (1-\omega) \times \sigma_m \right) - S_{d,s} \times \left( \lambda_S + \lambda_R + \alpha_{S,\phi} + \alpha_{R,\phi} + \delta \right) + C_{d,s}^S \times \left( \gamma_{S,\eta} + \sigma_S \right) + \\
&\quad C_{d,s}^R \times \left( \gamma_{R,\eta} + \sigma_R \right) + \Delta_{S_{d,s}} \\
\frac{dS_{d,r}}{dt} &= S_{e,s} \times \left( \omega \times \sigma_m \right) + S_{e,r} \times \sigma_m - S_{d,r} \times \left( \lambda_S + \lambda_R + \alpha_{S,\phi} + \alpha_{R,\phi} + \delta \right) + C_{d,r}^S \times \left( \gamma_{S,\eta} + \sigma_S \right) \\
&\quad + C_{d,r}^R \times \left( \gamma_{R,\eta} + \sigma_R \right) + \Delta_{S_{d,r}} \\
\frac{dC_{e,s}^S}{dt} &= S_{e,s} \times \left( \lambda_{S,\varepsilon} + \alpha_S \right) - C_{e,s}^S \times \left( \gamma_S + \sigma_S + \sigma_m \right) + C_{d,s}^S \times \delta + \Delta_{C_{e,s}^S} \\
\frac{dC_{e,r}^S}{dt} &= S_{e,r} \times \left( \lambda_{S,\varepsilon} + \alpha_S \right) - C_{e,r}^S \times \left( \gamma_S + \sigma_S + \sigma_m + \chi_e \right) + C_{d,r}^S \times \delta + \Delta_{C_{e,r}^S} \\
\frac{dC_{d,s}^S}{dt} &= S_{d,s} \times \left( \lambda_S + \alpha_{S,\phi} \right) + C_{e,s}^S \times \left( (1-\omega) \times \sigma_m \right) - C_{d,s}^S \times \left( \gamma_{S,\eta} + \sigma_S + \delta \right) + \Delta_{C_{d,s}^S} \\
\frac{dC_{d,r}^S}{dt} &= S_{d,r} \times \left( \lambda_S + \alpha_{S,\phi} \right) + C_{e,s}^S \times \left( \omega \times \sigma_m \right) + C_{e,r}^S \times \sigma_m - C_{d,r}^S \times \left( \gamma_{S,\eta} + \sigma_S + \delta + \chi_d \right) + \Delta_{C_{d,r}^S} \\
\frac{dC_{e,s}^R}{dt} &= S_{e,s} \times \left( \lambda_{R,\varepsilon} + \alpha_R \right) - C_{e,s}^R \times \left( \gamma_R + \sigma_R + \sigma_m + \chi_e \right) + C_{d,s}^R \times \delta + \Delta_{C_{e,s}^R} \\
\frac{dC_{e,r}^R}{dt} &= S_{e,r} \times \left( \lambda_{R,\varepsilon} + \alpha_R \right) + C_{e,r}^S \times \chi_e + C_{e,s}^R \times \chi_e - C_{e,r}^R \times \left( \gamma_R + \sigma_R + \sigma_m \right) + C_{d,r}^R \times \delta + \Delta_{C_{e,r}^R} \\
\frac{dC_{d,s}^R}{dt} &= S_{d,s} \times \left( \lambda_R + \alpha_{R,\phi} \right) + C_{e,s}^R \times \left( (1-\omega) \times \sigma_m \right) - C_{d,s}^R \times \left( \gamma_{R,\eta} + \sigma_R + \delta + \chi_d \right) + \Delta_{C_{d,s}^R} \\
\frac{dC_{d,r}^R}{dt} &= S_{d,r} \times \left( \lambda_R + \alpha_{R,\phi} \right) + C_{d,r}^S \times \chi_d + C_{e,s}^R \times \left( \omega \times \sigma_m \right) + C_{e,r}^R \times \sigma_m + C_{d,s}^R \times \chi_d - \\
&\quad C_{d,r}^R \times \left( \gamma_{R,\eta} + \sigma_R + \delta \right) + \Delta_{C_{d,r}^R}
\end{aligned}
\tag{A1}
$$

This system of equations describes epidemiological colonization dynamics of a bacterial pathogen among host classes $H_{j,k}$, corresponding to patients of pathogen colonization status $H$ ($S$, $C^S$, $C^R$), microbiome status $j$ (e, equilibrium; or d, dysbiosis) and microbiome resistance profile $k$ (s, not bearing R; r, bearing R). Equations can be found in an R file and Mathematica notebook available online (https://github.com/drmsmith/microbiomeR), and modeling assumptions are listed below.

### Patient demography

We assume constant population size (N=1), balanced by a daily rate of hospital admission and discharge μ. The number of patients in each compartment is expressed as proportions of the total population. The proportion of patients entering each compartment $j$ is given by admission fractions $f_j$. These are interpreted as representing prevalence of different host types in the community, which are assumed to be stable over time and unaffected by hospital dynamics. For simplicity, demography is expressed as $\Delta_j$ for each class $j$, which expand to:

$$\begin{aligned}
\Delta_{S_{e,s}} &= \mu \times \left( (1-f_C) \times (1-f_d) \times (1-f_\omega) - S_{e,s} \right) \\
\Delta_{S_{e,r}} &= \mu \times \left( (1-f_C) \times (1-f_d) \times f_\omega - S_{e,r} \right) \\
\Delta_{S_{d,s}} &= \mu \times \left( (1-f_C) \times f_d \times (1-f_\omega) - S_{d,s} \right) \\
\Delta_{S_{d,r}} &= \mu \times \left( (1-f_C) \times f_d \times f_\omega - S_{d,r} \right) \\
\Delta_{C_{e,s}^S} &= \mu \times \left( f_C \times (1-f_R) \times (1-f_d) \times (1-f_\omega) - C_{e,s}^S \right) \\
\Delta_{C_{e,r}^S} &= \mu \times \left( f_C \times (1-f_R) \times (1-f_d) \times f_\omega - C_{e,r}^S \right) \\
\Delta_{C_{d,s}^S} &= \mu \times \left( f_C \times (1-f_R) \times f_d \times (1-f_\omega) - C_{d,s}^S \right) \\
\Delta_{C_{d,r}^S} &= \mu \times \left( f_C \times (1-f_R) \times f_d \times f_\omega - C_{d,r}^S \right) \\
\Delta_{C_{e,s}^R} &= \mu \times \left( f_C \times f_R \times (1-f_d) \times (1-f_\omega) - C_{e,s}^R \right) \\
\Delta_{C_{e,r}^R} &= \mu \times \left( f_C \times f_R \times (1-f_d) \times f_\omega - C_{e,r}^R \right) \\
\Delta_{C_{d,s}^R} &= \mu \times \left( f_C \times f_R \times f_d \times (1-f_\omega) - C_{d,s}^R \right) \\
\Delta_{C_{d,r}^R} &= \mu \times \left( f_C \times f_R \times f_d \times f_\omega - C_{e,s}^R \right)
\end{aligned} \tag{A2}$$

where $f_C$ is the proportion of patients colonized with the pathogen upon admission, $f_R$ is the proportion of colonized patients bearing the focal resistant strain (i.e. the community resistance rate), $f_d$ is the proportion of patients undergoing dysbiosis upon hospital admission, and $f_\omega$ is the proportion of admitted patients with microbiota bearing R.

## Pathogen epidemiology

Pathogen colonization can be acquired through dynamic patient-to-patient *transmission* via the dynamic force of infection λ, which is defined as the product of the pathogen's intrinsic transmission rate (β) and the prevalence of patients colonized at a particular time *t*. For each strain P$^j$, with *j* in {S, R}, this is given by

$$\lambda_j = \frac{\beta \times \left( C_{e,s}^j + C_{e,r}^j + C_{d,s}^j + C_{d,r}^j \right)}{N} \tag{A3}$$

In hospital environments, pathogen colonization incidence also results from *endogenous acquisition* (α). This subsumes alternative routes of acquisition not resulting from direct patient-to-patient transmission, and can include processes like translocation and within-host outgrowth of a subdominant/non-detectable/non-transmissible colony (into a dominant/detectable/transmissible colony) (*Archambaud et al., 2019*; *Bootsma et al., 2007*; *Duval et al., 2019*; *Gurieva et al., 2018*). This reflects the tenet that 'everything is everywhere, but the environment selects' (*de Wit and Bouvier, 2006*). We assume that pathogen colonization is not acquired endogenously among patients undergoing antibiotic treatment capable of clearing that pathogen, such that for each strain P$^j$,

$$\alpha_j = \alpha' \times \left( 1 - a \times (1 - r_j) \right) \tag{A4}$$

where α′ is the baseline rate in untreated patients. For the hospital simulation study (below), available literature estimates for endogenous acquisition rates were not strain-specific. These were generated by scaling species-specific rates by the baseline prevalence of each strain in the community, or

$$\begin{aligned}
\alpha_S &= \alpha' \times \left( 1 - a \times (1 - r_S) \right) \times (1 - f_R) \\
\alpha_R &= \alpha' \times \left( 1 - a \times (1 - r_R) \right) \times f_R
\end{aligned} \tag{A5}$$

Pathogen colonization is naturally cleared by the host at rate γ. To reflect potential metabolic costs associated with bearing antibiotic resistance genes (*Melnyk et al., 2015*), we assumed that C$^R$

(colonization with the resistant strain) is naturally cleared at a faster rate than $C^S$ (colonization with the sensitive strain), such that $\gamma_S < \gamma_R$. This is given by

$$\gamma_R = \gamma_S \times (1 + c) \tag{A6}$$

where $c$ is interpreted as the fitness cost of bearing R.

## Antibiotic resistance

Pathogen colonization is cleared by antibiotics according to strain-specific rates of effective antibiotic treatment, given by

$$\begin{aligned} \sigma_S &= a \times (1 - r_S) \times \theta_C \\ \sigma_R &= a \times (1 - r_R) \times \theta_C \end{aligned} \tag{A7}$$

Antibiotic exposure prevalence $a$ is applied independently of colonization status to reflect high estimated rates of bystander selection for ARB, which predominantly colonize patients asymptomatically, are rarely detected and only opportunistically cause disease (***Tedijanto et al., 2018***). Among patients exposed to effective antibiotic therapy, pathogen colonization is cleared at a constant rate $\theta_C$. The proportion of antibiotics that are ineffective is given by the antibiotic resistance level $r_j$, which is a proportion reflecting the pathogen's innate resistance to the antibiotics to which it is exposed, ranging from complete antibiotic sensitivity ($r_j = 0$) to complete resistance ($r_j = 1$), For model application, effective antibiotic treatment is considered at the level of antibiotic class, with rates for each class varying across strains and species (see main text ***Equation 8***).

## Microbiome dynamics

Patient microbiota are considered in terms of their population dynamic stability, and are assumed either to be at stable equilibrium or undergoing temporary dysbiosis (population dynamic disequilibrium). Microbiome stability is disrupted via antibiotic treatment according to

$$\sigma_m = a \times \theta_m \tag{A8}$$

For model application, rates of antibiotic-induced microbiome dysbiosis are also considered at the level of antibiotic class (see main text ***Equation 7***). Microbiome stability recovers at rate $\delta$, but only among patients not actively undergoing antibiotic treatment, such that

$$\delta = \delta' \times (1 - a) \tag{A9}$$

where $\delta$ 'is the baseline rate in untreated hosts. Further, during model application we assume that microbiota can still recover among patients exposed to aminoglycosides and tetracyclines, the antibiotic classes inducing microbiome dysbiosis at the lowest rate ($k=3$, see Materials and methods), such that

$$\delta = \delta' \times (1 - a \times (1 - a_{k=3})) \tag{A10}$$

## Microbiome-pathogen competition

We assume that the host microbiome and the focal pathogen compete ecologically within the host, such that host microbiota limit pathogen transmission, persistence and endogenous acquisition. We propose three examples of microbiome-pathogen competition – colonization resistance ($\varepsilon$), resource competition ($\eta$), and ecological release ($\phi$) – and conceptualize how they affect pathogen colonization processes, and by extension how microbiome dysbiosis predisposes patients to pathogen colonization (see ***Figure 2*** in the main text). Interactions are given by

$$\begin{aligned} \beta_\varepsilon &= (1 - \varepsilon) \times \beta \\ \gamma_\eta &= (1 - \eta) \times \gamma \\ \alpha_\phi &= \phi \times \alpha \end{aligned} \tag{1}$$

where, respectively, ε reduces the transmission rate to patients with stable microbiota, η reduces the clearance rate among patients undergoing dysbiosis, and ϕ favours endogenous acquisition among patients undergoing dysbiosis. We assume that microbiome-pathogen interactions are species- and not strain-specific, that is that ε, η and ϕ apply equally to $P^S$ and $P^R$.

## Horizontal gene transfer

HGT is conceptualized as two-way within-host transfer of the focal resistance gene R, either from an R-bearing pathogen strain $P^R$ to co-colonizing microbiota, or from R-bearing microbiota to a co-colonizing drug-sensitive pathogen strain $P^S$. For simplicity, we assumed (i) a symmetric rate of HGT (χ) from each donor, (ii) no loss of resistance upon donation, (iii) no accumulation of resistance (e.g. plasmid copy number dependence), (iv) that all patients bear microbiota capable of receiving and transferring R, (v) that dysbiosis can accelerate the rate of transfer ($\chi_d \geq \chi_e$), (vi) no impact of R on rates of microbiome dysbiosis and recovery, (vii) that a proportion $f_\omega$ of patients are admitted to hospital with microbiota bearing R, and lastly (viii) that microbiota of a proportion ω of patients can spontaneously acquire R subsequent to dysbiosis. The latter assumption was made to reflect increased expression of antibiotic resistance genes among host microbiota following antibiotic therapy, and can be interpreted as a corollary to endogenous pathogen acquisition (*Ruppé et al., 2019*).

## Public health interventions

Three public health interventions τ were incorporated into Appendix *Equation A1* for subsequent model application. First, contact precautions were assumed to represent physical or behavioral barriers that block opportunities for transmission, reducing transmission rates by the same fraction $\tau_{ipc}$ across all pathogens relative to baseline. This is given by

$$\beta_{ipc} = \left(1 - \tau_{ipc}\right) \times \beta \tag{A12}$$

Second, antibiotic stewardship programmes were assumed to alter antibiotic consumption patterns in the hospital. Two main types were considered: antibiotic reduction, which limits overall antibiotic prescribing by a fraction $\tau_{asp}$, given by

$$a_{asp} = \left(1 - \tau_{asp}\right) \times a \tag{A13}$$

and antibiotic restriction, which adjusts the distribution of antibiotic classes consumed in the hospital. We considered two types of restriction, the first favouring classically narrow-spectrum antibiotics (e.g. macrolides) over broad-spectrum (e.g. quinolones), and the second favouring antibiotics that cause dysbiosis at very low or low rates (k={3,2}) over those causing dysbiosis at medium or high rates (k={1,0}). For both, the proportion of antibiotics in the restricted group ($p_{restrict}$) was reduced by the same fraction $\tau_{asp}$ without adjusting the distribution of antibiotics consumed within that group, given by

$$p_{restrict|asp} = p_{restrict} - \tau_{asp} \tag{A14}$$

and the proportion of non-restricted antibiotics was increased symmetrically. With this structure, $\tau_{asp}$ alters antibiotic consumption for the same proportion of hospital patients across all three stewardship interventions. In all simulations, $p_{restrict} > \tau_{asp}$.

Third, microbiome recovery therapy was assumed to trigger recovery at rate 0.5 day$^{-1}$ and was apportioned to the fraction $\tau_{pbt}$ of patients, such that the overall rate of microbiome recovery when including these interventions ($\delta_{pbt}$) is given by

$$\delta_{pbt} = \delta + 0.5 \times \tau_{pbt} \tag{A15}$$

For simplicity, different values assumed for intervention parameters ($\tau_{ipc}$, $\tau_{asp}$, $\tau_{pbt}$) are interpreted as different levels of compliance to the respective interventions.

## Part 1: Model evaluation and parameterization

### $R_0$ expressions

$R_0$ was calculated for $P^R$ across *Equations 1, 3, 4 and 6* from the main text (see Mathematica notebook), and was interpreted as the expected number of secondary patients colonized by an index patient in a fully susceptible (uncolonized) hospital population (i.e. at *disease-free equilibrium*, DFE, as indicated by the equilibrium symbol $\hat{}$ above state variables). To reflect this epidemiological context, we made two additional assumptions specifically for $R_0$ calculations: no $C^R$ input from the community ($f_R=0$), and that the rate of endogenous acquisition $\alpha_R$ scales linearly with current colonization prevalence $C^R$. $R_0$ expressions were derived following *van den Driessche, 2017*.

For the susceptible-colonized model,

$$R_0 = \hat{S}\frac{\beta + \alpha_R}{\gamma_R + \sigma_R + \mu} \tag{A16}$$

where $\hat{S} = 1$, such that pathogens with higher rates of transmission ($\beta$) and endogenous acquisition ($\alpha_R$) have higher $R_0$, while those with higher rates of natural clearance ($\gamma_R$) have lower $R_0$. Higher rates of effective antibiotic treatment ($\sigma_R$) and patient admission and discharge ($\mu$) also reduce $R_0$.

For the strain competition model, we derive $R_0$ for $P^R$ assuming that $P^S$ is at endemic equilibrium with input from the community ($f_C > 0$, $f_R = 0$). In this context, the same $R_0$ expression was found as for the susceptible-colonized model (Appendix *Equation A16*), except $R_0$ is reduced because of a lower equilibrium prevalence of susceptible hosts when $P^S$ is endemic ($\hat{S}<1$). DFE were not found analytically for models with strain competition and were solved numerically (details below).

For the microbiome model, $R_0$ was calculated using next-generation theory as the spectral radius of the next-generation matrix (NGM),

$$R_0 = \rho(\mathbf{NGM}) \tag{A17}$$

where NGM is the product of the transmission matrix **F** (describing the rates at which existing colonies cause colonization in new hosts) and the inverse transition matrix **V** (describing the rates at which colonized hosts shift between colonized classes or are removed). These are given by

$$\mathbf{F} = \begin{pmatrix} (\alpha_R + (1-\varepsilon) \times \beta) \times \hat{S}_e & (\alpha_R + (1-\varepsilon) \times \beta) \times \hat{S}_e \\ (\alpha_R \times \phi + \beta) \times \hat{S}_d & (\alpha_R \times \phi + \beta) \times \hat{S}_d \end{pmatrix} \tag{A18}$$

and

$$\mathbf{V} = \begin{pmatrix} \gamma_R + \sigma_R + \sigma_m + \mu & -\delta \\ -\sigma_m & \delta + (1-\eta) \times \gamma_R + \sigma_R + \mu \end{pmatrix} \tag{A19}$$

which give

$$R_0 = \frac{\hat{S}_e \times (\alpha_R + (1-\varepsilon) \times \beta)(\delta + (1-\eta) \times \gamma_R + \sigma_m + \sigma_R + \mu) + \hat{S}_d \times (\alpha_R \times \phi + \beta)(\delta + \gamma_R + \sigma_m + \sigma_R + \mu)}{(\sigma_R + \mu)(\delta + \sigma_m + \sigma_R + \mu) + (\delta + \sigma_R + \mu + (1-\eta) \times (\sigma_m + \sigma_R + \mu)) \times \gamma_R + (1-\eta) \times \gamma_R^2} \tag{A20}$$

where the two terms in the numerator correspond to pathogen acquisition, respectively, in hosts with a stable microbiome and in those undergoing dysbiosis. Antibiotic-induced dysbiosis ($\sigma_m$) and treatment ($\sigma_R$) are found in both numerator and denominator.

When assuming no microbiome-pathogen interactions ($\varepsilon = 0$, $\eta = 0$, $\phi = 1$), DFE is

$$\{\hat{S}_e, \hat{S}_d, \hat{C}_e, \hat{C}_d\} = \left\{ \frac{\mu \times (1-f_d) + \delta}{\sigma_m + \mu + \delta}, \frac{\mu \times f_d + \sigma_m}{\sigma_m + \mu + \delta}, 0, 0 \right\} \tag{A21}$$

and $R_0$ reduces to the same expression as given by the susceptible-colonized model

$$R_0 = \frac{\beta + \alpha_R}{\gamma_R + \sigma_R + \mu} \tag{A22}$$

With two strains, as in the microbiome-strain competition model, the same $R_0$ expression is found for $P^R$ as the single-strain model, but when $P^S$ is endemic,

$$\hat{S}_e + \hat{S}_d < 1 \tag{A23}$$

so strain competition dampens $R_0$ relative to the single-strain microbiome competition model. $R_0$ expressions are evaluated numerically in *Figure 1* and *Appendix 1—figure 2* and *3*.

## Epidemiological outcomes
### Colonization prevalence

Colonization prevalence was defined as the proportion of hospital patients colonized ($C^R$) by the focal drug-resistant strain ($P^R$) at endemic equilibrium:

$$C^R = \hat{C}_{e,s}^R + \hat{C}_{e,r}^R + \hat{C}_{d,s}^R + \hat{C}_{d,r}^R \tag{A24}$$

Colonization prevalence of the drug-sensitive strain $P^S$ was also calculated:

$$C^S = \hat{C}_{e,s}^S + \hat{C}_{e,r}^S + \hat{C}_{d,s}^S + \hat{C}_{d,r}^S \tag{A25}$$

These were found by substituting a corresponding vector of numerical parameter values $\Omega$ and solving ODE systems through numerical integration. Solutions were found using the lsoda method of the function *ode* from the package *deSolve* in R (version 3.6.0). For evaluation of the theoretical pathogen (part 1), solutions were corroborated using the function NSolve in Mathematica.

### Resistance rate

The resistance rate was defined as the proportion of patients colonized with the focal resistant strain $P^R$ relative to the sensitive strain $P^S$, calculated using equilibrium prevalence values as $C^R/(C^S + C^R)$.

### Colonization incidence

Nosocomial colonization incidence was defined as the daily rate of colonization acquisition within the hospital, calculated separately for each route of acquisition. For $P^R$, incidence rates corresponding to transmission ($inc_\beta$), endogenous acquisition ($inc_\alpha$) and HGT ($inc_\chi$) were calculated as

$$
\begin{aligned}
inc_\beta &= \int_t^{t+1} \lambda_{R,\varepsilon} \times \left(S_{e,s} + S_{e,r}\right) + \lambda_R \times \left(S_{d,s} + S_{d,r}\right) dt \\
inc_\alpha &= \int_t^{t+1} \alpha_R \times \left(S_{e,s} + S_{e,r}\right) + \alpha_{R,\phi} \times \left(S_{d,s} + S_{d,r}\right) dt \\
inc_\chi &= \int_t^{t+1} \chi_e \times C_{e,r}^S + \chi_d \times C_{d,r}^S dt
\end{aligned}
\tag{A26}
$$

We evaluated incidence at dynamic equilibrium by solving ODE systems numerically for each $\Omega$, then using resulting equilibria as initial values for subsequent numerical integration to calculate incidence from time $t$ to $t+1$.

## Parameter space for $P^R$ colonization

**Appendix 1—table 1.** Parameter values and ranges for the generic pathogen $P^R$ evaluated over five different colonization models (*Figures 1*, *3* and *4*, and *Appendix 1—figures 1–7*).
For endogenous acquisition and microbiome recovery, rates presented are assumed rates in untreated hosts, represented by the' (prime) symbol. Model 1: susceptible-colonized model; Model 2: strain competition model; Model 3: microbiome competition model; Model 4: microbiome-strain competition model; Model 5: microbiome-strain competition model with HGT.

| | | | Value | Model | | | | |
|---|---|---|---|---|---|---|---|---|
| Symbol | Parameter | Unit | {Range} | 1 | 2 | 3 | 4 | 5 |

*Continued on next page*

*Appendix 1—table 1 continued*

| Symbol | Parameter | Unit | Value {Range} | Model 1 2 3 4 5 |
|---|---|---|---|---|
| **Pathogen colonization** | | | | |
| $\beta$ | Transmission rate | day$^{-1}$ | 0.2 | X X X X X |
| $\alpha'$ | Endogenous acquisition rate | day$^{-1}$ | 0.01 | X X X X X |
| $\gamma$ | natural clearance rate | day$^{-1}$ | 0.03 | X X X X X |
| c | Fitness cost of resistance | / | 1 | X X X X X |
| **Patient demography** | | | | |
| $\mu$ | Admission / discharge rate | day$^{-1}$ | 0.1 | X X X X X |
| $f_C$ | Admission fraction (colonized) | / | 0.1 | X X X X X |
| $f_R$ | Admission fraction (bearing resistant strain) | / | 0.5 | X X X X X |
| $f_d$ | Admission fraction (dysbiosis) | / | 0 | X X X |
| $f_\omega$ | Admission fraction (microbiota bearing resistance gene) | / | 0 | X |
| **Antibiotics** | | | | |
| a | Antibiotic exposure prevalence | / | 0.2 {0–1} | X X X X X |
| $r_R$ | Antibiotic resistance level (P$^R$) | / | 0.8 {0–1} | X X X X X |
| $r_S$ | Antibiotic resistance level (P$^S$) | / | 0 | X X X |
| $\theta_c$ | Antibiotic-induced pathogen clearance rate | day$^{-1}$ | 0.2 | X X X X X |
| $\theta_m$ | Antibiotic-induced microbiome dysbiosis rate | day$^{-1}$ | 1 | X X X |
| **Microbiome ecology** | | | | |
| $\varepsilon$ | Colonization resistance | / | 0.5 {0.2–0.8} | X X X |
| $\eta$ | Resource competition | / | 0.5 {0.2–0.8} | X X X |
| $\phi$ | Ecological release | / | 5 {2–8} | X X X |
| $\chi_e$ | HGT rate (equilibrium) | day$^{-1}$ | {0, 0.01, 0.1} | X |
| $\chi_d$ | HGT rate (dysbiosis) | day$^{-1}$ | $\chi_e \times 10$ | X |
| $\delta'$ | Microbiome recovery rate | day$^{-1}$ | 0.143 | X X X |
| $\omega$ | Proportion of patients acquiring the resistance gene among microbiota following antibiotic exposure | / | 0.01 | X |

## Part 1: Supplementary results

Here, we expand on results for Part one from the main text. For both strain competition and microbiome competition models, C$^R$ increases monotonically with antibiotic use under the assumption of complete antibiotic resistance ($r_R$ = 1) (**Appendix 1—figure 1**). In **Appendix 1—figure 2**, we evaluate $R_0$ in the context of each microbiome-pathogen interaction separately, and in concert, over a range of assumed values. This demonstrates how colonization resistance dampens pathogen $R_0$, while resource competition and ecological release augment it. Taken together, strong microbiome-pathogen interactions can protect patients from P$^R$ colonization in some conditions (e.g. at low a and $r_R$), while predisposing them to P$^R$ colonization in others (high a and $r_R$). In **Appendix 1—figure 3**, we demonstrate numerically that introducing a drug-sensitive strain P$^S$ to the microbiome model reduces $R_0$ of P$^R$. In **Appendix 1—figure 4**, we again demonstrate how the strength of a pathogen's microbiome interactions ($\varepsilon$, $\eta$, $\phi$) and the level of antibiotic resistance for P$^R$ ($r_R$) drive epidemiological consequences of antibiotic use, here disentangling the role of antibiotic-induced pathogen clearance ($\theta_c$) from antibiotic-induced microbiome dysbiosis ($\theta_m$). In **Appendix 1—figure 5**, we simulate dynamic responses of P$^R$ to public health interventions. First, dynamic equilibria were found, from which ODEs were integrated for an additional 365 days, evaluating C$^R$ and the resistance rate over

time in the context of each microbiome-pathogen interaction (separately and in concert). We introduced theoretical interventions at days 90 (halving $r_R$ from 0.8 to 0.4), 180 (halving $\theta_m$ from 1.0 to 0.5), and 270 (halving $a$ from 0.2 to 0.1), demonstrating that an otherwise identical pathogen can experience diverse, and sometimes opposing responses to public health interventions in the context of different microbiome-pathogen interactions and epidemiological outcomes.

In *Appendix 1—figure 6*, we show how HGT impacts $P^R$ colonization incidence, in contrast to prevalence and resistance rate as presented in *Figure 4* in the main text. In *Appendix 1—figure 7*, we demonstrate effects of HGT on pathogen colonization outcomes. First, we vary key epidemiological parameters over broad ranges to show how the absolute impact of HGT depends on how other parameters mediate colonization dynamics (*Appendix 1—figure 7A*). Second, we show how increasing the rate of HGT in hosts undergoing dysbiosis ($\chi_d$), while holding the rate constant in hosts with stable microbiota ($\chi_e$), augments HGT's impact but does not substantively shift the level of antibiotic exposure that maximizes $C^R$ (*Appendix 1—figure 7B*). Third, we demonstrate how effects of HGT depend on characteristics of the resistance gene being transmitted (*Appendix 1—figure 7C*). For instance, a metabolically costly but highly drug-resistant gene R (c = 2, $r_R$ = 0.8) is potentially disadvantageous for the pathogen species as a whole at low antibiotic use (reducing total pathogen prevalence) but advantageous at high antibiotic use (increasing total prevalence).

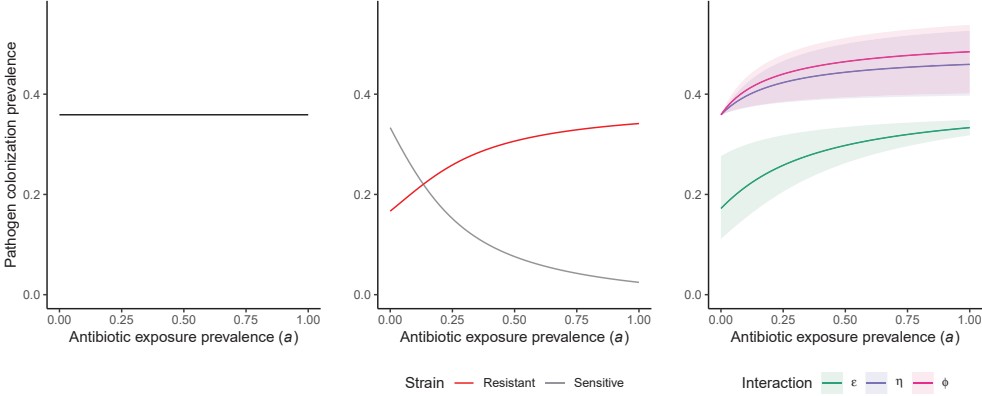

**Appendix 1—figure 1.** Antibiotic selection for the epidemiological spread of an antibiotic-resistant bacterial pathogen $P^R$ with full resistance to all antibiotics ($r_R$ = 1; for the middle panel, $r_S$ = 0). As in *Figure 1* in the main text, we compare results from the susceptible-colonized model (left), strain competition model (middle) and microbiome competition model (right). There are no selection trade-offs here because antibiotics have no epidemiological 'benefit', i.e. no ability to clear $P^R$. See *Appendix 1—table 1* for parameter values.

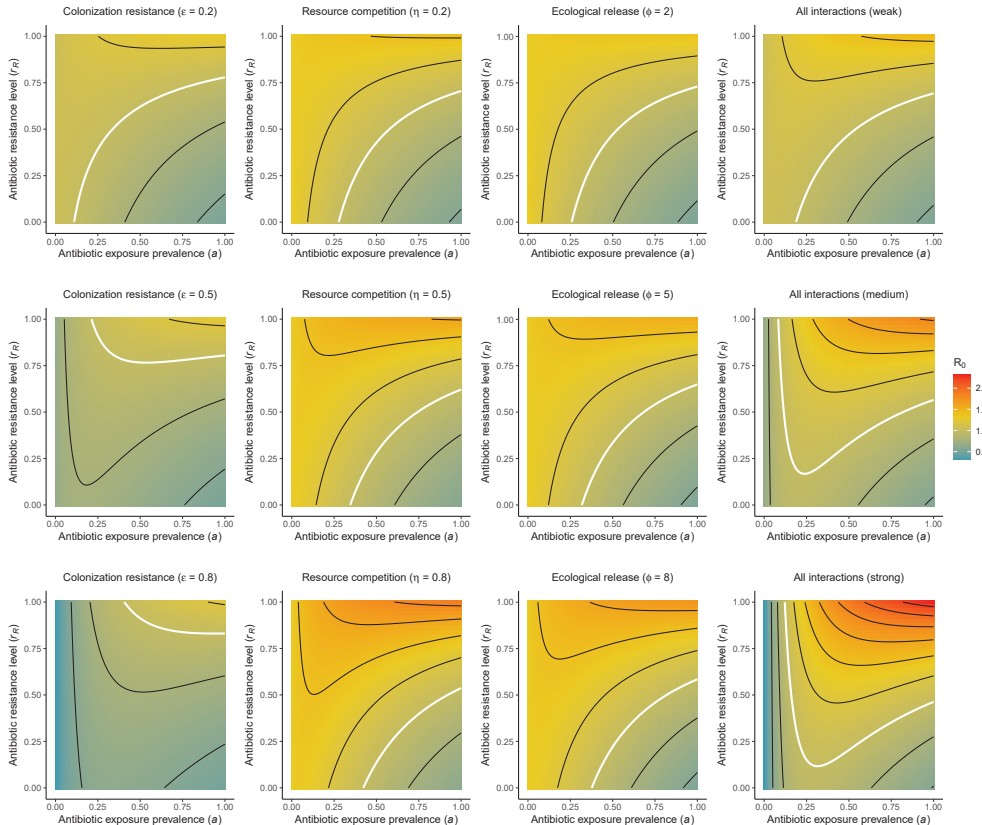

**Appendix 1—figure 2.** Different microbiome-pathogen interactions of different strengths (figure sub-titles) mediate how antibiotic use ($a$, x-axis) and resistance ($r_R$, y-axis) drive $R_0$ for $P^R$ (z-axis, color). White contour lines indicate $R_0=1$, and each successive black contour line represents an incremental change of 0.2. Microbiome-pathogen interactions are included separately (columns 1, 2, 3) and together (column 4), and their strengths are varied from weak (top row) to medium (middle row) to strong (bottom row) using values from *Figure 2* in the main text. See *Appendix 1—table 1* for all parameter values.

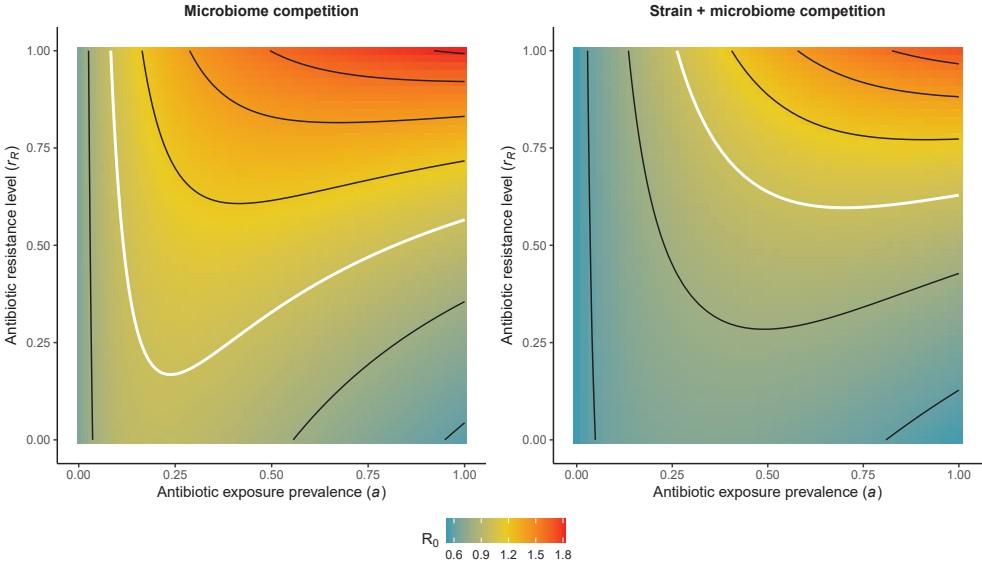

**Appendix 1—figure 3.** Introducing strain competition to the microbiome competition model reduces $R_0$ for $P^R$. Over the whole parameter space, the focal strain $P^R$ of a two-strain microbiome competition model (right) has a lower $R_0$ than the same pathogen evaluated in the absence of strain competition (left). We assume the competing strain $P^S$ is at endemic equilibrium and is completely sensitive to antibiotics ($r_S=0$). White contour lines indicate $R_0=1$, and each successive black contour line represents an incremental change of 0.2. See *Appendix 1—table 1* for all parameter values.

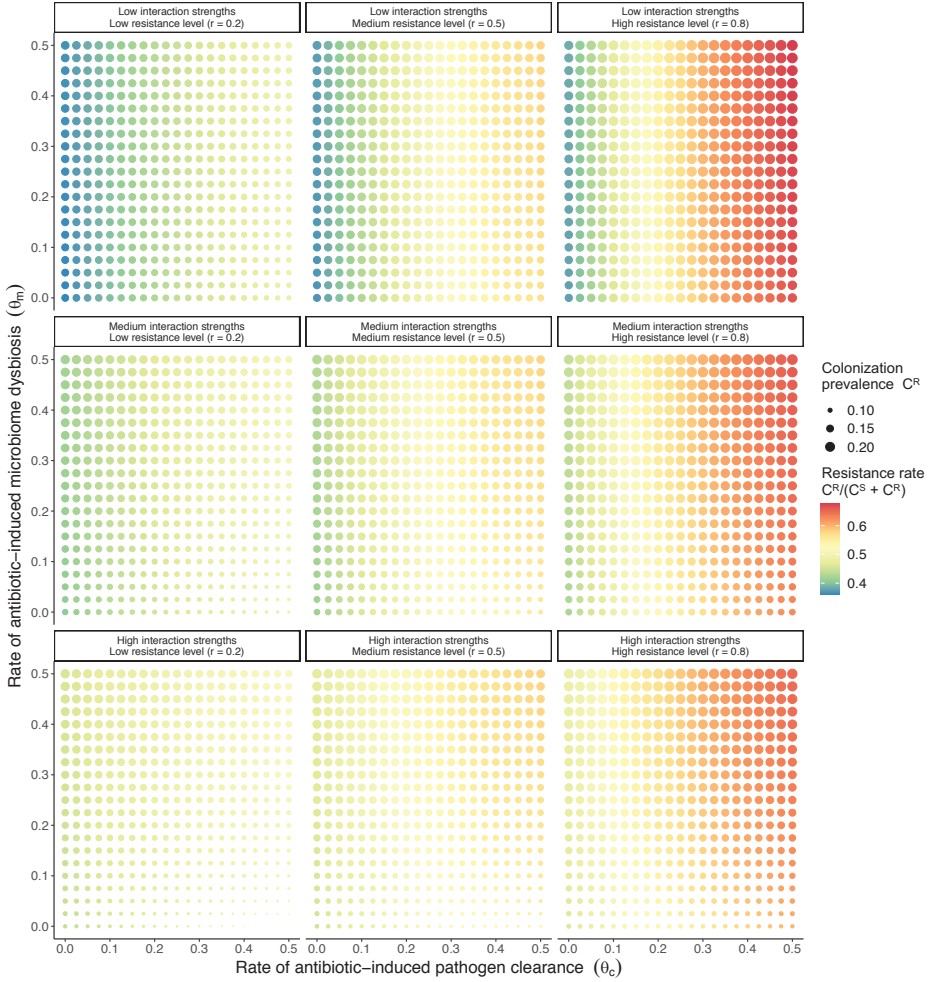

**Appendix 1—figure 4.** Antibiotic selection for the spread of an antibiotic-resistant pathogen strain $P^R$ depends on the strength of its interactions with microbiota (rows) and its level of resistance to antibiotics $r_R$ (columns). We assume complete antibiotic sensitivity for $P^S$ ($r_S = 0$), and low, medium, and high interactions strengths correspond to values in *Figure 2* in the main text, and columns in *Appendix 1—figure 2* ({$\varepsilon = 0.2$, $\eta = 0.2$, $\phi = 2$}, {$\varepsilon = 0.5$, $\eta = 0.5$, $\phi = 5$} and {$\varepsilon = 0.8$, $\eta = 0.8$, $\phi = 8$}, respectively). See *Appendix 1—table 1* for all parameter values.

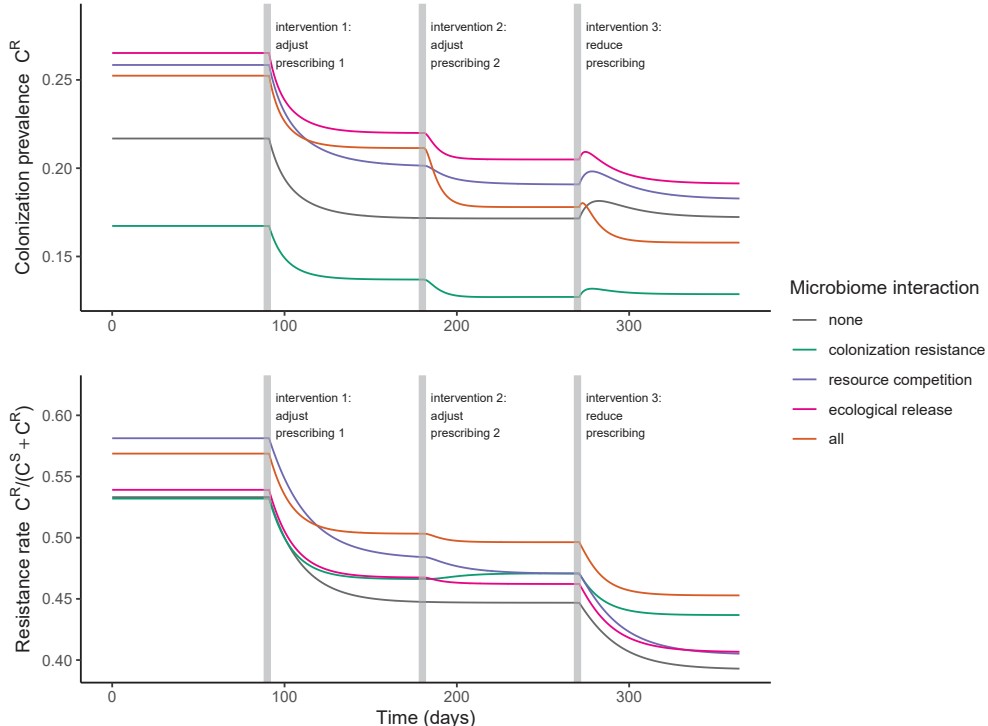

**Appendix 1—figure 5.** Antibiotic prescribing interventions have mixed impacts on $P^R$ colonization dynamics, depending on the microbiome interactions in effect (colors) and the epidemiological outcomes measured (top: colonization prevalence; bottom: the resistance rate). ODEs were integrated numerically for the baseline pathogen $P^R$, introducing successive interventions at 3, 6, and 9 months. Interventions represent changes to parameter values corresponding to presumed changes in antibiotic consumption: for intervention 1, $P^R$'s resistance level $r_R$ was halved from 0.8 to 0.4; for intervention 2, the rate of antibiotic-induced microbiome dysbiosis $\theta_m$ was halved from 1 to 0.5; and for intervention 3, the baseline antibiotic exposure prevalence $a$ was halved from 0.2 to 0.1. See *Appendix 1—table 1* for all parameter values.

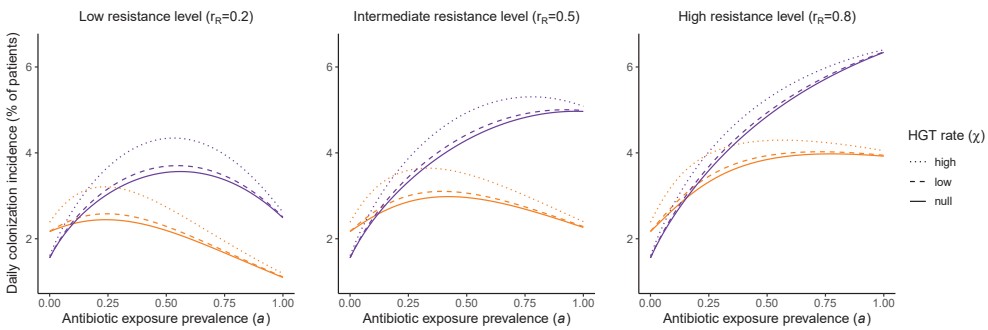

**Appendix 1—figure 6.** Impacts of horizontal gene transfer (HGT) on steady-state daily colonization incidence of $P^R$. The relative impact of HGT depends on the gene's rate of transfer ($\chi$, line type), antibiotic exposure prevalence ($a$, x-axis), competitive interactions between pathogen strains and host microbiota (colors: purple=combined strain and microbiome competition; orange=strain competition only), and the level of resistance conferred by the gene ($r_R$, columns). Assumptions match those for *Figure 4* in the main text. See *Appendix 1—table 1* for all parameter values.

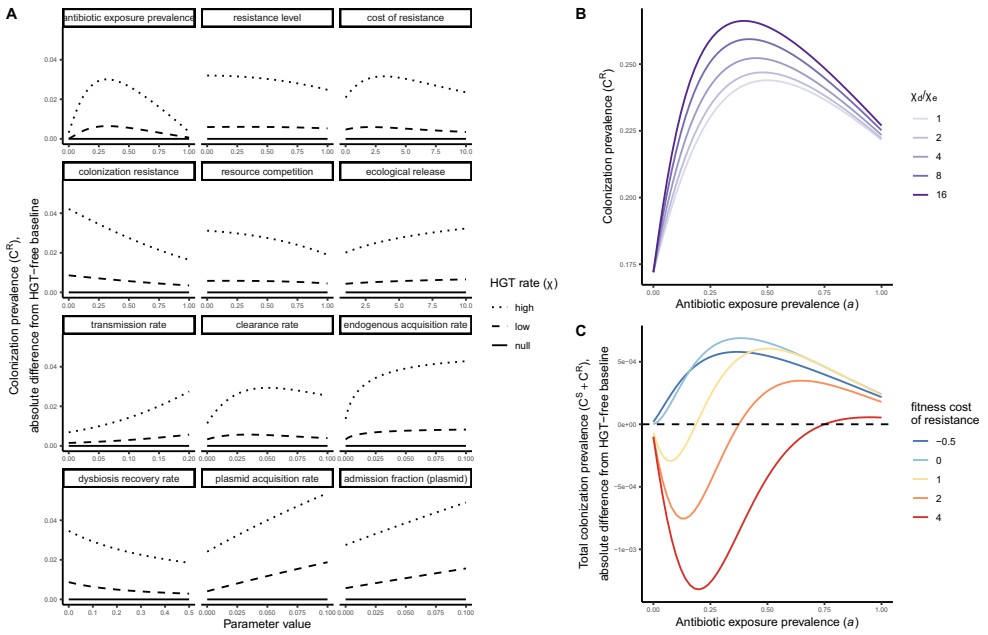

**Appendix 1—figure 7.** Impacts of HGT on pathogen colonization dynamics are tied to other parameters that mediate the prevalence of competing pathogen strains. (**A**) The absolute difference in $P^R$ colonization prevalence when including HGT (dashed and dotted lines) compared to prevalence in the absence of HGT (solid horizontal line) depends on assumed values of other parameters (panels) that drive colonization dynamics. For brevity, $\omega$ is described as the plasmid acquisition rate, and $f_\omega$ as the plasmid admission fraction. (**B**) Assuming a higher rate of HGT in patients undergoing dysbiosis ($\chi_d$) than in patients with stable microbiota ($\chi_e$) has a modest impact on $C^R$. Here, $\chi_e$ is held constant at $\chi_e=0.05$, such that changes in the fraction $\chi_d/\chi_e$ result from corresponding increases in $\chi_d$. (**C**) Impacts of HGT on total pathogen prevalence ($C^S + C^R$) depend on how selectively (dis)advantageous the resistance gene R is for the pathogen bearing it. Here, we show total prevalence (both strains) as proportional to a model assuming the same parameter values but excluding HGT (dashed horizontal line). Colors represent different fitness costs of resistance $c$, demonstrating that HGT not only changes the relative frequency of competing strains, but can feed forward to alter total prevalence of all strains, tending to increase total prevalence when R has little metabolic cost (low $c$), but decrease prevalence when R is costly (high $c$). See *Appendix 1—table 1* for all parameter values.

## Part 2: Model application, species characterization, and parameterization
### Host and pathogen parameterization

**Appendix 1—table 2.** Parameters and probability distributions for baseline hospital and host parameters applied across all ARB.

| Symbol | Parameter | Unit | Distribution | Reference | Reference setting | Notes |
|--------|-----------|------|--------------|-----------|-------------------|-------|
| $\mu$ | Admission / discharge rate | day$^{-1}$ | 1 / Normal (8, 2.55) | *Touat et al., 2019* | French hospitals | / |

*Continued on next page*

*Appendix 1—table 2 continued*

| Symbol | Parameter | Unit | Distribution | Reference | Reference setting | Notes |
|---|---|---|---|---|---|---|
| $f_d$ | Admission fraction (dysbiosis) | / | Normal (0.0756, 0.0190) | *Bernier et al., 2014* | French community | Taken as proportion of the French community exposed to antibiotics in previous 28 days, extrapolating weekly reimbursed antibiotic prescriptions/1000 inhabitants (18.9, 9.6–28.3) to 4 weeks and assuming independent prescriptions = 75.6 (38.4–113.2) prescriptions/1000 inhabitants |
| $a$ | Antibiotic exposure prevalence | / | Normal (0.195, 0.0195) | *Alfandari et al., 2015* | 314 French hospitals | / |
| $r_{i,j}$ | Antibiotic resistance level | / | if sensitive, 0; if resistant, 1; if intermediate sensitivity, $r_{i,j} \sim$ Uniform (0,1) | | | For all strains $i$, the resistance level to each antibiotic class $j$ depends on whether the strain is classified as sensitive, resistant, or of intermediate sensitivity in its assumed antibiogram |
| $\theta_c$ | Antibiotic-induced pathogen clearance rate | day$^{-1}$ | 1 / Uniform (1, 10) | *Tepekule et al., 2017* | Simulation study | / |
| $\theta_m$ | Antibiotic-induced microbiome dysbiosis rate | day$^{-1}$ | 1 / Normal (2, 0.4) | *Bhalodi et al., 2019* | Mixed | Circumstantial evidence of same-day microbiome disruption following antibiotic therapy; assumed an average minimum 12 hr to disruption |
| $\delta'$ | Microbiome recovery rate | day$^{-1}$ | 1 / Normal (28, 10.71) | *Burdet et al., 2019*; *Rafii et al., 2008* | Mixed; French hospital | Across studies in a review of antibiotic-induced microbiome disruption, intestinal microflora were observed to 'return to normal' 1–49 days after antibiotic cessation; in a French hospital, two measures of microbiome diversity were observed to 'return to normal' after 16–21 days. |

**Appendix 1—table 3.** Parameters and probability distributions for *C. difficile*.

| Symbol | Parameter | Unit | Distribution | Reference | Reference setting | Notes |
|---|---|---|---|---|---|---|
| $\beta$ | Transmission rate | day$^{-1}$ | Normal (0.00555, 0.000944) | *van Kleef et al., 2016* | English hospitals (modeling study) | Mean of point estimates of the daily probability of transmission from colonized patients (0.0037) and infected patients (0.0074) |
| $\alpha'$ | Endogenous acquisition rate | day$^{-1}$ | Normal (0.0000253, 0.0000114) | *Durham et al., 2016* | USA hospitals (modeling study) | Proxy measure: the estimated daily rate of progression from colonization to infection in hospital patients, divided by the relative risk of progression in patients exposed to antibiotics |
| $\gamma$ | Natural clearance rate | day$^{-1}$ | Normal (0.0119, 0.00170) | *Simor et al., 1993* | Canadian care home | Fit longitudinal colonization data using exponential decay model |

*Continued on next page*

*Appendix 1—table 3 continued*

| Symbol | Parameter | Unit | Distribution | Reference | Reference setting | Notes |
|--------|-----------|------|--------------|-----------|-------------------|-------|
| $f_C$ | Admission fraction (colonized) | / | Binomial (229, 0.048) / 229 | *Barbut, 1996* | 11 French hospitals | Stool prevalence among asymptomatic patients |
| $r$ | Antibiotic resistance level | / | median 94.3%, (range 93.3–95.4%) | estimated | / | Cumulative resistance level across simulated antibiotic consumption data and assumed antibiograms |
| ε | Colonization resistance | / | 1–1 / Cauchy (52.85, 1.62) | estimated | / | From expert opinion |
| η | Resource competition | / | $(1/\gamma)/(1/\gamma +$ Cauchy (121.11, 4.85)) | estimated | / | From expert opinion |
| φ | Ecological release | / | Cauchy (39.22 0.922) | estimated | / | From expert opinion |

**Appendix 1—table 4.** Parameters and probability distributions for *S. aureus*.

| Symbol | Parameter | Unit | Distribution | Reference | Reference setting | Notes |
|--------|-----------|------|--------------|-----------|-------------------|-------|
| β | Transmission rate | day$^{-1}$ | Normal (0.057, 0.0057) | *Di Ruscio et al., 2019* | Norwegian hospitals (modeling study) | / |
| α′ | Endogenous acquisition rate | day$^{-1}$ | Normal (0.0016, 0.0008) | *Coello et al., 1997*; *Di Ruscio et al., 2019* | Spanish hospital | Proxy measure: the estimated daily rate of progression from colonization to infection in hospital patients |
| γ | Natural clearance rate | day$^{-1}$ | 1 / Normal (287, 17.9) | *Shenoy et al., 2014* | Mixed | / |
| $c$ | Fitness cost of resistance | / | Normal (0.2, 0.02) | *Kouyos et al., 2013*; *Laurent et al., 2001* | French hospitals | Growth cultures showed 20% fitness benefit to MSSA over MRSA strains |
| $f_C$ | Admission fraction (colonized) | / | Normal (0.0757, 0.00364) | *Cravo Oliveira Hashiguchi et al., 2019*; *Scanvic et al., 2001* | French hospitals | Estimated as the proportion of patients arriving to a French hospital with MRSA colonization, divided by the estimated proportion of *S. aureus* strains that are methicillin-resistant in France |
| $f_R$ | Admission fraction (bearing resistant strain) | / | Normal (0.16, 0.016) | *Cravo Oliveira Hashiguchi et al., 2019* | France | / |
| $r_S$ | Antibiotic resistance level (MSSA) | / | median 33.1% (range 17.2–48.9%) | estimated | / | Cumulative resistance level across simulated antibiotic consumption data and assumed antibiograms |
| $r_R$ | Antibiotic resistance level (MRSA) | / | median 94.5% (range 90.8–98.2%) | estimated | / | Cumulative resistance level across simulated antibiotic consumption data and assumed antibiograms |

*Continued on next page*

*Appendix 1—table 4 continued*

| Symbol | Parameter | Unit | Distribution | Reference | Reference setting | Notes |
|---|---|---|---|---|---|---|
| ε | Colonization resistance | / | 1–1 / Cauchy (2.21, 0.15) | estimated | / | From expert opinion |
| η | Resource competition | / | $(1/\gamma)$ / $(1/\gamma$ + Cauchy (73.09, 3.09)) | estimated | / | From expert opinion |
| φ | Ecological release | / | Cauchy (2.97, 0.28) | estimated | / | From expert opinion |

**Appendix 1—table 5.** Parameters and probability distributions for *E. coli*.

| Symbol | Parameter | Unit | Distribution | Reference | Reference setting | Notes |
|---|---|---|---|---|---|---|
| β | Transmission rate | day$^{-1}$ | Normal (0.0078, 0.00334) | *Gurieva et al., 2018* | 13 European ICUs | / |
| α′ | Endogenous acquisition rate | day$^{-1}$ | Normal (0.0024, 0.000663) | *Gurieva et al., 2018* | 13 European ICUs | / |
| γ | Natural clearance rate | day$^{-1}$ | Normal (0.00269, 0.000216) | *Bar-Yoseph et al., 2016* | Mixed | Fit longitudinal colonization data using exponential decay model |
| c | Fitness cost of resistance | / | Normal (0.2, 0.02) | / | / | In absence of data for ESBL resistance, used same distribution as for MRSA |
| $f_C$ | Admission fraction (colonized) | / | Normal (0.275, 0.0140) | *Ebrahimi et al., 2016*; *Gurieva et al., 2018* | Mixed | Estimated as the proportion of patients arriving to 13 European ICUs with ESBL-EC carriage, divided by the estimated proportion of *E. coli* that are ESBL-producing in a Hungarian hospital |
| $f_R$ | Admission fraction (bearing resistant strain) | / | Normal (0.119, 0.0413) | *Ebrahimi et al., 2016* | Hungarian hospital | Proportion of fecal *E. coli* that were ESBL-producing from a non-outbreak setting |
| $f_\omega$ | Admission fraction (microbiota bearing ESBL gene) | | Binomial (857, 0.0665)/857 | *Pilmis et al., 2018*; *Vidal-Navarro et al., 2010* | 2 French hospitals | Estimated by pooling 857 samples from two studies reporting fecal carriage of ESBL-producing species other than *E. coli* |
| $r_S$ | Antibiotic resistance level (EC) | / | median 23.1% (range 9.6–36.5%) | Estimated | / | Cumulative resistance level across simulated antibiotic consumption data and assumed antibiograms |
| $r_R$ | Antibiotic resistance level (ESBL-EC) | / | median 84.9% (range 77.4–92.2%) | Estimated | / | Cumulative resistance level across simulated antibiotic consumption data and assumed antibiograms |
| ε | Colonization resistance | / | 1–1/Cauchy (6.06, 0.64) | Estimated | / | From expert opinion |

*Continued on next page*

*Appendix 1—table 5 continued*

| Symbol | Parameter | Unit | Distribution | Reference | Reference setting | Notes |
|---|---|---|---|---|---|---|
| η | Resource competition | / | $(1/\gamma)/(1/\gamma +$ Cauchy (76.38, 5.35)) | Estimated | / | From expert opinion |
| φ | Ecological release | / | Cauchy (11.80, 0.80) | Estimated | / | From expert opinion |
| $\chi_e$ | HGT rate (equilibrium) | day$^{-1}$ | $\chi_d$/Log-Normal (1.36, 0.81) | Estimated | / | From expert opinion |
| $\chi_d$ | HGT rate (dysbiosis) | day$^{-1}$ | -log(1-Weibull (0.94, 0.11)) / 10 | Estimated | / | From expert opinion |
| ω | Proportion of patients whose microbiota acquire ESBL gene following antibiotic exposure | / | Binomial (132, 18/132)/132 × 0.382 | *Agence nationale de sécurité du médicament et des produits de santé, 2017*; *Bar-Yoseph et al., 2016* | Mixed | The proportion of patients in a meta-analysis who, subsequent to treatment, express resistance to the antibiotic with which treated (18/132), multiplied by the proportion of ESBLs among antibiotics consumed in French hospitals (38.2%) |

**Appendix 1—table 6.** Parameters and probability distributions for *K. pneumoniae*.

| Symbol | Parameter | Unit | Distribution | Reference | Reference setting | Notes |
|---|---|---|---|---|---|---|
| β | Transmission rate | day$^{-1}$ | Normal (0.029, 0.00842) | *Gurieva et al., 2018* | 13 European ICUs | Estimate for non-*E. coli* Enterobacteriaceae |
| α′ | Endogenous acquisition rate | day$^{-1}$ | Normal (0.0048, 0.00133) | *Gurieva et al., 2018* | 13 European ICUs | Estimate for non-*E. coli* Enterobacteriaceae |
| γ | Natural clearance rate | day$^{-1}$ | Normal (0.00267, 0.000324) | *Bar-Yoseph et al., 2016* | Meta-analysis | Fit longitudinal colonization data using exponential decay model |
| c | Fitness cost of resistance | / | Normal (0.2, 0.02) | / | / | In absence of data for CP resistance, used same distribution as for MRSA |
| $f_C$ | Admission fraction (colonized) | / | Binomial (11420, (928 / 11420)) / 11420 | *Cravo Oliveira Hashiguchi et al., 2019*; *Gurieva et al., 2018* | Mixed | The proportion of patients arriving to 13 European ICUs with CP-KP carriage, divided by the estimated proportion of *K. pneumoniae* that produce carbapenemase in France |
| $f_R$ | Admission fraction (bearing resistant strain) | / | Normal (0.01,0.001) | *Cravo Oliveira Hashiguchi et al., 2019* | France | / |

*Continued on next page*

*Appendix 1—table 6 continued*

| Symbol | Parameter | Unit | Distribution | Reference | Reference setting | Notes |
|---|---|---|---|---|---|---|
| $f_\omega$ | Admission fraction (microbiota bearing CP gene) | | Binomial (1135, 0.00441) / 1135 | *Pantel et al., 2015* | 7 French hospitals | Rectal carriage of carbapenemase-producing bacteria |
| $r_S$ | Antibiotic resistance level (KP) | / | median 23.1% (range 9.6–36.5%) | estimated | / | Cumulative resistance level across simulated antibiotic consumption data and assumed antibiograms |
| $r_R$ | Antibiotic resistance level (CP-KP) | / | median 91.7% (range 89.7–93.7%) | estimated | / | Cumulative resistance level across simulated antibiotic consumption data and assumed antibiograms |
| $\varepsilon$ | Colonization resistance | / | 1–1 / Cauchy (17.16, 0.97) | estimated | / | From expert opinion |
| $\eta$ | Resource competition | / | $(1/\gamma)$ / $(1/\gamma +$ Cauchy (74.93, 4.03)) | estimated | / | From expert opinion |
| $\phi$ | Ecological release | / | Cauchy (36.63, 0.82) | estimated | / | From expert opinion |
| $\chi_e$ | HGT rate (equilibrium) | $day^{-1}$ | $\chi_d$ / Gamma (2.01,0.36) | estimated | / | From expert opinion |
| $\chi_d$ | HGT rate (dysbiosis) | $day^{-1}$ | -log(1-Gamma (0.54, 3.67))/ 10 | estimated | / | From expert opinion |
| $\omega$ | Proportion of patients whose microbiota acquire CP gene following antibiotic exposure | | Binomial (132, 0.1363) / 132 × 0.0151 | estimated from *Agence nationale de sécurité du médicament et des produits de santé, 2017*; *Bar-Yoseph et al., 2016* | Mixed | The proportion of patients in a meta-analysis who, subsequent to treatment, express resistance to the antibiotic with which treated (18/132), multiplied by the proportion of carbapenems among antibiotics consumed in French hospitals (1.5%) |

## Expert elicitation
### Protocol

Expert elicitation is a scientific consensus methodology involving the estimation of unknown parameter values from subject-matter experts. We developed an expert elicitation protocol, using published recommendations for elicitation of clinical parameters,(*Johnson et al., 2010*) to characterize the role of microbiome dysbiosis in driving hospital colonization dynamics across the ARB included in this study (provided separately). First, the scientific context of the study was introduced, including the potential epidemiological relevance of microbiome-pathogen interactions, and potential roles for intraspecific strain competition and microbiome dysbiosis in explaining how antibiotics select for resistance. Second, the MATCH uncertainty elicitation tool and 'chips and bins' method were introduced, allowing experts to build distributions visually in the form of histograms to quantify parameter values and associated uncertainty. (*Morris et al., 2014*) Third, potential cognitive biases were reviewed and tips for reliable parameter estimation were provided. Finally, experts were asked to respond to questions and quantify their beliefs and uncertainty about how microbiome dysbiosis

among hospital patients affects pathogen colonization processes (acquisition, clearance, growth and horizontal gene transfer). It was stressed that the goal of the elicitation was not to characterize precise numerical estimates of particular mechanistic interactions, but rather to quantify the relative impact of microbiome dysbiosis on different colonization outcomes in the hospital setting. Experts were initially identified through a review of authors of relevant publications in the field. At the end of each interview, experts were further asked to refer us to additional experts in the form of snowball sampling. Elicitation results are reported anonymously, but all experts agreed to their acknowledgment in this article.

## Results and interpretation

Of 23 invited experts, ten ultimately participated in elicitation interviews (response rate 43.5%). Their specialties include bacteriology, internal medicine, intensive care, infectious disease epidemiology and clinical microbiology. Model parameters, the clinical parameters that experts were asked to estimate, and assumed links between them are provided in *Appendix 1—table 7*. First, experts were asked about the relevance of intraspecific strain competition for the nosocomial epidemiology of each pathogen. Then, for key colonization parameters ($\beta$, $\alpha$, $\gamma$, $\chi$), experts were asked whether hospital patients experiencing antibiotic-induced dysbiosis are more or less likely to experience the corresponding colonization outcome than patients not experiencing dysbiosis (*Appendix 1—figure 8*). If yes, they were asked to quantify the relative impact of dysbiosis on that outcome $i$ for each ARB $j$ as a histogram (estimated using MATCH) given by a vector $x_{i,j,k}$ for each expert $k$. Experts made estimates for each ARB consecutively, such that parameters for each ARB were estimated relative to other included ARB.

Raw data for expert distributions are provided in *Appendix 1—figure 9*. Experts differed substantially in estimated ranges of different parameters, but Friedman's tests using medians of each parameter distribution suggested that the species rank order was conserved across experts, with *C. difficile* generally having the strongest estimated microbiome-pathogen interaction coefficients, and MRSA the weakest. To conserve species rank order when combining expert estimates $x_{i,j,k}$ to form pooled histograms, distributions were re-centred by the mean relative distance $z_{i,j}$ between each $j$ and the reference ARB (taken as MRSA) over all experts $k$ (*Appendix 1—figure 10*). For *C. difficile* and MRSA, HGT was excluded from simulations and pooled distributions were not re-centred. Each expert distribution was weighted equally, contributing to 10% of the final pooled distribution for each species and parameter. Pooled histograms were fit to six candidate distributions (normal, lognormal, Weibull, Cauchy, exponential and gamma), and the final distribution was selected as the fitted distribution giving the lowest Akaike Information Criterion (using the function fitdist from the R package fitdistrplus). Final fitted distributions for each parameter and ARB are given in *Appendix 1—table 3–6*, and for select parameters in *Figure 5*.

Among other observations, experts highlighted (i) uncertainty about an influence of strain competition on hospital colonization dynamics for all ARB, with the majority of experts believing strain competition was irrelevant for *C. difficile* but at least partially relevant for other ARB, (ii) unanimous certainty about a role for antibiotic-induced microbiome dysbiosis as a driver of colonization for *C. difficile*, ESBL-EC and CP-KP, with somewhat less certainty for MRSA, and (iii) certainty about a role for horizontal gene transfer (HGT) for ESBL-EC and CP-KP (*Appendix 1—figure 8*).

**Appendix 1—table 7.** Relationship between model parameters and the clinical parameters estimated by experts.

| Model parameter | | Clinical parameter | | | Assumed relationship between model and clinical parameters |
|---|---|---|---|---|---|
| Name | Symbol | Description | | Symbol | |
| Colonization resistance | $\varepsilon$ | Relative risk of acquiring colonization among patients experiencing microbiome dysbiosis | | $RR_\beta$ | $\varepsilon = 1 - \frac{1}{RR_\beta}$ |
| Resource competition | $\eta$ | Excess duration of colonization among patients experiencing microbiome dysbiosis | | $d$ | $\eta = \frac{\frac{1}{\gamma}}{\frac{1}{\gamma}+d}$ |
| Ecological release | $\phi$ | Relative risk of pathogen outgrowth among patients experiencing microbiome dysbiosis | | $RR_\alpha$ | $\phi = RR_\alpha$ |

*Continued on next page*

*Appendix 1—table 7 continued*

| Model parameter | | | | | Assumed relationship between model and clinical parameters |
|---|---|---|---|---|---|
| Name | Symbol | Description | | Symbol | |
| HGT rate (dysbiosis) | $\chi_d$ | Proportion of antibiotic-exposed patients colonized with the specified species that acquire the specified resistance via HGT during their hospital stay | | p$\chi_d$ | $\mathrm{p}_{\chi_d} = 1 - e^{-\chi_d \times \frac{1}{\mu}}$ |
| HGT rate (equilibrium) | $\chi_e$ | Relative risk of acquiring the specified resistance via HGT among patients experiencing microbiome dysbiosis | | RR$\chi$ | $\mathrm{RR}_\chi = \frac{\chi d}{\chi e}$ |

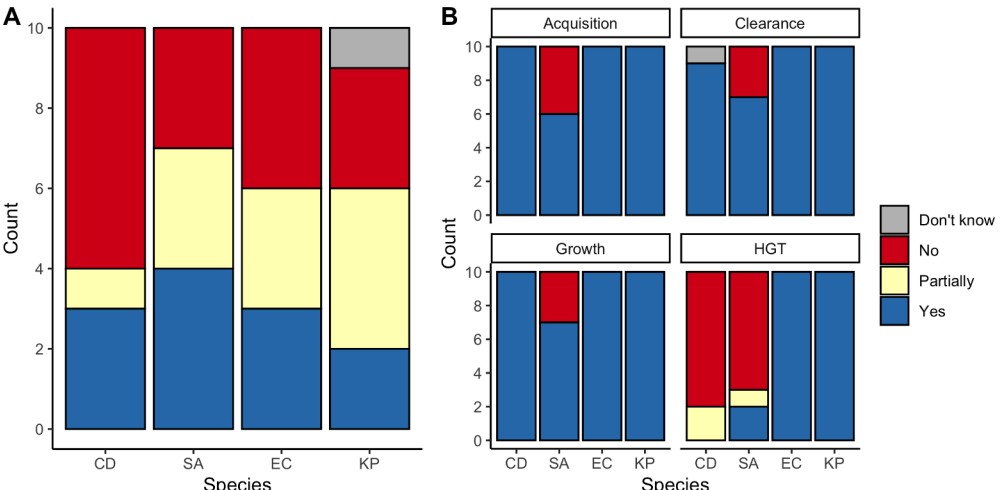

**Appendix 1—figure 8.** Expert belief about which mechanisms drive epidemiological dynamics of bacterial pathogens. (**A**) Expert belief in whether or not intraspecific strain competition influences nosocomial colonization dynamics for each pathogen. (**B**) Expert belief in whether or not each of the given colonization processes is affected by microbiome dysbiosis, and for HGT, whether or not this process is relevant in clinical settings. CD: *Clostridioides difficile*; SA: *Staphylococcus aureus*; EC: *Escherichia coli*; KP: *Klebsiella pneumoniae*; HGT: horizontal gene transfer.

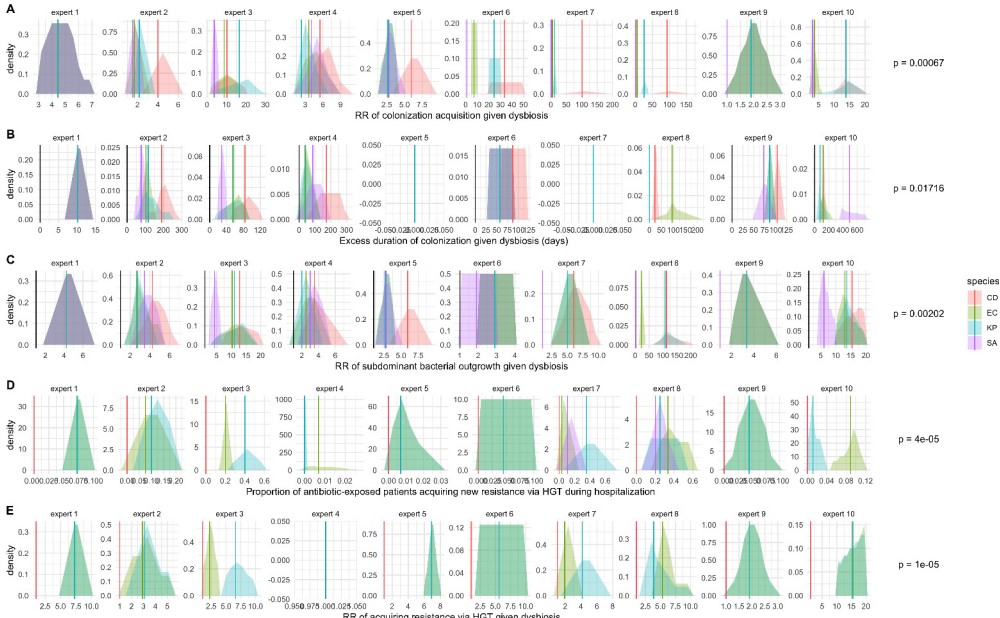

**Appendix 1—figure 9.** Expert elicitation results (raw data): expert belief and uncertainty about the impact of microbiome dysbiosis on nosocomial colonization dynamics of included bacterial pathogens (colors). Rows A through E represent, respectively, responses to questions two through six from the expert elicitation (protocol provided separately). Distributions were generated during expert interviews using the MATCH Uncertainty Elicitation Tool with the chips and bins method (*Morris et al., 2014*) Experts are anonymized and represented by different columns. Vertical bars represent medians of each distribution to visualize the rank order for each pathogen as estimated by each expert. p-values represent results of Friedman's tests for distribution medians, considering species as groups and experts as blocks for each question; when p<0.05, the species rank order across experts is interpreted as non-random.

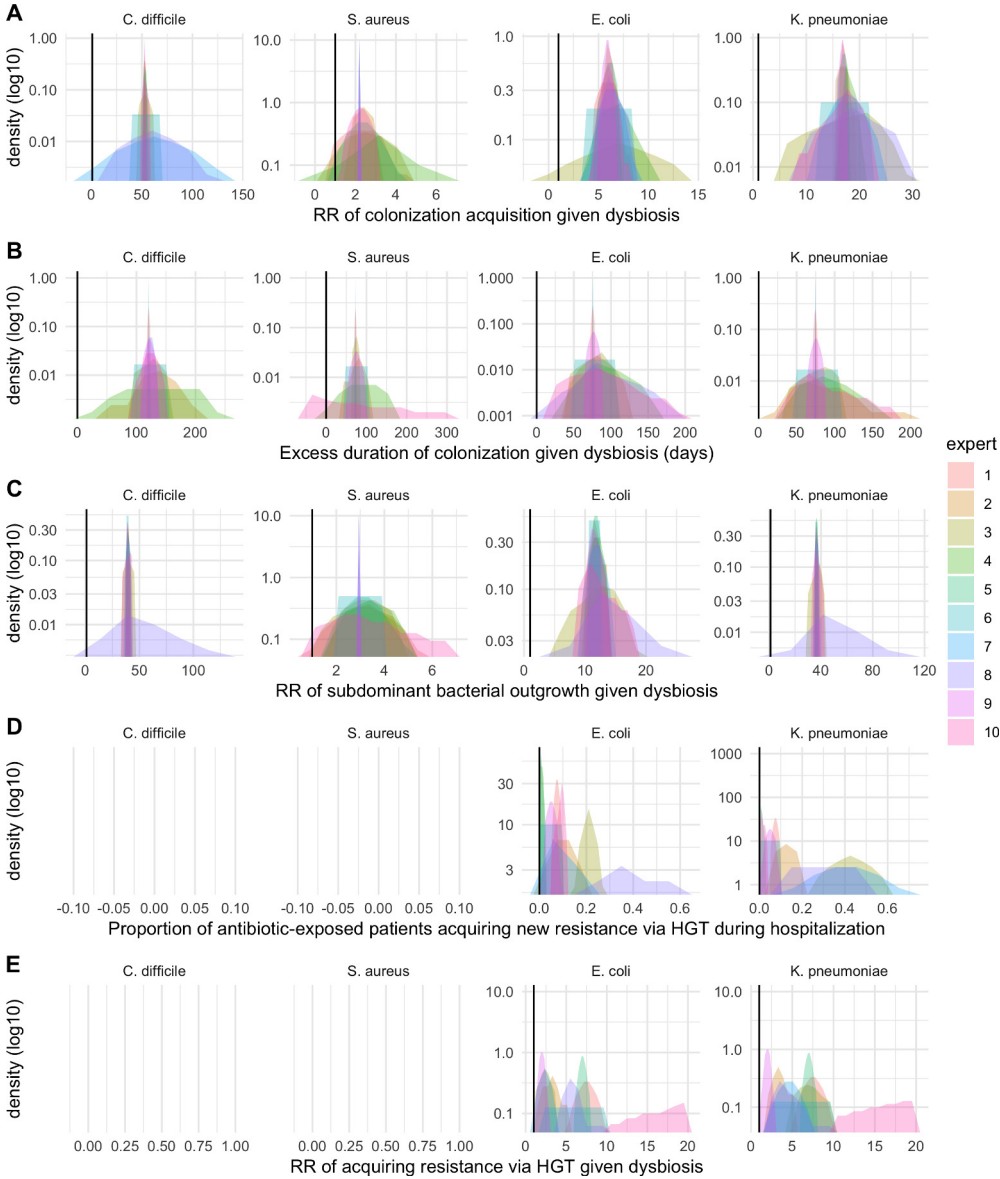

**Appendix 1—figure 10.** Expert elicitation results (re-centred data): expert belief and uncertainty about the impact of microbiome dysbiosis on nosocomial colonization dynamics of included bacterial pathogens (colors). Rows A through E represent, respectively, responses to questions two through six from the expert elicitation protocol (provided separately).

## Antibiotic exposure data

**Appendix 1—table 8.** Antibiotic classes, their contribution to total hospital antibiotic consumption, their spectrum, and their relative rate of inducing microbiome dysbiosis.
Consumption data come from the French ANSM and are supplemented with data from Baggs et al. (*Agence nationale de sécurité du médicament et des produits de santé, 2017*; *Baggs et al., 2016*). The literature was used to classify antibiotic classes in terms of their spectrum, (*Abbara et al., 2020*; *Tan et al., 2017*) and relative rate of causing microbiome dysbiosis. (*Baggs et al., 2018*; *Brown et al., 2013*) The percentage column does not total to 100 due to rounding error.

| Antibiotic class | ACT code | % of consumption | Spectrum | Rate of inducing microbiome dysbiosis |
|---|---|---|---|---|
| Amoxicillin and beta-lactamase inhibitor | J01CR02 | 32.4 | Broad | High |
| Penicillins with extended spectrum | J01CA | 21.9 | Narrow | Medium |
| Quinolones | J01M | 11.0 | Broad | High |
| C3G | J01DD | 8.2 | Broad | Very high |
| C1G | J01DB | 3.7 | Narrow | High |
| Macrolides | J01FA | 3.4 | Narrow | Medium |
| Imidazole | J01XD | 2.9 | Narrow | Medium |
| Piperacillin and beta-lactamase inhibitor | J01CR05 | 2.3 | Broad | High |
| Aminoglycosides | J01G | 2.3 | Narrow | Low |
| Tetracyclines | J01A | 2.0 | Narrow | Low |
| Sulfonamides, trimethoprim | J01E | 1.8 | Narrow | Medium |
| Glycopeptides | J01XA | 1.8 | Narrow | Very high |
| Lincosamides | J01FF | 1.6 | Broad | Very high |
| Carbapenems | J01DH | 1.5 | Broad | High |
| Penicillins (other) | J01C_other | 1.4 | Narrow | Medium |
| C2G | J01DC | 0.9 | Narrow | High |
| C4G | J01DE | 0.6 | Broad | Very high |
| Other | Other | 2.1 | Narrow | Medium |

## Part 2: Supplementary results
### Baseline colonization dynamics

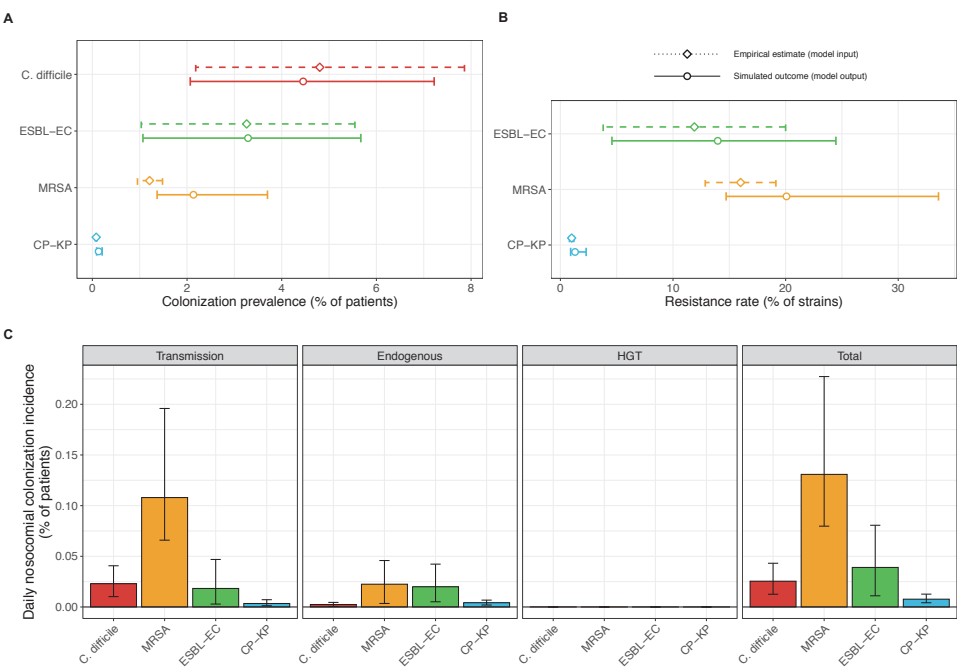

**Appendix 1—figure 11.** Baseline steady-state pathogen colonization outcomes for single-species
*Appendix 1—figure 11 continued on next page*

*Appendix 1—figure 11 continued*

simulations excluding the microbiome ($\varepsilon = 0$, $\eta = 0$, $\phi = 1$, $\chi = 0$). Compared to microbiome simulations (**Figure 6** in the main text), pathogens are less prevalent (**A**), and incidence is more than halved for all ARB except MRSA (**C**). However, resistance rates are largely unchanged (**B**). Points (**A** and **B**) and bars (**C**) represent medians, and error bars represent 95% uncertainty intervals across 10,000 Monte Carlo simulations.

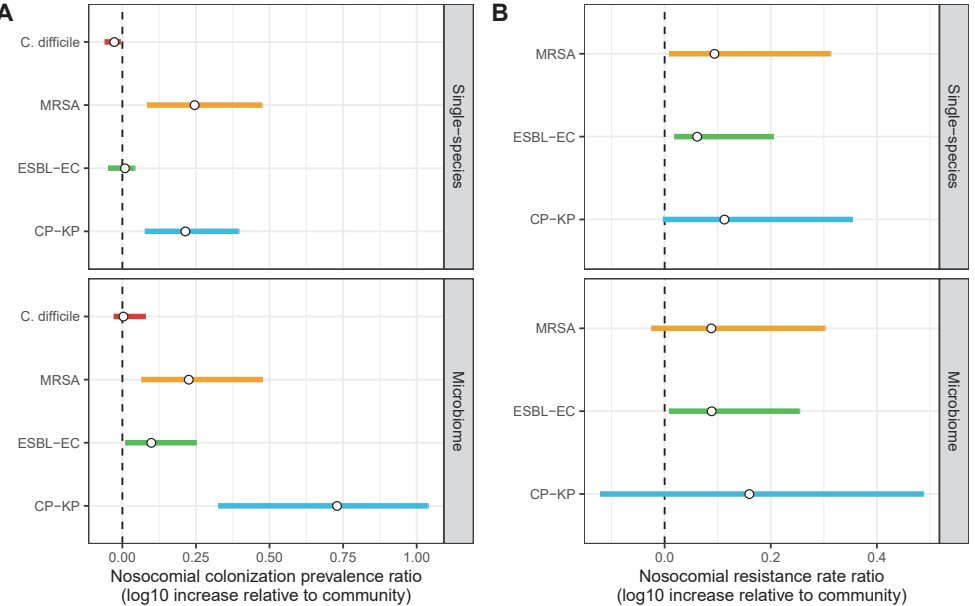

**Appendix 1—figure 12.** Change in ARB colonization outcomes in the hospital relative to the community ($\log_{10}$ scale), comparing single-species and microbiome simulations. (**A**) The ratio of colonization prevalence among hospital patients relative to baseline colonization prevalence in the community. (**B**) The ratio of resistance rates in the hospital relative to baseline resistance rates in the community. Points represent medians and error bars represent 95% uncertainty intervals across 10,000 Monte Carlo simulations.

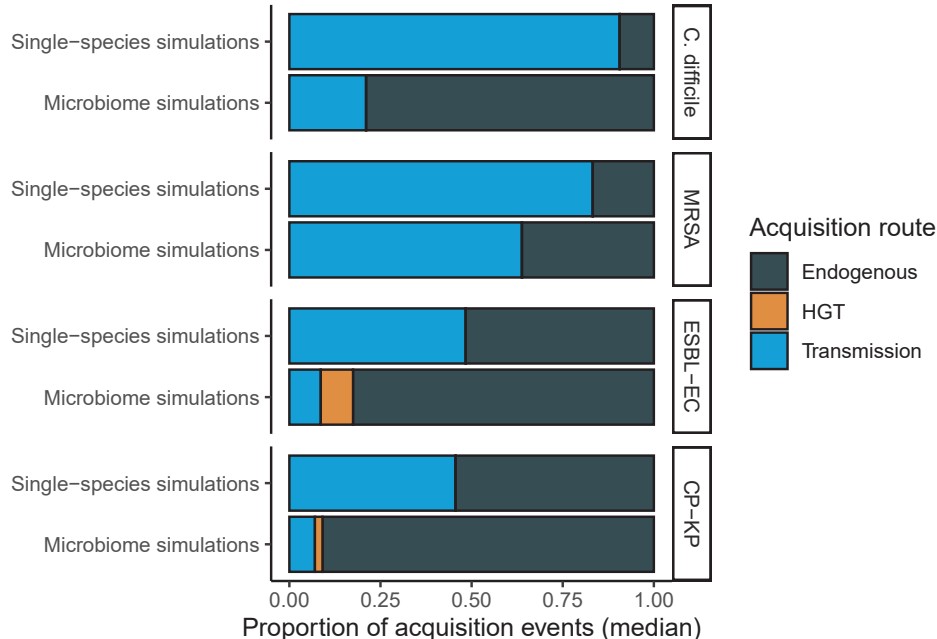

**Appendix 1—figure 13.** Different ARB spread in different ways, and the importance of different routes depends on potential microbiome interactions. Each shaded region represents the median estimated proportion of acquisition events explained by each acquisition route over 10,000 Monte Carlo simulations.

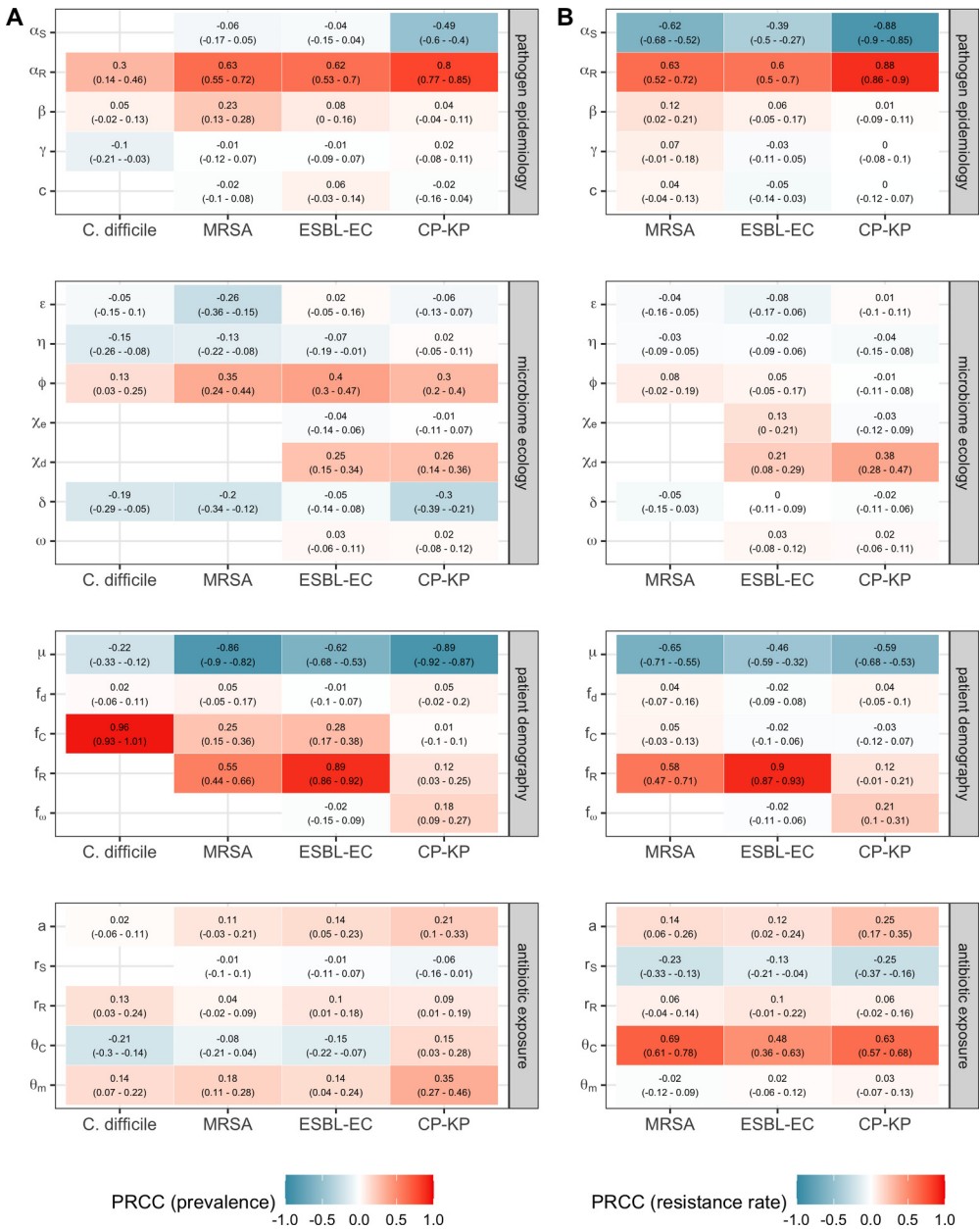

**Appendix 1—figure 14.** Multivariate sensitivity analysis describing partial rank correlation coefficients (PRCCs) between model parameters and two epidemiological outcomes evaluated at population dynamic equilibrium: in panel A, colonization prevalence of the focal strain $P^R$; in panel B, the pathogen resistance rate. For all pathogens, prevalence was positively associated with prior colonization upon hospital admission ($f_C$ for *C. difficile*, $f_R$ for MRSA, ESBL-EC and CP-KP) and ecological release ($\phi$), and negatively associated with a higher rate of discharge and admission, i.e. shorter duration of hospitalisation ($\mu$). Across ARB, higher rates of antibiotic-induced dysbiosis ($\theta_m$) and microbiome recovery ($\delta$) were generally positively and negatively associated with prevalence, respectively. Conversely, microbiome parameters were minimally associated with resistance rates, with the exception of the HGT rate among patients undergoing dysbiosis ($\chi_d$).

Intervention evaluation

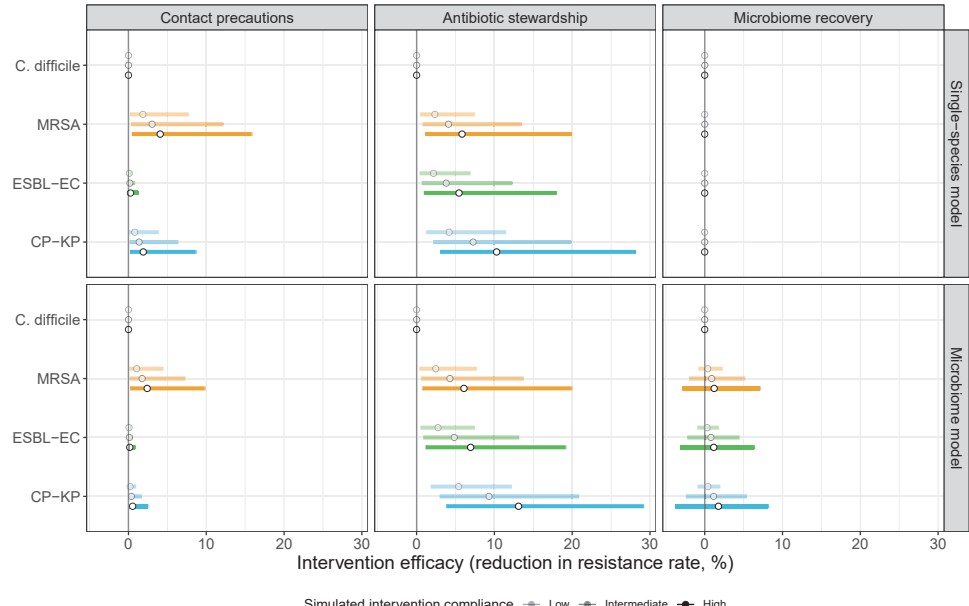

**Appendix 1—figure 15.** Compared to pathogen incidence (*Figure 7*), dynamic responses of pathogen resistance rates to public health interventions were similar across single-species and microbiome simulations (left panels, contact precautions; middle, antibiotic stewardship; right, microbiome recovery interventions). Top panels show results from simulations using 'single-species models' that only account for the focal pathogen species (including intraspecific strain competition for MRSA, ESBL-EC and CP-KP); bottom panels show simulation results when models also include microbiome-pathogen interactions and antibiotic-induced microbiome dysbiosis. For each intervention, three levels of intervention compliance (shading) are simulated. Points correspond to medians, and bars to 95% uncertainty intervals across 10,000 Monte Carlo simulations.

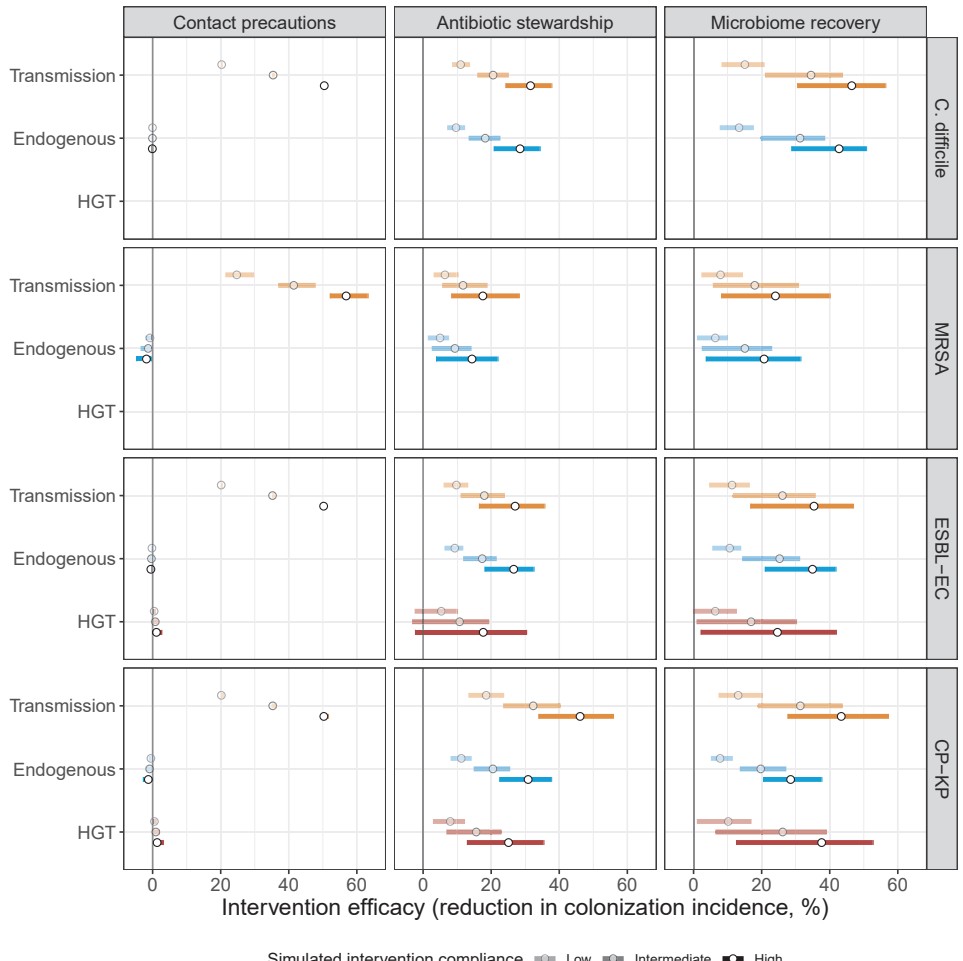

**Appendix 1—figure 16.** Interventions act on different routes of acquisition. Intervention efficacy (x-axis) for reducing colonization incidence via different routes of colonization acquisition (colors) for different interventions (columns) and ARB (rows). Unlike contact precautions, which only reduced incidence via transmission, antibiotic stewardship and microbiome recovery interventions reduced colonization incidence through all considered routes. Points correspond to medians, and bars to 95% uncertainty intervals across 10,000 Monte Carlo simulations.

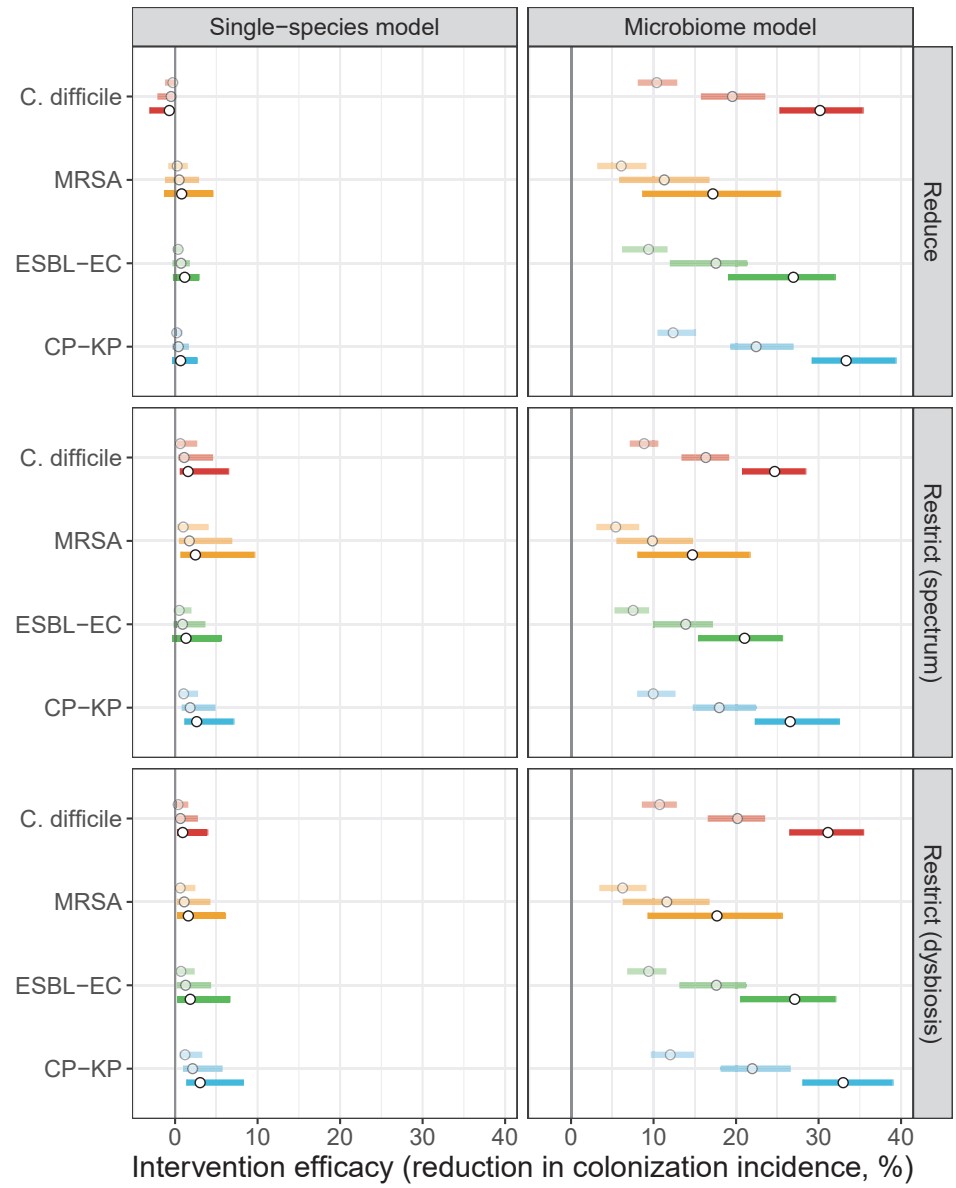

**Appendix 1—figure 17.** Intervention efficacy for three considered types of antibiotic stewardship: (**i**) reducing overall antibiotic prescribing, (ii) restricting broad-spectrum antibiotics in favour of narrow-spectrum antibiotics, and (iii) restricting antibiotics categorized as inducing microbiome dysbiosis at high or very high rates in favour of those that induce dysbiosis at medium or low rates. In microbiome simulations, restricting antibiotics that induce dysbiosis at a high rate was approximately as effective as reducing overall antibiotic prescribing. In single-species simulations, all stewardship interventions were of limited to negligible efficacy. Points correspond to medians, and bars to 95% uncertainty intervals across 10,000 Monte Carlo simulations.

## Meta-analysis of hospital antibiotic stewardship interventions

To assess consistency of intervention outcomes with the literature, we compared antibiotic steward-ship results to a systematic review and meta-analysis of hospital antibiotic stewardship interventions

initially conducted by *Baur et al., 2017.* They included interventional studies published worldwide from Jan 1960 to May 2016 describing incidence of bacterial colonization or infection among hospital inpatients. Stewardship interventions were heterogeneous and included audits, antibiotic restriction, antibiotic cycling, antibiotic mixing, feedback, guideline implementation and education. The authors found no evidence of publication bias or small study effects.

Their analysis included 32 studies over 9 million patient-days, though stewardship interventions were co-implemented with other infection control measures (e.g. hand hygiene, patient isolation) in 31% of studies. When limited to stewardship, patients were still significantly less likely to experience colonization and infection (IRR=0.81, 0.67–0.97), but species-specific estimates were not made. When excluding studies that included non-stewardship interventions, ten studies remained for MRSA, (*Arda et al., 2007*; *Frank et al., 1997*; *Mach et al., 2007*; *Marra et al., 2009*; *Meyer et al., 2010*; *Miyawaki et al., 2010*; *Peto et al., 2008*; *Smith et al., 2008*; *Yeo et al., 2012*; *Zou et al., 2015*) seven for *C. difficile*, (*Borde et al., 2015*; *Dubrovskaya et al., 2012*; *Frank et al., 1997*; *Leung et al., 2011*; *Lübbert et al., 2014*; *Malani et al., 2013*; *Schön et al., 2011*) two for ESBL-*E. coli*, (*Arda et al., 2007*; *Chong et al., 2013*) and one for CP-*K. pneumoniae* (*Arda et al., 2007*) For ESBL-*E.* coli, we further included studies of ESBL-producing Enterobacteriaceae and Gram-negative bacteria (*Grohs et al., 2014*; *Takesue et al., 2010*) Using their published data and methodology, we calculated incidence risk ratios (IRR) using the standard inverse-variance method, and pooled IRRs across species using mixed-effects meta-analysis models (R package *metafor*). Results for each study and pooled results for each pathogen are presented in *Appendix 1—figure 18*.

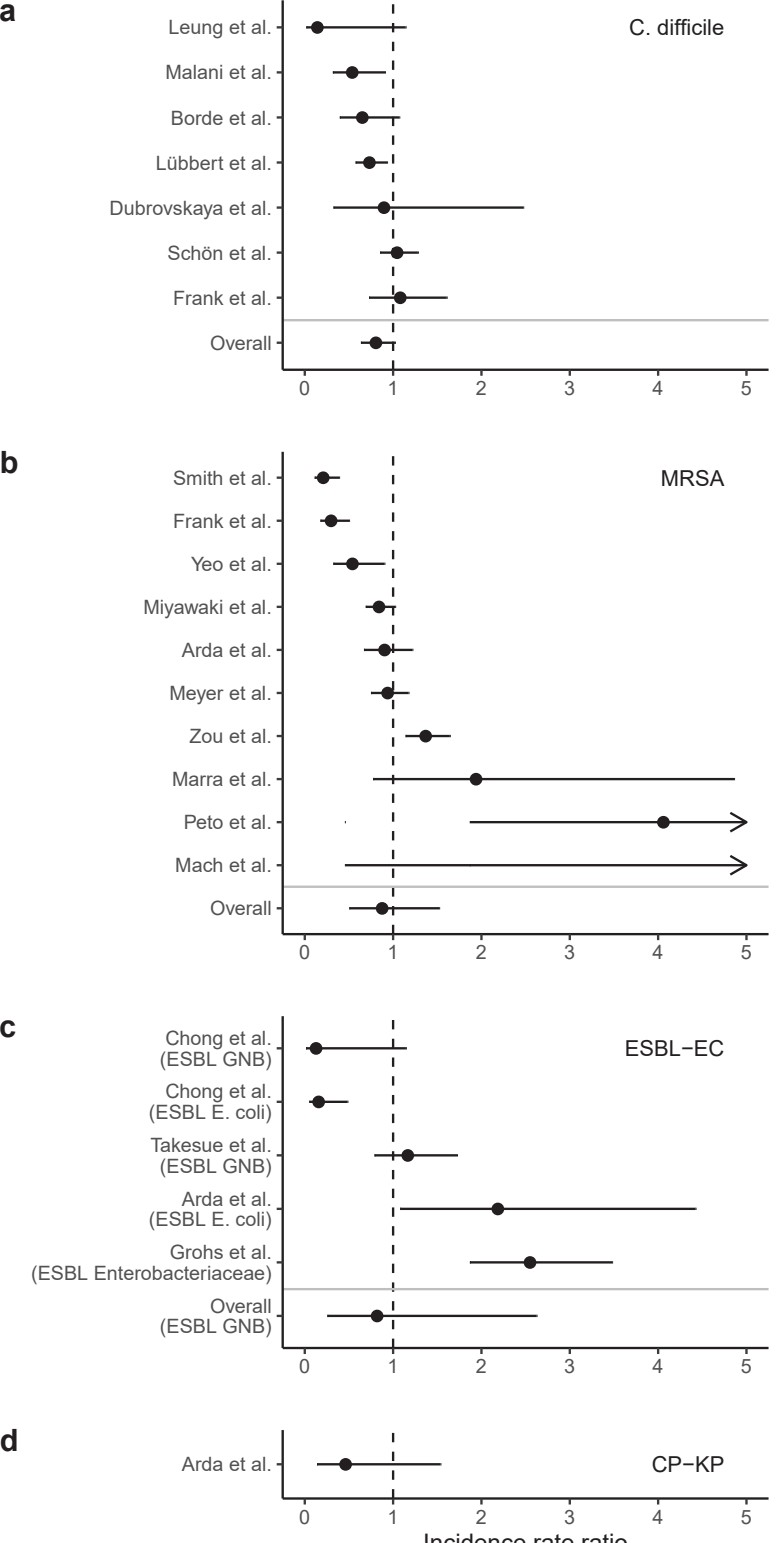

**Appendix 1—figure 18.** Incidence rate ratios (IRRs) for ARB colonization and infection among hospital inpatients exposed to antibiotic stewardship interventions. IRRs were calculated using data presented in *Baur et al., 2017* Results are stratified by four pathogens: (**a**) *C. difficile*, (**b**) methicillin-resistant *S. aureus*, (**c**) ESBL-producing *E. coli* (and here also including ESBL-producing

*Appendix 1—figure 18 continued on next page*

*Appendix 1—figure 18 continued*

Enterobacteriaceae or Gram-negative bacteria), and (**d**) carbapenemase-producing *K. pneumoniae*. Points represent means and bars represent 95% confidence intervals.

