## [Decision Letter]

**Acceptance summary:**

This paper provides a mathematical modelling framework for a stepwise incorporation of ecological complexity up to the effects of host microbiome on the spread of antibiotic-resistance. This provides a key first step in our understanding of the heterogeneous impact of the host microbiome on the spread of resistance, as demonstrated by the author's application of the model to four key pathogens.

**Decision letter after peer review:**

Thank you for submitting your article "Microbiome-pathogen interactions drive epidemiological dynamics of antibiotic resistance: modelling insights for infection control" for consideration by *eLife*. Your article has been reviewed by 3 peer reviewers, and the evaluation has been overseen by a Reviewing Editor and George Perry as the Senior Editor. The following individuals involved in review of your submission have agreed to reveal their identity: Esther van Kleef (Reviewer #1); Erik S Wright (Reviewer #2).

Summary:

The reviewers agreed that the approach is thoughtful, and the analysis performed is well complemented by simulations using relevant parameters in healthcare settings in hospitals. The big problem with this area, which is stressed by the authors, is the lack of information to support parameter choice. Despite being stressed, it was often not completely clear how and why parameters were chosen. Further clarity is required to support publication of this work which does offer a rational framework for linking the epidemiology and the microbiome, with an important hypothesis for antibiotic resistance.

Essential revisions:

The main revisions required are

– A restructuring of the methods to provide greater clarity of model structures and justification of parameter choices (is there no data? why was this chosen if so?). The driving force behind this should be the reproducibility of the work and the exploration of the results based on the parameter choices.

– A rewriting of the introduction and discussion to include more context of prior modelling work as well as to provide justification for the focus of the work

– In line with (1), all reviewers were in agreement that this work needs a clear single model description: a formal description of the model framework in one place. This could be in the supplementary, but it needs to be a well referenced place.

1) In general, the model the authors use is highly parameterized. There is a worry about circular reasoning when modelling. Parameters are derived from observations, then a model is constructed to recapitulate this observation. Then similar observations are used to show the model's validity. How did the authors ensure this type of bias was not incorporated into their modelling? This can become a bigger issue with more highly parameterized models, and that is why simpler models are preferable. Another approach is to use cross-validation, although we are not sure how that would work here.

Abstract

2) The Abstract is vague and somewhat convoluted as written. It did not do the best job of selling the paper and does not convey any specific information.

Introduction

3) The authors need to tailor this to their specific question: where is the knowledge gap? We recommend that the authors think carefully about the statements they are making and consider revising the Introduction to include a more thorough analysis of prior modelling work, which is largely absent from the Introduction.

4) The study focuses on four particular pathogens, however, no motivation of why they are chosen is provided, nor the relevance of their analysis in a more general overview is discussed. The authors could expand on how their results could be extended to other contexts/species in which the role of microbiome on resistance is also relevant. This could also be included in the discussion.

5) The authors conclude that disruption of the microbiome on an individual level (due to antibiotics) can result in selection for AMR on a population level. In a way, the work is, at least in part, a model-based implementation of the theoretical perspective of Lipsitch and Samore 2002 EID doi: 10.3201/eid0804.010312 (which describes how antibiotic effects on an individual, within-host level, can affect population level transmission dynamics). Could the comparison to Lipsitch and Samore and the novelty of the improved data-driven approach (although still limited) be further discussed?

6) Antibiotic stewardship is treated a single entity, whereas in reality there are countless types of stewardship interventions and a concomitant variety of outcomes. Comparing the model's results directly to an analysis of previous stewardship interventions and attempting to show alignment does not make sense to me. How were these stewardship interventions reflective of the model? Why would we expect agreement or disagreement a priori? The fact that there was agreement with the model was somewhat concerning given the complexity of many stewardship interventions.

Methods

7) In attempting to replicate this study, it became intractably difficult to recreate the ODEs because the equations and variables are interspersed throughout the main text and supplementary appendix. For example, in Figure 1e the authors mention differences in resistance levels (γ) and not transmission rate (λ, related to β as in Equation (3)), the value of which could not be found anywhere in the text. Similarly, where is patient demography (Δ) incorporated into equations 3 or 4? There are certainly answers to these questions, but it would have been helpful to have the model more clearly presented in a central location. Finding the parameters and making sense of them was extremely challenging, if not infeasible. Therefore, we were unable to replicate the results to any extent.

8) It is unclear how the authors arrived at their parameter values throughout the manuscript. For example, colonization resistance (epsilon) as it is formulated is not compelling, and where did the value of this parameter come from? It is listed in Table S5 as being drawn from a Cauchy distribution. Why? Where did these numbers come from? Some of the parameters seem justified by literature, but the rest are perhaps debatable. This might be inevitable with a complex model, but the authors should minimally provide a robustness analysis to each of their less-supported parameter choices. What would be the outcome if the sampled distributions had been wider (e.g., if the true value had been different by an order of magnitude)?

9) The inclusion of the 'r' parameter, allowing for partial resistance in the resistant strain, is interesting (and innovative for modelling frameworks to our knowledge). Nonetheless, it is not clear why the authors use a baseline of r = 0.8 to illustrate the trade-off between antibiotic induced clearance and selection for Cr (Figure 1F), whereas in figure 2, the authors use r=0.4, i.e. higher levels of sensitivity in Cr. I suppose with a lower r, strain coexistence is more likely, and thus higher likelihood of Horizontal Gene Transfer (HGT) (?), but it would be, for consistency and comparison of the different within-host dynamics, more transparent to use the same baseline parameter values, unless the authors can provide a good reason why different values of r are assumed at baseline between Figure 1 and Figure 2B?

10) The model presented in Eq. (1) needs a better introduction. Is this a new model or based on other models previously studied? This is explained later in the text citing references (30) and (31), but we recommend to introduce them earlier. It would be also good to point the reader earlier in the main text to the first sections of the supplement for a better understanding of the model assumptions and parameters employed.

11) Also, should the parameters be more consistently labelled across frameworks? For example, we recommend writing λ(N,C), or something similar, in all the equations to explicitly state it depends on these parameters. The way it is written gives the impression that λ is a constant parameter.

12) The r vs resistance rate (Cr/Cr+Cs) caused some confusion. This as a resistance rate of Cr = 0.8 but an r = 0.4 could actually mean that 0.8*0.4 = 0.08 of pathogens carried are fully resistant against the antibiotic, while this is 0.64 when r=0.8. Can the same baseline values for r be used in Figure 1 and Figure 2 or the differences justified?

13) Furthermore, in figure 1F: this is representing a one strain model. Is the Ce+Cd strain similar to the Cr? Can this be clarified?

14) In line with this. In Figure S3, R0 seems to represent the R0 of Cr. However, for Figure 1H and 1I, how should R0 be interpreted respectively? And for Figure S2? Please clarify how similar Cr (Figure 1H) and the one strain modelled (which could be similar to the Cr strain of Figure 1H) for figure 1I are

15) Figure 2A, the minimum resistance rate is 0.35 (see legend, and this appeared the case when no antibiotic induced clearance nor antibiotic induced microbiome disruption). This seems rather high. In particular as in Figure 2B, minimum levels of less than 0.1 are shown. Could the authors explain where this discrepancy is coming from and to what extend these high baseline resistance rates are representative?

16) For parameterisation of the models for the different pathogens, estimates from literature, notably existing modelling studies are chosen. However, these estimates, at least for C. difficile and MRSA, are coming from models that don't incorporate endogenous acquisition explicitly (for the Enterobacterieacia, model estimates from Gurieva are used, which do use a modelling framework incorporating an exo- and endogenous acquisition). Therefore, the acquisition rates may be overestimated for these gram-positives, in particular for C. diff the endogenous acquisition, which is listed as the main acquisition route (pp 14 line 330).

This may affect the estimated intervention effectiveness (in particular for antibiotic stewardship (less effective) and contact precautions (more effective)) under assumptions of the microbiome model. Could the authors use more realistic estimates (not sure models with explicit exo- and endogenous acquisition exist for MRSA and C. diff), or at least, reflect on how different values of α and β affect the model results?

17) It would be helpful to explicitly say how the simulations were performed, i.e., mention that the ODEs where integrated. My first impression was that the authors performed stochastic simulations using the processes illustrated in panels A, B, C from Figure 1. This is probably an issue that only mathematical modellers would have, but could the authors please add this clarification.

Discussion

18) The model is described by a set of differential equations. This means the model ignores stochasticity of the population size dynamics of each strain. The authors could expand a bit more on the assumptions of the model and why fluctuations are ignored.

19) Also, spatial structure is particularly important when taking into account interactions between microbiome and pathogens, but it is also neglected in the model. It would be useful if the authors could discuss this as well.

20) The results shown often focus on steady-state quantities (such as colonisation prevalence), but the temporal evolution of the model is not discussed. It would be interesting if the authors could investigate the timescale of the different interactions taken into account, and how relevant they may be in the healthcare settings they investigate. Is the temporal evolution of the model any relevant for their conclusions?

21) Can the authors please add examples of the hypothetical microbiome recovery interventions they have in mind? Are the authors thinking of things like faecal transplantation? Please add, also to provide more practical interpretation of the work.

*Reviewer #1:*

The authors use a theoretical dynamic transmission modelling framework, incorporating both between and within-host dynamics of antimicrobial resistant (AMR) pathogens, to stress the importance of the microbiome in the transmission dynamics of bacterial species. The work considered five different colonisation models, comprising different variations of 'traditional' epidemiological models, which, according to the authors, often do not include within-host microbiome-pathogen interactions, and add such interactions. With regards to the latter, three different within-host microbiome-pathogen interactions, as well as horizontal gene transfer are considered, and their effect on population transmission as well as interventions, evaluated.

The authors have done an extensive amount of work, and have done a great job in documenting all the model assumptions, data used and methods employed. Also, visualisations and illustrations are nice and informative.

However, the main difficulty with incorporating within-host dynamics of AMR pathogens in a human-to-human transmission modelling framework is the lack of data to inform key parameters. The authors stress this limitation is also part of their work. The authors emphasise that they aimed to show in theory the importance of microbiome-pathogen interactions, and have conducted an expert elicitation to partly inform parameters representing these dynamics.

The authors conclude that disruption of the microbiome on an individual level (due to antibiotics) can result in selection for AMR on a population level. In a way, the work is, at least in part, a model-based implementation of the theoretical perspective of Lipsitch and Samore 2002 EID doi: 10.3201/eid0804.010312 (which similar to here, describes how antibiotic effects on an individual, within-host level, can affect population level transmission dynamics). Therefore, I am somewhere inclined to think, are the main points made by the authors new? As this work here does, similar to this earlier work, not provide a data-driven approach (although tries to some extend). On the other hand, the work does, in contrast to Lipsitch and Samore, provide a framework for how to model these interactions, and illustrate to some extend which parameter values require what data, as well as which dynamics are most likely at play for which pathogens. Moreover, what is new, is that the authors try to illustrate how such interactions may affect the effectiveness of interventions, which is an interesting and, to the best of my knowledge, a novel component. My comments are largely requests for clarification.

*Reviewer #2:*

Here the authors tackle the important challenge of incorporating the microbiome into a traditional susceptible-colonized transmission model. The microbiome acts as a third party in infections, with roles in supplying or suppressing antibiotic resistance. The authors found that microbiome interventions hold the promise of avoiding antibiotic resistance dissemination. This theoretical work encourages continued research into how the microbiome could be used to mitigate antibiotic resistance. However, I am skeptical of the translatability of the model's predictions to real infections. The model is heavily parameterized, without clear reasoning for some parameter choices. The work's greatest contribution, in my opinion, is that it offers a rational framework for how the microbiome could be incorporated into epidemiology, and provides an intriguing hypothesis that the microbiome could play a substantial role in combating antibiotic resistance.

I would like to say the authors clearly did a good job of conceptualizing infections and resistance dissemination. This was a substantial body of work and the manuscript was interesting to read. My biggest concern is that some of the results intuitively reflect parameters of the model, suggesting they might be artifacts of the approach rather than independent results.

*Reviewer #3:*

This manuscript by Smith et al. proposes a novel mathematical framework that contributes to the understanding of the role of microbiome on the resistance colonisation of bacterial pathogen populations exposed to antibiotics. The model proposed incorporates microbiome competition and takes into account dysbiosis effects on the population dynamics. Extended versions of this model that incorporates strain-microbiome competition and horizontal gene transfer contributions are also presented. The model is simple to understand and takes into account important considerations in healthcare settings usually ignored in the literature. The simulations performed use parameters of four pathogens obtained carefully from a panel of experts. The fact they used these parameters makes the implications of their analysis quite relevant in clinical infections. Different assumptions on the interactions of each of the pathogens are incorporated in the model, and different public health interventions are analysed. The study performed is quite complete and supports the authors objectives. Their work demonstrate how relevant is the accounting for microbiome interactions for the understanding of antibiotics incidence on antibiotic resistance. This is particularly relevant for future work on mathematical modelling of resistance.

Even though the work presented is well supported by the analysis performed, there are certain aspects regarding the model that need to expanded. In particular:

1) The study focuses on four particular pathogens, however, no motivation of why they are chosen is provided, nor the relevance of their analysis in a more general overview is discussed. The authors could expand on how their results could be extended to other contexts/species in which the role of microbiome on resistance is also relevant.

2) The model is described by a set of differential equations. This means the model ignores stochasticity of the population size dynamics of each strain. The authors could expand a bit more on the assumptions of the model and why fluctuations are ignored. Also, spatial structure is particularly important when taking into account interactions between microbiome and pathogens, but it is also neglected in the model. It would be useful if the authors could discuss this as well.

3) The results shown often focus on steady-state quantities (such as colonisation prevalence), but the temporal evolution of the model is not discussed. It would be interesting if the authors could investigate the timescale of the different interactions taken into account, and how relevant they may be in the healthcare settings they investigate. Is the temporal evolution of the model any relevant for their conclusions?

---

## [Author Response]

Essential revisions:The main revisions required are– A restructuring of the methods to provide greater clarity of model structures and justification of parameter choices (is there no data? why was this chosen if so?). The driving force behind this should be the reproducibility of the work and the exploration of the results based on the parameter choices.

We have significantly restructured the paper to provide greater clarity of model structures and parameter choices. Changes include:

– Explicitly separating the work into two parts: Part 1, Development of a novel ODE modelling framework, evaluated using a generic nosocomial pathogen denoted P^R^; and Part 2, Model application to high-risk nosocomial pathogens with species-specific parameterization. This distinction is made clear from the end of the introduction.

– Part 1 now begins with a leading paragraph introducing the methods used and outcomes evaluated, with immediate reference to the Methods for more technical detail, and to the Supplement for the final model framework.

– The Methods section now begins by clearly explaining that the final model combines the 4 models described in the main text (bacterial colonization; strain competition; microbiome competition; strain-microbiome competition), and directs readers to the final model 5 incorporating HGT in supplementary equation S1.

– Model parameterization, including our use of expert elicitation to approximate microbiome-pathogen interaction coefficients, is expanded and more clearly presented from the end of the introduction, and when introducing Part 2 of the Model and Results section, with additional detail provided in the third paragraph of the Methods section, and with full context and methodological detail in the supplement (Appendix section 4.2, page 21), including raw elicitation data (Appendix figures 8-9) and final re-centred data (Appendix figure 10).

To support reproducibility of the work, in addition to the Mathematica notebook provided previously, R code has now been made available online at https://github.com/drmsmith/microbiomeR. Code includes all ODEs, functions used to calculate model results, parameter values for the generic pathogen P^R^, and reproduces figures from Part 1 of this work.

– A rewriting of the introduction and discussion to include more context of prior modelling work as well as to provide justification for the focus of the work

In the previous introduction paragraph describing existing modelling work, we now clearly state that antibiotic-induced microbiome dysbiosis is a long-standing hypothesis for the spread of resistance (and not our own), dating to at least Lipsitch and Samore 2002 *EID*.

“Disruption of the host microbiome is a long-standing theory explaining how antibiotics select for the spread of resistance at both the individual and population levels,(34) but most epidemiological models consider just one species of bacteria at a time, under the traditional assumption that antibiotic selection for resistance results from intraspecific competition between co-circulating strains.” (35–37)

We have included a new introductory paragraph describing contemporary modelling studies in the field of antibiotic resistance epidemiology. We stress the importance of within-host ecological competition among co-colonizing bacteria explored in contemporary work, e.g; Davies *et al.* 2019 *Nature Ecol Evol*.

“Accounting for other forms of complexity in epidemiological models – from treatment intensity, to age-assortative contact behaviour, to hospital referral networks, to animal-human interactions, to genetic linkage between resistance and non-resistance genes – has helped to unravel the many, disparate forces that contribute to drive the spread of resistance.(42–47) Nevertheless, within-host competition between co-colonizing bacteria is a key mechanism of selection for antibiotic resistance dissemination, and an active area of research at the forefront of resistance modelling.”

“For instance, the ‘mixed-carriage’ model by Davies *et al.* demonstrates how intraspecific competition results in negative frequency-dependent selection for either of two competing strains, and provides a satisfying mechanistic explanation for widespread strain coexistence at the population level.(49) However, contemporary work has stopped short of evaluating consequences of between-species competition on resistance epidemiology. Yet for many ARB, including emerging high-priority multidrug-resistant bacteria like extended-spectrum β-lactamase (ESBL) producing *Enterobacteriaceae*, interactions with the host microbiome may be important mediators of nosocomial colonization dynamics.”

Additional detail has also been provided in the Discussion to better contextualize our work, including:

– A sentence in the first paragraph broadly contextualizing our work:

“We present a modelling framework that includes within-host ecological costs of antibiotic use in the form of microbiome dysbiosis, incorporating a leading hypothesis for antibiotic selection into classical models of resistance epidemiology.”

– The new paragraph on microbiome recovery interventions now references relevant modelling work in this context: two within-host modelling papers newly released since initial submission of our work (Guittar et al. Am Nat 2021 doi: 10.1086/714527; Guk et al. Clin Pharmacol Ther 2021 doi: 10.1002/cpt.1977).

“Modelling has been used previously to evaluate impacts of such microbiome-oriented interventions at the within-host level,(74,75) but knock-on impacts on ARB transmission dynamics and epidemiological burden have not been evaluated previously.”

– The paragraph comparing modelling results to the literature now makes clear reference to which results come from modelling vs. clinical studies.

– The paragraph discussing alternative means of modeling antibiotic selection in previous work now makes a clear link to our work:

“These strategies reflect widespread recognition that antibiotic use favours ARB acquisition, through erosion of colonization resistance or other supposed mechanisms, and independent of potential competition with other strains. The present work formalizes examples of the microbiome-pathogen interactions that underlie these assumptions, demonstrating their relevance to various epidemiological outcomes, distinguishing them from strain-based selection, and providing a framework for their application.”

– In the first limitations paragraph, we provide examples of alternative modelling methods commonly used in the literature, and reference alternative factors not considered here (e.g. space, behaviour), but which have been found to drive antibiotic resistance dynamics in other modelling work:

“This study has a number of limitations. First, hospitals and healthcare settings are heterogeneous environments with non-random contact patterns and relatively small population sizes. Stochastic, individual-based models accounting for these factors reproduce more realistic nosocomial transmission dynamics than deterministic ODE simulations, allowing for local extinction events, super-spreaders, and other inherently random epidemiological phenomena. Nonetheless, our goal was to study how ecological mechanisms impact average epidemiological outcomes in the context of different model assumptions and parameter uncertainty, and in this context ODE modelling was the more appropriate tool, particularly for widely endemic ARB like *C. difficile*, MRSA and ESBL-EC. Still, further insights could certainly be gained by accounting for additional complexity and stochastic heterogeneity in future work, from within-host spatial organization,(91) to patient and staff contact behaviour,(92) to inter-institutional or inter-ward meta-population dynamics.(93) These distinctions may be particularly important for rare or non-endemic ARB (e.g. CP-KP in some regions).”

To provide further justification for the focus of the work, the introduction has been expanded:

– In intro paragraph 2, we provide a more clear description of relationships between antibiotic consumption, microbiome dysbiosis, and consequences for susceptibility to colonization (foreshadowing model content):

“Microbiome dysbiosis is associated with reduced abundance and diversity of commensal bacteria, impaired host immune responses, and loss of colonization resistance, altogether increasing host susceptibility to ARB colonization.(18–20) Antibiotic-induced dysbiosis may further result in elevated expression of antibiotic resistance genes, increased rates of horizontal transfer of such genes, and ecological release, whereby subdominant ARB are released from competition via clearance of drug-sensitive bacteria, growing out into dominant colonies.(21–24)”

– In intro paragraph 4, we more clearly define the research gap we have aimed to fill:

“However, contemporary work has stopped short of evaluating consequences of between-species competition on resistance epidemiology. Yet for many ARB, including emerging high-priority multidrug-resistant bacteria like extended-spectrum β-lactamase (ESBL) producing *Enterobacteriaceae*, interactions with the host microbiome may be important mediators of nosocomial colonization dynamics.(18,50,51)”

– In line with (1), all reviewers were in agreement that this work needs a clear single model description: a formal description of the model framework in one place. This could be in the supplementary, but it needs to be a well referenced place.

– A single model description is now provided from the beginning of the supplementary appendix (Appendix page 2), and is referenced throughout the main text, including at the end of the first paragraph of the Model and Results section; upon introduction of HGT at the end of part 1; and in first paragraph of the Methods, where we state:

“The final model, comprising pathogen colonization (equation 1), intraspecific strain competition (equation 3), microbiome-pathogen competition (equation 4), and horizontal gene transfer (HGT), is given alongside all assumptions in the supplementary appendix (equation S1).”

– This entailed a complete reformulation of the supplementary appendix, which is now presented in a simplified format (see Table of Contents on Appendix page 1) as follows:

– Model and assumptions (including final 12-compartment ODE system and describing all underlying modelling choices and assumptions).

– Part 1: Model evaluation and parameterization (including calculations for all outcomes evaluated).

– Part 1: Supplementary results.

– Part 2: Model application, species characterization and parameterization (including expert elicitation and parameter estimates from the literature).

– Part 2: Supplementary results.

1) In general, the model the authors use is highly parameterized. There is a worry about circular reasoning when modelling. Parameters are derived from observations, then a model is constructed to recapitulate this observation. Then similar observations are used to show the model's validity. How did the authors ensure this type of bias was not incorporated into their modelling? This can become a bigger issue with more highly parameterized models, and that is why simpler models are preferable. Another approach is to use cross-validation, although we are not sure how that would work here.Abstract2) The Abstract is vague and somewhat convoluted as written. It did not do the best job of selling the paper and does not convey any specific information.

We agree that simple models are preferable, and that our final model (equation S1) combines a large number of assumptions into a single ODE system (although still fewer than a typical agent-based approach). This was our primary motivation for decomposing the final 12-compartment model into individual sub-models and presenting these gradually through Part 1, in order to evaluate each component ecological interaction separately and understand its impacts on resistance epidemiology.

– By parameterizing all models identically to the same generic pathogen P^R^, by conducting identical univariate and bivariate analyses across models, and by evaluating a range of epidemiological outcomes, we believe that we have isolated and demonstrated epidemiological impacts of different ecological mechanisms that we have considered, and justified their inclusion in our final modelling framework (Figures 1, 2, 3)

By “Parameters are derived from observations, then a model is constructed to recapitulate this observation”, we presume that the reviewers refer to parameters like pathogen prevalence and resistance rates, which are simultaneously model input parameters (community burden) and epidemiological outputs (simulated hospital dynamics). We don’t disagree with the interpretation that there is inherently “circularity” here where model inputs drive outputs. However, we stress that such sink dynamics are a truism for healthcare institutions (see e.g. Blanquart et al. 2019 Evol App https://doi.org/10.1111/eva.12753): baseline prevalence and resistance rates for asymptomatic colonizers like MRSA and Enterobacteria inevitably reflect burden in the surrounding community.

– This is also why colonization incidence was evaluated as the primary outcome for evaluation of public health interventions, as this outcome reflects model dynamics but does not have a corresponding input parameter.

– We now better distinguish between interpretation of these input and output parameters in the caption for Figure 5:

“Dashed lines (model inputs) represent assumed community prevalence, i.e. the proportion of patients already colonized upon hospital admission (see Tables S3-S6). Solid lines represent simulated prevalence within the hospital, as resulting from both importation from the community and within-hospital epidemiology.”

For part 2, we stress our use of multivariate sensitivity analysis (PRCC) to understand the impacts of parameter uncertainty on model outcomes, and our analysis of “single-species” vs. “microbiome” simulations to understand impacts of microbiome parameters on model outcomes

Title [editor request, included here by authors]:

Could you please edit your title so the second part gives clear indication of the study design under investigation, if appropriate, and avoiding abbreviations where possible. The remainder of the title should read as a single, concise sentence containing no dashes or question-answer, and avoiding abbreviations where possible.

We propose the following title:

“Microbiome-pathogen interactions drive epidemiological dynamics of antibiotic resistance: a modelling study applied to nosocomial pathogen control”

Introduction3) The authors need to tailor this to their specific question: where is the knowledge gap? We recommend that the authors think carefully about the statements they are making and consider revising the Introduction to include a more thorough analysis of prior modelling work, which is largely absent from the Introduction.

A more thorough analysis of previous modelling work is now provided as the fourth paragraph of the introduction. In the context of this paragraph, we present the following knowledge gap.

“Accounting for other forms of complexity in epidemiological models – from treatment intensity, to age-assortative contact behaviour, to hospital referral networks, to animal-human interactions, to genetic linkage between resistance and non-resistance genes – has helped to unravel the many, disparate forces that contribute to drive the spread of resistance.(42–47) Nevertheless, within-host competition between co-colonizing bacteria is a key mechanism of selection for antibiotic resistance dissemination, and an active area of research at the forefront of resistance modelling.(34,36,48) For instance, the ‘mixed-carriage’ model by Davies et al. demonstrates how intraspecific competition results in negative frequency-dependent selection for either of two competing strains, and provides a satisfying mechanistic explanation for widespread strain coexistence at the population level.(49) However, contemporary work has stopped short of evaluating consequences of between-species competition on resistance epidemiology. Yet for many ARB, including emerging high-priority multidrug-resistant bacteria like extended-spectrum β-lactamase (ESBL) producing *Enterobacteriaceae*, interactions with the host microbiome may be important mediators of nosocomial colonization dynamics.”

4) The study focuses on four particular pathogens, however, no motivation of why they are chosen is provided, nor the relevance of their analysis in a more general overview is discussed. The authors could expand on how their results could be extended to other contexts/species in which the role of microbiome on resistance is also relevant. This could also be included in the discussion.

We thank the reviewer for this helpful comment which we have taken into account in several instances:

In the introduction, in addition to previous reference to *C. difficile*, we now reference the emerging threat of ESBL-*Enterobacteriaceae* when introducing the knowledge gap motivating this work (see previous comment).

In the discussion, second paragraph:

– Additional context for these species as key drivers of nosocomial infection:

“MRSA, *C. difficile* and ESBL-producing *Enterobacteriaceae* are leading causes of antibiotic-resistant and healthcare-associated infection, while carbapenemase-producing *Enterobacteriaceae* represent emerging threats of particular concern due to limited therapeutic options for effective treatment of invasive infection.” (4,63–65)

“Antibiotic stewardship is a core component of public health efforts to limit the emergence and spread of these ARB in clinical settings,(66) and an important focus of antibiotic resistance modelling.”

– Relevance for other species and settings:

“Our simulations were limited to a select few ARB, but our framework and findings likely have relevance for other bacteria known to interact with the microbiome, including vancomycin-resistant Enterococci and other multidrug-resistant *Enterobacteriaceae*,(23,76) and could be further extended to explore impacts of the microbiome on resistance dynamics and intervention efficacy beyond healthcare settings.”

5) The authors conclude that disruption of the microbiome on an individual level (due to antibiotics) can result in selection for AMR on a population level. In a way, the work is, at least in part, a model-based implementation of the theoretical perspective of Lipsitch and Samore 2002 EID doi: 10.3201/eid0804.010312 (which describes how antibiotic effects on an individual, within-host level, can affect population level transmission dynamics). Could the comparison to Lipsitch and Samore and the novelty of the improved data-driven approach (although still limited) be further discussed?

We agree that our work implements a long-standing hypothesis previously described in the perspective by written by Lipsitch and Samore 2002 *EID*. This is now emphasized first in the introduction:

“Disruption of the host microbiome is a long-standing theory explaining how antibiotics select for the spread of resistance at both the individual and population levels,(34) but most epidemiological models consider just one species of bacteria at a time, under the traditional assumption that antibiotic selection for resistance results from intraspecific competition between co-circulating strains.”

In the Model and Results, as previously:

“….intraspecific strain competition is not the only mechanism by which antibiotic consumption can drive ARB spread,(34) and may have limited relevance for certain species, settings, timescales and epidemiological indicators.”

And anew in the first paragraph of the discussion

“We present a modelling framework that includes within-host ecological costs of antibiotic use in the form of microbiome dysbiosis, incorporating a leading hypothesis for antibiotic selection into classical models of resistance epidemiology.”

We also highlight the following passage from Lipsitch and Samore, which supports the present work:

“The use of mathematical models, and more generally the attempt to predict the relative merits of different interventions, will depend on an improved understanding of the mechanisms of antibiotic selection in particular organisms.”

The novelty of our approach is further emphasized in the discussion, when comparing our approach to previous models assuming increased susceptibility to colonization/infection among antibiotic-exposed individuals:

“These strategies reflect widespread recognition that antibiotic use favours ARB acquisition, through erosion of colonization resistance or other supposed mechanisms, and independent of potential competition with other strains. The present work formalizes examples of the microbiome-pathogen interactions that underlie these assumptions, demonstrating their relevance to various epidemiological outcomes, distinguishing them from strain-based selection, and providing a framework for their application.”

In the second limitations paragraph of the discussion, we now highlight that, despite clear data limitations, our expert elicitation-driven approach provided useful proxy measures that allowed us to characterize the species-specific ecology of different ARB:

“Finally, the nature of microbiome-pathogen interactions and their epidemiological consequences remain poorly understood and largely unquantified. We show in theory why these interactions matter at the population level, but empirical data were unavailable to inform their parameterization. Instead, we translated microbiome-pathogen competition coefficients into clinical parameters, designed a structured expert elicitation protocol and conducted interviews allowing subject-matter experts to quantify their beliefs. Although these estimates are subject to substantial bias and uncertainty, they facilitated species-specific characterization of the epidemiological impact of microbiome dysbiosis, and represent useful proxy measures in the absence of clinical data.”

And in the penultimate paragraph we further discuss our approach and findings in the context of limited data, including perspectives for future empirical work:

“Despite data limitations, epidemiological conclusions from Monte Carlo simulations were largely consistent with empirical findings (discussed above), suggesting that final parameter distributions were reasonable approximations. Uncertainty in parameter inputs translated to uncertainty in model outputs and reflects the knowledge gaps underlying our simulations (Figure S14). This is exemplified by HGT and its highly uncertain role in driving colonization incidence; to date, HGT modelling has largely been limited to within-host dynamics, and impacts on epidemiological dynamics are only just beginning to come to light.(51,57,94) Increasing availability and synthesis of high-quality within-host microbiological data will help to further characterize epidemiological impacts of microbiome-pathogen interactions. Studies are needed that describe ecological impacts of antibiotic exposure on microbiome population structure across control and treatment groups, with longitudinal follow-up evaluating subsequent nosocomial ARB colonization risk. In the absence of clinical data, insights from experiments and within-host models nonetheless suggest that antibiotic disruption of microbiome-pathogen competition is a key driver of selection for resistance.” (95–100).

6) Antibiotic stewardship is treated a single entity, whereas in reality there are countless types of stewardship interventions and a concomitant variety of outcomes. Comparing the model's results directly to an analysis of previous stewardship interventions and attempting to show alignment does not make sense to me. How were these stewardship interventions reflective of the model? Why would we expect agreement or disagreement a priori? The fact that there was agreement with the model was somewhat concerning given the complexity of many stewardship interventions.

– We totally agree that stewardship interventions are highly heterogeneous and thank the reviewer for raising this point. Heterogeneity likely drives high uncertainty in efficacy estimates from the meta-analysis by *Baur* et al., which groups interventions together as a single entity. Despite this high heterogeneity, stewardship interventions – whether reducing rates of antibiotic consumption, favouring lower-spectrum antibiotics, decreasing treatment duration, or otherwise limiting patient exposure to unnecessary antibiotics – should in theory all reduce population-level antibiotic-induced dysbiosis (and hence reduce microbiome-induced selection for resistance)

– Owing to our ODE-based approach, we were unable to reproduce highly granular stewardship interventions that reflect real-world heterogeneity, but we nonetheless simulated 3 types of stewardship, and found similar efficacy estimates (1 – IRR) for each (Appendix figure 17), which were also concordant with pooled estimates from the meta-analysis

– We agree that visualizing validation of model results to published results from Baur et al. was perhaps not appropriate, and have removed those estimates from Figure 6. Instead, we make comparison to their findings in the discussion:

“By contrast, simulated antibiotic stewardship interventions were broadly effective for reducing incidence across all included ARB. This is consistent with findings from a meta-analysis of clinical trials evaluating the efficacy of hospital antibiotic stewardship interventions for reducing incidence of ARB colonization and infection,(66) which we updated to exclude studies co-implementing stewardship with alternative interventions (Figure S18). In comparison to our findings, estimates from the meta-analysis were associated with greater uncertainty across more heterogeneous interventions, but predicted the same rank order of efficacy across included ARB, and similar mean efficacy for *C. difficile* (19% from n=7 studies, vs. 18% under intermediate compliance in our simulations), ESBL-EC (18% from n=4 studies, vs. 15%) and MRSA (12% from n=10 studies, vs. 10%), but higher efficacy for CP-KP (54% from n=1 study, vs. 20%).”

Methods7 In attempting to replicate this study, it became intractably difficult to recreate the ODEs because the equations and variables are interspersed throughout the main text and supplementary appendix. For example, in Figure 1e the authors mention differences in resistance levels (γ) and not transmission rate (λ, related to β as in Equation (3)), the value of which could not be found anywhere in the text. Similarly, where is patient demography (Δ) incorporated into equations 3 or 4? There are certainly answers to these questions, but it would have been helpful to have the model more clearly presented in a central location. Finding the parameters and making sense of them was extremely challenging, if not infeasible. Therefore, we were unable to replicate the results to any extent.

We appreciate that some aspects of our model were unclear to reviewers and regret any confusion caused. We have taken a number of steps to improve the clarity of our modelling assumptions and to improve reproducibility of our work:

– We provide equations for each of the analyzed sub-models in the manuscript (the strain-microbiome competition was previously in the supplement), more explicitly reference them in the text (equations 1, 3, 4 and 6, respectively), and include δ (demography) terms throughout

The final complete model and all of its underlying assumptions – the ‘framework’ – is now provided at the beginning of the supplement, delineating all assumptions, composite parameters and strain-specific differences. This final framework is referenced on multiple occasions in the manuscript, including:

– In the first paragraph of Methods and Results:

“See Methods for technical details, and the supplementary appendix for the complete modelling framework and assumptions (equation S1).”

– After introduction of HGT, to clarify where the final model including HGT can be found:

“(See final model and assumptions, Appendix equations 1 – 11.)”

– At the beginning of the methods, to clarify our approach of building subsequent sub-models together to produce our final framework:

“We evaluated ODE systems describing colonization dynamics of bacterial pathogens in the healthcare setting. The final model, comprising pathogen colonization (equation 1), intraspecific strain competition (equation 3), microbiome-pathogen competition (equation 4), and horizontal gene transfer (HGT), is given alongside all assumptions in the supplementary appendix (equation S1).”

In addition to the previously available Mathematica code, R code has been made available for the 5 ODE systems at https://github.com/drmsmith/microbiomeR, as referenced in the Methods:

“(See the supplementary appendix for technical details, and R and Mathematica files available online at https://github.com/drmsmith/microbiomeR.)”

In the caption for Figure 1E we have defined strain-specific differences in force of infection, as suggested:

“For E, P^S^ and P^R^ circulate simultaneously, assuming strain-specific differences in antibiotic resistance (*r_S_* = 0, *r_R_* = 0.8), natural clearance (γ_S_ = 0.03 day^-1^, γ_R_ = 0.06 day^-1^) and transmission (λ_S_ = β × C^S^/N, λ_R_ = β × C^R^/N).”

8) It is unclear how the authors arrived at their parameter values throughout the manuscript. For example, colonization resistance (epsilon) as it is formulated is not compelling, and where did the value of this parameter come from? It is listed in Table S5 as being drawn from a Cauchy distribution. Why? Where did these numbers come from? Some of the parameters seem justified by literature, but the rest are perhaps debatable. This might be inevitable with a complex model, but the authors should minimally provide a robustness analysis to each of their less-supported parameter choices. What would be the outcome if the sampled distributions had been wider (e.g., if the true value had been different by an order of magnitude)?

We thank the reviewer for raising this important point and have taken a number of steps to address it. First, by explicitly splitting the work into 2 parts, we hope that it is now more clear that in Part 1 we evaluate impacts of different within-host ecological interactions on a generic nosocomial pathogen P^R^ – hence with arbitrary parameter values, but using univariate and bivariate analysis to explore the epidemiology of this theoretical pathogen across wide parameter space – and that in Part 2 we tailored our model to particular ARB using species-specific parameter estimates from the literature and expert opinion

For consistency, we have introduced concepts of *colonization resistance* and *ecological release* – both well-established in cited literature – earlier in the manuscript:

“The microbiome can also protect against colonization with infectious bacterial pathogens, a phenomenon known as colonization resistance, limiting their capacities to establish colonies, grow, persist and transmit.”

“Antibiotic-induced dysbiosis may further result in elevated expression of antibiotic resistance genes, increased rates of horizontal transfer of resistance genes, and ecological release, whereby subdominant ARB are released from competition via clearance of drug-sensitive bacteria, growing out into dominant colonies.”

We now also discuss limitations of our conceptualization of these interactions in our modelling framework:

“More broadly, our characterizations of microbiome-pathogen interactions are conceptual, and were mapped mechanistically to particular colonization processes (transmission, clearance, endogenous acquisition), but we note that in other contexts terms like *colonization resistance, resource competition* and *ecological release* may map to specific biochemical processes that could affect epidemiological parameters in different ways.”

We have taken a number of steps to more clearly describe parameterization of microbiome-pathogen interaction coefficients and our use of expert elicitation:

– The protocol document for our expert elicitation exercise is provided as a separate supplementary file, an exact copy of which was used by experts during interviews.

– Use of expert elicitation to inform model parameters is now referenced as early as the introduction:

“Expert elicitation interviews were conducted to characterize the clinical relevance of microbiome dysbiosis for each species, and to qualify and quantify interaction coefficients with uncertainty.”

– In the Model and Results section of the manuscript, we more clearly reference the expert elicitation and what it was used for, with reference to Methods and relevant sections of the supplement:

“Data from the literature were used to parameterize the model to each pathogen (Figure 4, Tables S2 – S6). Literature estimates for microbiome-pathogen interaction coefficients are scarce, and the species-specific relevance of different within-host interactions (intraspecific strain competition, microbiome competition, HGT) in the hospital environment are not well-defined. To characterize ecological interactions for each species and inform model structure, we conducted interviews with a panel of subject-matter experts in medical microbiology and antibiotic resistance epidemiology (details in Methods). Based on their beliefs, all pathogens were assumed to compete with microbiota; MRSA, ESBL-EC and CP-KP were further assumed to compete intra-specifically with non-focal strains, for simplicity characterized as methicillin-sensitive *S. aureus* (MSSA), *E. coli* (EC) and *K. pneumoniae* (KP); and both ESBL resistance and carbapenem resistance were assumed to be borne by plasmids capable of horizontal transfer between patient microbiota and, respectively, EC and KP (Figure 4A). To quantify species-specific strengths of microbiome-pathogen interactions, within-host ecological parameters were translated into clinical parameters (Table S7), and experts were asked to quantify these using standardized expert elicitation methodology (Figures 4B, S8 – S10).”

– In the Methods, description of the expert elicitation has been condensed into one succinct paragraph for clarity, while full methodological details have been provided in the corresponding section of the supplement (Appendix section 4.2, page 21), where we:

– Describe expert elicitation and our protocol.

– Describe elicitation results and their interpretation. Here we also detail how final pooled distributions were found (including the Cauchy distribution listed in Table S5 and referenced by the reviewer):

“Raw data for expert distributions are provided in Figure S9. Experts differed substantially in estimated ranges of different parameters, but Friedman’s tests using medians of each parameter distribution suggested that the species rank order was conserved across experts, with *C. difficile* generally having the strongest estimated microbiome-pathogen interaction coefficients, and MRSA the weakest. To conserve species rank order when combining expert estimates *x_i,j,k_* to form pooled histograms, distributions were re-centred by the mean relative distance *z_i,j_* between each *j* and the reference ARB (taken as MRSA) over all experts *k* (Figure S10). For *C. difficile* and MRSA, HGT was excluded from simulations and pooled distributions were not re-centred. Each expert distribution was weighted equally, contributing to 10% of the final pooled distribution for each species and parameter. Pooled histograms were fit to six candidate distributions (normal, log-normal, Weibull, Cauchy, exponential and γ), and the final distribution was selected as the fitted distribution giving the lowest Akaike Information Criterion (using the function fitdist from the R package fitdistrplus). Final fitted distributions for each parameter and ARB are given in Tables S3-S6, and for select parameters in Figure 4.”

– Provide Table S7, as previously, which translates the clinical parameters experts estimated into microbiome-pathogen interaction coefficients.

– Provide Figure S8, as previously, showing expert opinion about the clinical relevance of different clinical parameters for different species.

– Provide Figure S9, the raw data resulting from the expert elicitation, i.e. the distributions created by each expert for each microbiome-pathogen interaction for each species.

– Provide Figure S10, final re-centred data used to generate pooled distributions while maintaining species rank order.

We agree that alternative model parameterization for each species would change modelling results and conclusions, but stress that the goals of Part 2 – evaluation of species-specific colonization dynamics and intervention efficacy – necessitates species-specific parameterization, which we estimated to the best of our ability given available data, while:

– accounting for parameter uncertainty;

– conducting multivariate sensitivity analysis with PRCC to quantify impacts of parameter uncertainty on model outcomes;

– conducting simulations without microbiome interactions, to show their impact on outcomes.

A key result of these analyses is that alternative species-specific model parameterization leads to different modelling results and estimates for intervention efficacy, and that these are further impacted by including microbiome-pathogen interactions. We believe that additional analysis arbitrarily adjusting parameter distributions would not be appropriate:

– Widening distributions for all parameters should produce approximately the same median results with inflated uncertainty across outcomes.

– Univariate widening of parameter distributions is unnecessary given PRCC results (Appendix figure 14, page 30), which already convey which parameters have greatest impact on outcomes, and hence which parameters should most increase or decrease resistance if adjusting the value.

– Arbitrarily adjusting certain parameters goes against the stated goal of species-specific characterization. For example, it would be possible to run simulations with a high transmission rate for *E. coli* or low microbiome-pathogen interaction coefficients for *C. difficile*, and outcomes would certainly change substantially. But these simulations would no longer represent these pathogens, and it is not clear to us that worthwhile insights would be gained that are not already demonstrated by PRCC results, by comparisons between the four included ARB, and by ‘single-species’ vs. ‘microbiome’ simulations

We nonetheless agree that imperfect parameterization is a key limitation to this study, and dedicate a new paragraph to discuss this, highlighting limited availability of several parameters in the literature, uncertain generalizability of results across settings, sensitivity of model conclusions to assumed parameters, the conceptual nature of our characterization of microbiome-pathogen interactions, and the gut-specific nature of our microbiome dysbiosis scale (which could potentially bias results for predominantly skin-residing *S. aureus*).

Finally, for completeness, we re-ran our multivariate sensitivity analysis conducted previously (Latin Hypercube Sampling + PRCC) for an additional outcome: the pathogen resistance rate (Appendix figure 14B, page 30). Findings for both outcomes are now presented together in the Model and Results section:

“In multivariate sensitivity analysis, community prevalence (*f_C_* for *C. difficile*, *f_R_* for others) and rates of endogenous acquisition (α_R_) had overall the strongest positive impacts (highest PRCCs) on hospital prevalence across ARB, while rates of hospital admission/discharge (μ) and microbiome recovery (δ) had the strongest negative impacts (lowest PRCCs; Figure S14A). For resistance rates, parameters with the strongest positive impacts (highest PRCCs) were community prevalence (*f_R_*), the rate of endogenous acquisition (α_R_) and the rate of antibiotic-induced pathogen clearance (σ_C_). Parameters with the strongest negative impacts (lowest PRCCs) were rates of hospital admission/discharge (μ) and endogenous acquisition of competing drug-sensitive strains (α_S_) (Figure S14B). Across ARB, prevalence estimates, but not resistance rates, were generally sensitive to microbiome parameters.”

9) The inclusion of the 'r' parameter, allowing for partial resistance in the resistant strain, is interesting (and innovative for modelling frameworks to our knowledge). Nonetheless, it is not clear why the authors use a baseline of r = 0.8 to illustrate the trade-off between antibiotic induced clearance and selection for Cr (Figure 1F), whereas in figure 2, the authors use r=0.4, i.e. higher levels of sensitivity in Cr. I suppose with a lower r, strain coexistence is more likely, and thus higher likelihood of Horizontal Gene Transfer (HGT) (?), but it would be, for consistency and comparison of the different within-host dynamics, more transparent to use the same baseline parameter values, unless the authors can provide a good reason why different values of r are assumed at baseline between Figure 1 and Figure 2B?

We thank you for raising this and agree that identical parameter values across plots for Part 1 makes for more consistent and transparent results. We have now reproduced all Figures using the same default parameter set (Appendix table 1 on page 11, results also provided in R code), such that all results for Part 1 now represent the same generic pathogen P^R^ (adjusting only the parameters explicitly made to vary for univariate or bivariate analysis in each figure). In particular, the previous Figure 2B has been expanded into a new Figure 3, which includes the default value of *r_R_*=0.8, as well as results for lesser resistance levels (*r_R_* = 0.2, 0.5) to demonstrate how different epidemiological outcomes (pathogen prevalence and resistance rate) vary with this important parameter.

10) The model presented in Eq. (1) needs a better introduction. Is this a new model or based on other models previously studied? This is explained later in the text citing references (30) and (31), but we recommend to introduce them earlier. It would be also good to point the reader earlier in the main text to the first sections of the supplement for a better understanding of the model assumptions and parameters employed.

This model is adapted from classic colonization models of antibiotic-resistant bacteria – the first example to our knowledge being Austin and Anderson *Proc Biol Sci* 1997 – with the key difference being our novel approach to modelling antibiotic treatment. We now introduce this model in this context, including the following:

“We start with a Susceptible-Colonized transmission model (Figure 1A) representing a population of N hospitalized patients as either susceptible to colonization (S) or colonized (C^R^) by P^R^, the focal strain or species: *[equations]* This model is adapted from classic colonization models of antibiotic-resistant bacteria,(51) includes no ecological interactions with non-focal bacteria, and reflects a suite of common assumptions relevant to the healthcare setting, including: …”

And subsequently highlighting how we model antibiotic exposure and its impacts on pathogen clearance:

“Modelling the latter as a continuous proportion reflects that bacteria are not necessarily fully drug-sensitive (*r_R_* = 0) nor -resistant (*r_R_* = 1), but can range in their sensitivity to different antibiotics (0 ≤ *r_R_* ≤ 1). The resistance level *r_R_* is thus a model input interpreted as an overall measure of the pathogen’s innate degree of resistance to the particular antibiotics to which it is exposed.”

Finally, we place this first model in the context of subsequent models

“Following models build upon these assumptions, representing the same pathogen P^R^, but altering its ecological interactions with other bacteria from one model to the next. Models were evaluated over the same generic parameter space, to isolate impacts of model structure on epidemiological outcomes in the context of antibiotic use (see parameters in Table S1).”

11) Also, should the parameters be more consistently labelled across frameworks? For example, we recommend writing λ(N,C), or something similar, in all the equations to explicitly state it depends on these parameters. The way it is written gives the impression that λ is a constant parameter.13) Furthermore, in figure 1F: this is representing a one strain model. Is the Ce+Cd strain similar to the Cr? Can this be clarified?14) In line with this. In Figure S3, R0 seems to represent the R0 of Cr. However, for Figure 1H and 1I, how should R0 be interpreted respectively? And for Figure S2? Please clarify how similar Cr (Figure 1H) and the one strain modelled (which could be similar to the Cr strain of Figure 1H) for figure 1I are

We respond to these 3 comments simultaneously: yes, the focal antibiotic-resistant pathogen (previously denoted C in model 1, C^R^ in model 2 and C_e_+C_d_ in model 3) is identical, and we understand the previous confusion. We thank the reviewer for raising these points and we have consequently made changes to clarify notation

We have rewritten the manuscript to explicitly distinguish between the pathogen in question and patient colonization with that pathogen: drug-sensitive and -resistant pathogen strains are denoted P^S^ and P^R^, respectively, while patient colonization with those strains is denoted C^S^ and C^R^

All ODE systems are rewritten to describe dynamics of C^R^, i.e. all systems describe colonization with the same focal antibiotic-resistant pathogen P^R^. Epidemiological parameters for P^R^ are always denoted by subscript R (e.g. α_R_). Where strain competition is also included, ODE systems include C^S^ (representing patient colonization with the strain P^S^), with epidemiological parameters for P^S^ denoted by subscript S (e.g. α_S_)

We now make the reader aware of these notations several times early in the paper. For instance we:

– Introduce P^R^ from the first sentence of the Model and Results:

“We propose a series of five models describing colonization dynamics of a focal antibiotic-resistant bacterial *pathogen*, denoted P^R^, among hospital inpatients in an acute care setting. Each model accounts for different within-host ecological interactions between P^R^ and other bacteria.”

– Distinguish between P^R^ and C^R^ upon introduction of the first model:

“We start with a Susceptible-Colonized transmission model (Figure 1A) representing a population of N hospitalized patients as either susceptible to colonization (S) or colonized (C^R^) by P^R^, the focal strain or species:”

– Distinguish between P^S^ and P^R^ upon introduction of strain competition:

“… selection results from intraspecific competition between two or more drug-sensitive strains P^S^ and drug-resistant strains P^R^.”

– And again upon introduction of the strain competition model:

“A simple two-strain ‘exclusive colonization’ model (Figure 1B) can be written as: [equations] where patients can be colonized (C^S^, C^R^) by either strain (P^S^, P^R^).”

– And so on throughout the manuscript.

We further make explicit that we are always evaluating the same P^R^ across models using the same parameter set, changing only its within-host interactions with other bacteria across models:

“Following models build upon these assumptions, representing the same pathogen P^R^, but altering its ecological interactions with other bacteria from one model to the next. Models were evaluated over the same generic parameter space, to isolate impacts of model structure on epidemiological outcomes in the context of antibiotic use (see parameters in Table S1).”

We have updated model diagrams in Figure 1 to represent these changes

We understand that not all readers may be familiar with the classic force of infection term λ, but in the final model λ already has up to 2 subscripts (λ_S,ε_). We believe that adding more (e.g. λ_S,ε,C,N_) could unnecessarily complexify model interpretation. Instead, we have proposed clear definitions of λ in 3 instances:

– as previously, upon introduction of model 1;

“(iii) a dynamic rate of colonization acquisition λ_R_=β⨉C^R^/N, for host-to-host transmission.”

– as suggested by reviewers, in the caption for Figure 1 for the strain competition model:

“For E, P^S^ and P^R^ circulate simultaneously, assuming strain-specific differences in antibiotic resistance (*r_S_* = 0, *r_R_* = 0.8), natural clearance (γ_S_ = 0.03 day^-1^, γ_R_ = 0.06 day^-1^) and transmission (λ_S_ = β × C^S^/N, λ_R_ = β × C^R^/N).”

– in the supplement, as the final generalized expression for any strain *j.* (Appendix equation 3 page 3)

12) The r vs resistance rate (Cr/Cr+Cs) caused some confusion. This as a resistance rate of Cr = 0.8 but an r = 0.4 could actually mean that 0.8*0.4 = 0.08 of pathogens carried are fully resistant against the antibiotic, while this is 0.64 when r=0.8. Can the same baseline values for r be used in Figure 1 and Figure 2 or the differences justified?

We regret the confusion. The same baseline value *r_R_*=0.8 is now used throughout.

To clarify interpretation of the parameter *r_R_* (and *r_S_*), the following text has been added upon its introduction in the first model:

“…and *r_R_* is the antibiotic resistance level (the proportion of antibiotics that are ineffective against P^R^). Modelling the latter as a continuous proportion reflects that bacteria are not necessarily fully drug-sensitive (*r_R_* = 0) nor -resistant (*r_R_* = 1), but can range in their sensitivity to different antibiotics (0 ≤ *r_R_* ≤ 1). The resistance level *r_R_* is thus a model input interpreted as an overall measure of the pathogen’s innate degree of resistance to the particular antibiotics to which it is exposed.”

And we have also more clearly defined the resistance rate in the first paragraph of Model and Results:

“Across models, three primary epidemiological outcomes are calculated at steady-state equilibrium: P^R^ prevalence (the proportion of patients colonized), P^R^ incidence (the daily rate of colonization acquisition within the hospital), and the pathogen resistance rate (the proportion of patients colonized with the focal antibiotic-resistant strain P^R^ relative to a competing drug-sensitive strain P^S^).”

We have kept the initial name used for the *r_R_* parameter: the antibiotic resistance level. This name was chosen to avoid confusion with alternative possible names like the antibiotic resistance rate/proportion/frequency/share (which are used elsewhere to describe the share of strains of a species – or the share of clinical isolates – that are drug-resistant, as above). However, we understand that interpretation of this novel parameter may not be obvious and we propose the following alternatives to the reviewers and editors: antibiotic resistance coverage, antibiotic resistibility, antibiotic resistance degree, innate antibiotic resistance level/degree.

– It is not clear to us that any of these alternatives improve understanding, but our goal is to maximize clarity and interpretability of this important parameter, and we welcome any feedback and are happy to update it, if so desired by the editorial team.

15) Figure 2A, the minimum resistance rate is 0.35 (see legend, and this appeared the case when no antibiotic induced clearance nor antibiotic induced microbiome disruption). This seems rather high. In particular as in Figure 2B, minimum levels of less than 0.1 are shown. Could the authors explain where this discrepancy is coming from and to what extend these high baseline resistance rates are representative?

For ease of interpretation for the generic pathogen P^R^ in part 1, in our baseline parameter set (Table S1) we now assume that 50% of colonized individuals admitted from the community are colonized with the resistant strain (*f_R_* = 0.5): when the steady-state resistance rate exceeds 50%, it can easily be understood that the hospital has provided a competitive advantage to the resistant strain despite assumed costs of resistance. We agree that this is a higher resistance rate than many real-world pathogens, examples of which are provided in Part 2 (see for e.g. Figure 5B).

In the previous Figure 2B, minimum levels of C^R^ prevalence were <0.1 (not the resistance rate, which was not shown), but we acknowledge that this may have been confusing as the adjacent Figure 2A presented both resistance rate and prevalence. We have now separated these figures: Figure 2A is presented as Figure 2; and Figure 2B has been expanded into a new Figure 3, which includes both prevalence and resistance rates to avoid potential misinterpretation, and to demonstrate asymmetry in these outcomes. We further demonstrate how outcomes related to HGT depend on the antibiotic resistance level *r_R_* (columns). We have also added translucent grey bars to this figure representing assumed community P^R^ colonization prevalence (*f_C_* × *f_R_*) and resistance rate (*f_R_*), to help the reader to contextualize results against baseline assumptions.

16) For parameterisation of the models for the different pathogens, estimates from literature, notably existing modelling studies are chosen. However, these estimates, at least for C. difficile and MRSA, are coming from models that don't incorporate endogenous acquisition explicitly (for the Enterobacterieacia, model estimates from Gurieva are used, which do use a modelling framework incorporating an exo- and endogenous acquisition). Therefore, the acquisition rates may be overestimated for these gram-positives, in particular for C. diff the endogenous acquisition, which is listed as the main acquisition route (pp 14 line 330).This may affect the estimated intervention effectiveness (in particular for antibiotic stewardship (less effective) and contact precautions (more effective)) under assumptions of the microbiome model. Could the authors use more realistic estimates (not sure models with explicit exo- and endogenous acquisition exist for MRSA and C. diff), or at least, reflect on how different values of α and β affect the model results?

Unfortunately, we are also unaware of models with explicit exo-endo estimates for the gram-positives, but we now dedicate substantial discussion to this limitation and discuss relevance to PRCC results from our sensitivity analysis:

“Third, Monte Carlo simulations were limited by the availability of species-specific model parameters from the literature, in some instances necessitating use of previous modelling results, approximations, or estimates from small studies in specific locations, making the generalizability of results unclear. For instance, Khader *et al.* estimated a four-fold difference in MRSA transmission rates between hospitals and nursing homes.(79) Such differences could have a substantial impact on dynamics and estimated intervention efficacy, with higher transmission rates favouring use of contact precautions, and higher rates of endogenous acquisition favouring antibiotic stewardship (in the context of a high ecological release coefficient). Uncertainty in endogenous acquisition rates may be particularly important: in multivariate sensitivity analyses, this parameter emerged as a key driver of both colonization prevalence and resistance rates across ARB (Figure S14).”

17) It would be helpful to explicitly say how the simulations were performed, i.e., mention that the ODEs where integrated. My first impression was that the authors performed stochastic simulations using the processes illustrated in panels A, B, C from Figure 1. This is probably an issue that only mathematical modellers would have, but could the authors please add this clarification.

We agree that more details on the modelling and simulations were missing and thank the reviewer for raising that point. This is now made explicit in the new introductory paragraph to the Methods and Results section:

“Models are described using systems of ordinary differential equations (ODEs) and are evaluated deterministically by numerical integration. Across models, three primary epidemiological outcomes are calculated at steady-state equilibrium: P^R^ prevalence (the proportion of patients colonized), P^R^ incidence (the daily rate of colonization acquisition within the hospital), and the pathogen resistance rate (the proportion of patients colonized with the antibiotic-resistant strain P^R^ relative to a drug-sensitive strain P^S^). We also derive and evaluate the basic reproduction number *R*_0_, an indicator of pathogen epidemicity representing the average number of patients expected to acquire a novel pathogen from an initial index patient. See Methods for technical details, and the supplementary appendix for the complete modelling framework and assumptions (equation S1).”

In the caption for Figure 1:

“For all models, ODEs are integrated numerically using the same parameter values representing a generic nosocomial pathogen P^R^ (see Table S1).”

And again, in the methods:

“ODEs were integrated numerically to calculate steady-state epidemiological outcomes for nosocomial P^R^ colonization: colonization prevalence (the sum of all compartments C^R^), colonization incidence (the daily rate of C^R^ acquisition), and the resistance rate (C^R^/(C^S^ + C^R^)). (See the supplementary appendix for technical details, and R and Mathematica files available online at https://github.com/drmsmith/microbiomeR.) For each model, outcomes were evaluated over the same parameter space representing a generic pathogen P^R^ (parameters in Table S1), while varying specific parameters through univariate and bivariate analysis to assess their impacts on dynamic equilibria in the context of different modelling assumptions.”

Discussion18) The model is described by a set of differential equations. This means the model ignores stochasticity of the population size dynamics of each strain. The authors could expand a bit more on the assumptions of the model and why fluctuations are ignored.

We have expanded upon these limitations as suggested, nuancing our statements with the intended goals of this study:

“This study has a number of limitations. First, hospitals and healthcare settings are heterogeneous environments with non-random contact patterns and relatively small population sizes. Stochastic, individual-based models accounting for these factors reproduce more realistic nosocomial transmission dynamics than deterministic ODE simulations, allowing for local extinction events, super-spreaders, and other inherently random epidemiological phenomena. Nonetheless, our goal was to study how ecological mechanisms impact average epidemiological outcomes in the context of different model assumptions and parameter uncertainty, and in this context ODE modelling was the more appropriate tool, particularly for widely endemic ARB like *C. difficile*, MRSA and ESBL-EC. Still, further insights could certainly be gained by accounting for additional complexity and stochastic heterogeneity in future work, from within-host spatial organization,(91) to patient and staff contact behaviour,(92) to inter-institutional or inter-ward meta-population dynamics.(93) These distinctions may be particularly important for rare or non-endemic ARB (e.g. CP-KP in some regions).”

19) Also, spatial structure is particularly important when taking into account interactions between microbiome and pathogens, but it is also neglected in the model. It would be useful if the authors could discuss this as well.

Importance of spatial structure to resistance dynamics is now referenced in the introduction:

“Accounting for additional forms of complexity in epidemiological models – from treatment intensity, to age-assortative contact behaviour, to hospital referral networks, to animal-human interactions, to genetic linkage between resistance and non-resistance genes – has helped to unravel the many, disparate forces that contribute to drive the spread of resistance.”

and in the discussion:

“Still, further insights could certainly be gained by accounting for additional complexity and stochastic heterogeneity in future work, from within-host spatial organization,(91) to patient and staff contact behaviour,(92) to inter-institutional or inter-ward meta-population dynamics.(93) These distinctions may be particularly important for rare or non-endemic ARB (e.g. CP-KP in some regions).”

20) The results shown often focus on steady-state quantities (such as colonisation prevalence), but the temporal evolution of the model is not discussed. It would be interesting if the authors could investigate the timescale of the different interactions taken into account, and how relevant they may be in the healthcare settings they investigate. Is the temporal evolution of the model any relevant for their conclusions?

We include a new supplementary figure which evaluates P^R^ prevalence and resistance rate over time for the generic pathogen P^R^ evaluated in part 1, separating and then combining each microbiome-pathogen interaction to assess their impacts on dynamics (Appendix Figure 5). We included three public health interventions, implemented at days 90, 180 and 270, to explore how pathogen responses to interventions vary with time. Associated code is provided in the R file. We include the following text in the main text with regards to these findings:

“Different microbiome-pathogen interactions also underlie distinct dynamic responses to theoretical public health interventions. For the same generic pathogen P^R^, antibiotic stewardship interventions generally, but do not always prevent colonization, with predictions depending on the ecological interactions in effect (e.g. strain competition, microbiome competition), the impact of the intervention (e.g. reduced microbiome disruption, reduced overall prescribing), and the epidemiological outcome considered (e.g. colonization prevalence, resistance rate) (Figure S5).”

As well as the following text in the supplement:

“In Figure S5, we simulate dynamic responses of P^R^ to public health interventions. First, dynamic equilibria were found, from which ODEs were integrated for an additional 365 days, evaluating C^R^ and the resistance rate over time in the context of each microbiome-pathogen interaction (separately and in concert). We introduced theoretical interventions at days 90 (halving *r_R_* from 0.8 to 0.4), 180 (halving θ_m_ from 1.0 to 0.5), and 270 (halving *a* from 0.2 to 0.1), demonstrating that an otherwise identical pathogen can experience diverse, and sometimes opposing responses to public health interventions in the context of different microbiome-pathogen interactions and epidemiological outcomes.”

Systems generally returned to equilibria after 1 to 3 months. However, we avoid over-interpretation of timescales and lags in the manuscript, owing to limitations to using deterministic ODE modelling for predictive modelling in the hospital setting (see comments 18 and 19).

21) Can the authors please add examples of the hypothetical microbiome recovery interventions they have in mind? Are the authors thinking of things like faecal transplantation? Please add, also to provide more practical interpretation of the work.

We now include a statement in the introduction more clearly referencing these interventions:

– From a clinical perspective, this motivates a need for public health interventions that minimize or reverse harm to patient microbiota, from antibiotic stewardship, to faecal microbiota transplantation, to microbiome protective therapies. (28,29).

And devote a new discussion paragraph to discuss these interventions and relevance for our modelling framework and findings.

– Findings also suggest promise for interventions that effectively restore microbiome stability and associated colonization resistance as a means to control ARB spread. Fecal microbiota transplantation is already used to treat recurrent *C. difficile* infection, and is under investigation for multidrug-resistant *Enterobacteriaceae* decolonization.(67–69) However, its appropriateness for dysbiosis recovery in the absence of other clinical indications is unclear. Transplantation requires rigorous donor screening and close longitudinal follow-up, and cases of donor stool contaminated with toxicogenic and multidrug-resistant bacteria highlight non-negligible risks.(70,71) Alternative microbiome protective therapies now exist, like DAV132, a novel activated-charcoal product currently undergoing clinical trials. When co-administered with antibiotics by the oral route, DAV132 has been shown to absorb antibiotic residues in the colon and preserve the richness and composition of intestinal microbiota, while maintaining systemic antibiotic exposure.(28,72,73) Modelling has been used previously to evaluate impacts of such microbiome-oriented interventions at the within-host level,(74,75) but knock-on impacts on ARB transmission dynamics and epidemiological burden have not been evaluated previously. Our simulations were limited to a select few ARB, but our framework and findings likely have relevance for other bacteria known to interact with the microbiome, including vancomycin-resistant Enterococci and other multidrug-resistant *Enterobacteriaceae*,(23,76) and could be further extended to explore impacts of the microbiome on resistance dynamics and intervention efficacy beyond healthcare settings.